# Ice-supersaturated air masses in the northern mid-latitudes from regular in-situ observations by passenger aircraft: vertical distribution, seasonality and tropospheric fingerprint

Andreas Petzold[1], Patrick Neis[1,3,$], Mihal Rütimann[1], Susanne Rohs[1], Florian Berkes[1,§],
Herman G.J. Smit[1], Martina Krämer[2,3], Nicole Spelten[2], Peter Spichtinger[3], Philippe Nedelec[4],
and Andreas Wahner[1],

[1] Forschungszentrum Jülich GmbH, Institute of Energy and Climate Research 8 – Troposphere, Jülich, Germany

[2] Forschungszentrum Jülich GmbH, Institute of Energy and Climate Research 7 – Stratosphere, Jülich, Germany

[3] Johannes Gutenberg University, Institute for Atmospheric Physics, Mainz, Germany

[4] CNRS Laboratoire d'Aérologie, and Université Paul Sabatier Toulouse III, Toulouse, France

[$] now at CGI Deutschland B.V. & CO. KG, Frankfurt, Germany

[§] now at P3 solutions GmbH, Aachen, Germany

*Correspondence to*: Andreas Petzold (a.petzold@fz-juelich.de)

**Abstract.** The vertical distribution and seasonal variation of water vapour volume mixing ratio ($H_2O$ VMR), relative humidity with respect to ice ($RH_{ice}$) and particularly of regions with ice-supersaturated air masses (ISSR) in the extratropical upper troposphere and lowermost stratosphere are investigated at northern mid-latitudes over the regions Eastern North America, the North Atlantic and Europe for the period 1995 to 2010. Observation data originate from regular and continuous long-term measurements on board of instrumented passenger aircraft in the framework of the European research program MOZAIC (1994 – 2010) which is continued as European research infrastructure IAGOS (from 2011). Data used in our study result from collocated observations of $O_3$ VMR, $RH_{ice}$ and temperature, and $H_2O$ VMR deduced from $RH_{ice}$ and temperature data. The in-situ observations of $H_2O$ VMR and $RH_{ice}$ with a vertical resolution of 30 hPa (< 800 m at the extratropical tropopause level) and a horizontal resolution of 1 km resolve detailed features of the distribution of water vapour and ice-supersaturated air relative to the thermal tropopause, including their seasonal and regional variability and chemical signatures at various distances from the tropopause layer. Annual cycles of the investigated properties document highest $H_2O$ VMR and temperatures above the thermal tropopause in the summer months, whereas $RH_{ice}$ above the thermal tropopause remains almost constant in the course of the year. Over all investigated regions, upper tropospheric air masses close to the tropopause level are nearly saturated with respect to ice and contain a significant fraction of ISSR with a distinct seasonal cycle of minimum values in summer (30% over the ocean, 20 - 25% over land), and maximum values in late winter (35 - 40% over both land and ocean). Above the thermal tropopause, ISSR are occasionally observed with an occurrence probability of 1.5±1.1%, whereas above the dynamical tropopause at 2 PVU, the occurrence probability increases 4-fold to 8.4±4.4%. In both tropopause-height (TPH) related coordinate systems, the ISSR occurrence probabilities drop to values below 1% for the next higher air mass layer with pressure levels $p < p_{TPH} – 15$ hPa. For both tropopause definitions, the tropospheric nature or fingerprint, respectively, based on $O_3$ VMR, indicate continuing tropospheric influence on ISSR inside and above the

respective tropopause layer. For the non-ISSR, however, the stratospheric nature is clearly visible above the thermal tropopause whereas above the dynamical tropopause the air masses show still substantial tropospheric influence. For all three regions, seasonal deviations from the long-term annual cycle of ISSR occurrence show no significant trends over the observation period of 15 years, whereas a statistically significant correlation between the North Atlantic Oscillation (NAO) index and the deviation of ISSR occurrence from the long-term average is observed for the North Atlantic, but not for the regions Eastern North America and Europe.

## 1 Introduction

Relative humidity over ice and in particular ice-supersaturated air masses ($RH_{ice} > 100\%$) are of ample importance for the occurrence and life cycle of high ice clouds, or cirrus clouds, respectively, which have a large but still not fully understood impact on Earth's climate, with its net radiation impact being unknown and even the sign being unclear (Chen et al., 2000; Boucher et al., 2013). In this context, long-term observations of water vapour properties are an indispensable prerequisite for the investigation of potential changes of its abundance in the global upper troposphere and lowermost stratosphere (e.g., Müller et al., 2016), and the resulting effects on atmospheric radiation (e.g., Riese et al., 2012) as well as on cirrus cloud occurrence and life cycle (Gettelman et al., 2012; Krämer et al., 2016; Heymsfield et al., 2017).

The extratropical upper troposphere and lowermost stratosphere (Ex-UTLS) is characterised by thermal gradients and dynamical barriers which inhibit mixing, give rise to specific trace gas distributions and lead to a variety of definitions of the tropopause (Gettelman et al., 2011; Ivanova, 2013). The thermal tropopause according to WMO criteria (WMO, 1957) is defined as the level, where the lapse-rate decreases to 2 K km$^{-1}$ or less and remains so small at least in the overlying layer of 2 km. This definition identifies the vertical change in the static stability and allows for the existence of multiple tropopause layers. The dynamical tropopause is based on the potential vorticity (PV) and includes both changes in static stability and vorticity (i.e., horizontal and vertical wind shear), also viewed as the dynamic stability. The PV values in the stratosphere exceed its values in the troposphere by an order of magnitude. The threshold value of 2 PVU ($1 \text{ PVU} = 10^{-6} \text{ m}^2 \text{ Ks}^{-1} \text{ kg}^{-1}$) for separating tropospheric and stratospheric air masses is commonly used in studies on stratosphere–troposphere transport. The chemical tropopause is based on the chemical change at the tropopause, identified from tracer-tracer correlations (Zahn and Brenninkmeijer, 2003), with a threshold value of $O_3$ VMR = 120 ppbv being used to distinguish stratospheric from tropospheric air (Thouret et al., 2006). The coexistence of different definitions of the tropopause and the observation that characteristics of air masses around the tropopause depend on the applied definition motivated the concept of the extratropical transition layer (ExTL) which describes the extratropical layer around the tropopause; see Gettelman et al. (2011) and references therein.

The vertical distribution of trace species in the Ex-UTLS is controlled by the strong static stability gradients and dynamic barriers to transport in this atmospheric layer. In the case of water vapour, the $H_2O$ VMR is also determined by the coldest temperature the air parcel has experienced on its way to the tropopause (Lagrangian dry/cold point), which decouples the abundance of water vapour from local cross-tropopause mixing to some extent (Hoor et al., 2010; Zahn et al., 2014). The distribution is described by a steep decrease of the $H_2O$ VMR up to the tropopause layer. Across the tropopause layer, $H_2O$ VMR decreases further but less steep until it reaches its near-constant stratospheric value at about 2 km altitude above. The thermal tropopause forms thus an efficient barrier for the large-scale vertical transport of $H_2O$ into the stratosphere, whereas troposphere-

stratosphere transport occurs for specific local-scale dynamic situations such as, e.g., tropopause folds (Hoor et al., 2004; Hoor et al., 2010; Gettelman et al., 2011).

These features are reported from extensive research campaigns like SPURT (Hoor et al., 2004) which was designed on a climatological approach and compared to climatological data from the research programme MOZAIC (Marenco et al., 1998), and from long-term sampling by the CARIBIC passenger aircraft which carries

an instrumented airfreight container (Dyroff et al., 2014; Zahn et al., 2014), or by instrumented balloons (Kunz et al., 2013)..

Of particular interest with respect to ice cloud formation and life cycle is the thermodynamic state parameter $RH_{ice}$ which controls the properties of ice clouds by setting the thermodynamic conditions for cirrus cloud formation, existence and dissolution (Pruppacher and Klett, 1997). Air masses supersaturated with respect to ice

($RH_{ice} > 100\%$), so called ice-supersaturated regions (ISSR), have mostly faced a decrease in temperature  or increase in water vapour mixing ratio, i.e. specific humidity during their past lifetime (Spichtinger and Leschner, 2016). As a result, these air parcels are generally both colder and of higher absolute humidity than the embedded sub-saturated atmosphere (Gierens et al., 1999; Spichtinger et al., 2003b) which did not experience similar changes in their atmospheric state parameters.

In the northern mid-latitudes, ISSR occurrence coincides strongly with the storm tracks over the North Atlantic (Spichtinger et al., 2003b; Gettelman et al., 2006; Lamquin et al., 2012), on the anticyclonic side of the polar jet stream (Diao et al., 2015), and inside the anvil cirrus clouds (D'Alessandro et al., 2017). Frequently occurring synoptic weather features such as fronts or warm conveyor belts lead to synoptic-scale upward motion and thus facilitate the formation of ISSR (Spichtinger et al., 2005). However, ice-supersaturation occurs as well in regions

of high pressure and anticyclonic flow (Gierens and Brinkop, 2012). Detailed studies of the ISSR life cycle by means of Lagrangian trajectory analyses (Irvine et al., 2014) indicate that the lifetime of an air parcel in the state of supersaturation below the tropopause is generally short with the median duration being less than 6 hours for both winter and summer conditions. In an Eulerian view, however, these ISSR regions as composed of many supersaturated air parcels may persist on a much longer time scale (Spichtinger et al., 2005).

In contrast to the strong negative gradient in $H_2O$ VMR at altitudes below but close to the thermal tropopause, ISSR occur frequently in the humid and cold upper tropospheric air masses. Detailed investigations of the distribution and structure of ice-supersaturation in the northern mid-latitude tropopause region over Lindenberg, Germany, from 15 months of balloon soundings showed that ice saturation occurs in most cases below the thermal tropopause, even in meteorological situations where the tropopause pressure is relatively high

(Spichtinger et al., 2003a). On the other hand, the occurrence of ISSR above the thermal tropopause is very rare with a fraction of approx. 6% of the observations over Lindenberg, reporting ice-supersaturation above the thermal tropopause. Direct evidence of the occurrence of ice-supersaturation above but close to the thermal tropopause report a fraction of 2% from an earlier analysis of MOZAIC data (Gierens et al., 1999). Furthermore, research aircraft observations over North America showed that most of the clear-sky ISSRs are located within

+/- 500 m from the thermal tropopause (Diao et al., 2015).

ISSR constitute potential formation regions for ice clouds, persistent contrails and contrail-cirrus. In these cold and humid air masses, natural cirrus clouds may form by heterogeneous or homogeneous freezing processes (Koop et al., 2000; Hoose and Möhler, 2012; Heymsfield et al., 2017), and long-lived contrails and contrail-

cirrus are generated by cruising aircraft, causing the major non-$CO_2$ climate impact of civil aviation (Aaltonen et

al., 2006; Stuber et al., 2006; Burkhardt et al., 2008; Lee et al., 2010; Burkhardt and Kärcher, 2011; Kärcher, 2018; Bock and Burkhardt, 2019).

The occurrence of ISSR and its close link to the occurrence of cirrus clouds is reported from a joint analysis of SAGE II data on subvisible cirrus and MOZAIC ice-supersaturation by Gierens et al. (2000) which provides an almost 1:1 relationship between subvisible cirrus occurrence and ice-supersaturation, but without discrimination between tropospheric and stratospheric air masses. From other platforms, there are only very few reports of cirrus clouds above the tropopause layer, either from satellite retrievals (Spang et al., 2015) or from research aircraft flights (Müller et al., 2015).

Despite the high climate-related relevance of the vertical distribution of water vapour VMR and related $RH_{ice}$ in the vicinity of the extratropical tropopause layer, there exist only very few approaches for the continuous global-scale monitoring of water vapour abundance and $RH_{ice}$ with sufficient precision and vertical resolution; see Müller et al. (2016) for an overview. Among space-borne techniques, the High-Resolution Infrared Radiation Sounder (HIRS) instruments are most important since they cover more than 3 decades of observations (Gierens et al., 2014), whereas the Microwave Limb Sounder (MLS) and the Atmospheric InfraRed Sounder (AIRS) were particularly used for the space-borne global mapping of ISSR (Spichtinger et al., 2003b; Lamquin et al., 2012) and cirrus cloud coverage (Stubenrauch et al., 2010). However, the vertical resolution provided by space-borne instruments in the Ex-UTLS is very limited and does not allow detailed studies on the vertical distribution of $RH_{ice}$ in this region. In addition, satellite observations such as NASA AIRS data contain biases in temperature and water vapour retrievals compared with aircraft observations by 1 - 2 Kelvin and 30% - 40% of $H_2O$ VRM, respectively (Diao et al., 2013).

Concerning in-situ observations of water vapour, the international network of weather balloons is in operation for many decades but the observations are considered insufficient for detecting trends and variability in UTLS water vapour; see Müller et al. (2016) and references therein. The GCOS Reference Upper-Air Network (GRUAN) targets the provision of climate-quality measurements of tropospheric and lower stratospheric variables (Seidel et al., 2009). GRUAN has established rigorous data quality assessment measures to provide reference-quality in situ and ground-based remote sensing observations of upper-air essential climate variables and serves as another source of high-quality water vapour data, however, for a limited number of certified surface stations yet (Bodeker et al., 2016).

The only other existing global-scale in-situ observation infrastructure for atmospheric composition in the Ex-UTLS uses instrumented passenger aircraft for routine measurements of trace gases like $H_2O$, $O_3$, CO, greenhouse gases and nitrogen oxides, aerosols and clouds at cruise altitude. IAGOS (In-service Aircraft for a Global Observing System; see Petzold et al. (2015), Nédélec et al. (2015), and www.iagos.org for details) and its predecessor research programs MOZAIC (Marenco et al., 1998) and CARIBIC (Brenninkmeijer et al., 1999; Brenninkmeijer et al., 2007) conduct regular measurements of water vapour and relative humidity since 1994. The transformation of both former research projects MOZAIC and CARIBIC into the current IAGOS Research Infrastructure took place in 2011. These regular flights on a global scale are unique in its quantity, continuity, and quality of measurements of Ex-UTLS air masses and have provided detailed insights into the distribution of $RH_{ice}$ (Gierens et al., 1999; Spichtinger et al., 2002), the distribution and properties of ISSR (Gierens and Spichtinger, 2000; Spichtinger and Leschner, 2016), their link to cirrus clouds (Gierens et al., 2000; Petzold et al., 2017), and the processes controlling the water vapour distribution (Zahn et al., 2014).

In the present study, we analysed the distribution properties of $RH_{ice}$ and of ISSR in the Ex-UTLS for a latitudinal band reaching from Eastern North America across the North Atlantic to Europe. We used the full MOZAIC period from 1995 to 2010 which permits the robust seasonal analysis for the identified target regions. Our studies focus on the structure of the vertical distribution of $RH_{ice}$, its variability and seasonality, and potential trends. The horizontal resolution of our data set is 1 km, set by the instrument time resolution of 4 s and the cruising speed of approx. 250 m s$^{-1}$. The vertical resolution is set to 30 hPa, which corresponds to a vertical distance of approx. 750 m at cruise altitude (Thouret et al., 2006) and assures sufficient statistical robustness of the conducted analyses. This vertical resolution is of similar order as the typical resolution of UTLS data with a vertical grid spacing of about 50 hPa in the vicinity of the tropopause (Reichler et al., 2003). Chemistry-climate models like EMAC with vertical resolutions L90MA and L47MA use a vertical grid spacing of 15 - 25 hPa near the extratropical tropopause (Jöckel et al., 2016) which is reflected in the selected vertical resolution of MOZAIC data layers.

## 2 MOZAIC RH data set

### 2.1 Data coverage and vertical distribution

The MOZAIC RH data set used for this analysis spans over the period from 1995 to 2010 and is constrained to cruise altitude conditions, i.e., pressure below 350 hPa (above approx. 8 km altitude), and to ambient temperatures below 233 K to exclude potential sensor contamination by supercooled liquid water droplets. The areal boundaries of the analysed data set are 40 °N to 60 °N and cover the regions Eastern North America (105 °W to 65 °W), North Atlantic (65 °W to 5 °W) and Europe (5 °W to 30 °E). Figure 1 illustrates the global coverage of water vapour observations by MOZAIC for the years 1995 to 2010. Inserted boxes mark the regions Eastern North America, North Atlantic and Europe. The annual data coverage for each analysed regional box varies between 30 and 65 flight hours of MOZAIC aircraft per season (3 months) which corresponds to 27,000 to 60,000 data points of 4 s duration each, per season per year. All investigated regions are characterized by continuous data coverage over the investigated period with no data gaps. Data are available to open access through the IAGOS data portal at www.iagos.org.

Since MOZAIC data are collected at constant-pressure cruise levels of passenger aircraft which may cross from the upper troposphere (UT) through the tropopause layer (TPL) into the lowermost stratosphere (LMS) and back, the data vertical coordinates are reported relative to the tropopause pressure level.

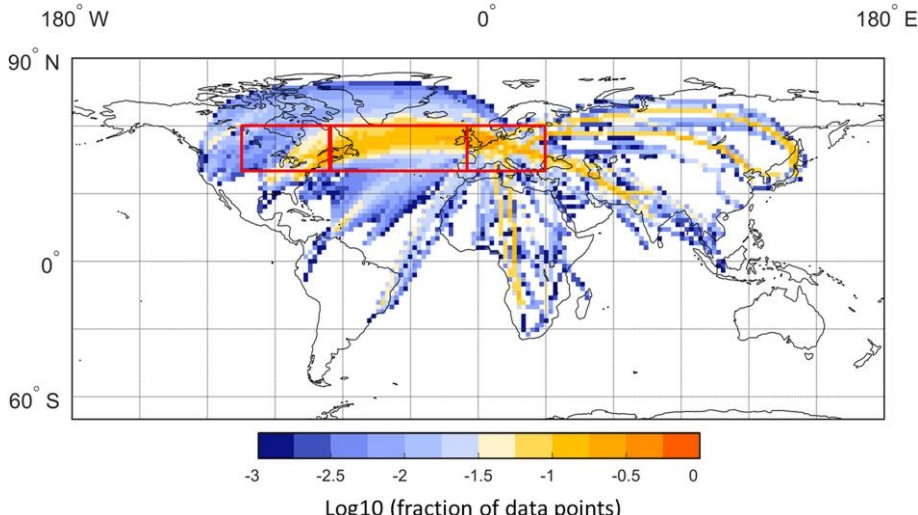

**Figure 1.** Global coverage of water vapour observations by MOZAIC for the period 1995 to 2010, shown as decadal logarithm of the fraction of measurements in a certain grid box; red boxes indicate the target areas for our analyses.

The pressure levels of the thermal tropopause ($p_{therm.TPH}$) and the dynamical 2 PVU tropopause ($p_{dyn.TPH}$) were derived from ERA-Interim (Dee et al., 2011) which uses 60 model layers with the top of the atmosphere located at 0.1 hPa. For our analysis, the 6-hourly outputs from ERA-I (0.75° x 0.75°) were interpolated onto a 1° x1° horizontal grid and on 60 vertical levels of constant pressure and potential temperature (Kunz et al., 2014; Berkes et al., 2017). Additionally, the variables of the PV, and the pressure of the thermal tropopause ($p_{therm.TPH}$) based on the WMO criteria were calculated (WMO, 1957; Reichler et al., 2003). The ERA-Interim data were then linearly interpolated with respect to longitude, latitude, pressure, and time onto each flight track with 4 s resolution, as described by Kunz et al. (2014). Interpolated tropopause pressure levels were finally used to determine the position of the aircraft relative to the thermal tropopause or to the 2 PVU iso-surface, respectively, and thus to distinguish whether the aircraft sampled air masses of UT, TPL or LMS origin with respect to the chosen tropopause definition.

In order to reach both a sufficiently large data set for robust statistical analyses and good vertical resolution, the Ex-UTLS is subdivided into seven layers of 30 hPa thickness each, with three layers located below the thermal tropopause height and three layers above. Thouret et al. (2006) used a similar definition, but referenced to the dynamical tropopause at 2 PVU, i.e. they defined the tropopause as a mixing zone 30 hPa thick across the 2 PVU potential vorticity surface.

The seven layers of 30 hPa thickness each are centred at $p_{therm.TPH} = 0$ hPa for the tropopause layer (TPL) itself and then at $p_{therm.TPH} \pm 30$ hPa, $p_{therm.TPH} \pm 60$ hPa, and finally at $p_{therm.TPH} \pm 90$ hPa. From this vertical spacing, the separation of air masses is achieved by applying the following criteria (formulated for the thermal tropopause only):

LMS : $p < p_{therm.TPH}$ -15hPa; which is limited by the maximum cruise altitude with $p \approx 190$ hPa;

TPL : $p = p_{therm.TPH} \pm 15$hPa;

UT : $p > p_{therm.TPH} + 15$hPa; limited to lower altitudes by $p < 350$hPa.

The bulk of our analyses refer to the classic thermal tropopause according to WMO criteria (WMO, 1957), with the exception of the occurrence of ISSR above the tropopause, where we present the analyses for both tropopause definitions and compare the results to learn more about the processes influencing the formation of ISSR; see Section 3.3.

Since each data set from one single flight provides only a one-dimensional snapshot of the state of the atmosphere along the flight track, and each aircraft cruises at a slightly different pressure level, the entire MOZAIC data are merged to season files of 3-months duration, allowing the analysis of vertical distributions of atmospheric state parameters on a robust statistical basis. For each season file, the statistical distribution (average and standard deviation, median and percentiles) of investigated properties (temperature, $O_3$ VMR, $H_2O$ VMR, $RH_{ice}$, ISSR fraction) is calculated with respect to the above defined UT, TP and LMS vertical layers. From these seasonal averages or percentiles, respective 15-year mean values and standard deviations are determined.

In our study, we use statistical analyses in the following manner: when assessing results from laboratory studies and calibration experiments based on reproducible observations, we apply the 2-σ criterion for the 95% confidence level; when interpreting results from atmospheric observations which are taken from fast-flying airborne platforms and cover 15 years of observations, including their interannual and lateral variabilities, we report the mean values and respective 1-σ standard deviations and state statistical significance or insignificance, respectively.

## 2.2 RH and $O_3$ instrumentation

The relative humidity measurements of MOZAIC and today IAGOS use a thin-film capacitive sensor of type Humicap (Vaisala) which is mounted inside an aeronautic Rosemount inlet attached to the aircraft skin. The MOZAIC Capacitive Hygrometers (MCH) are calibrated in the laboratory against a Lyman α resonance fluorescence hygrometer (Kley and Stone, 1978) with respect to RH over liquid water (Helten et al., 1998; Smit et al., 2014). The conversion to $RH_{ice}$ uses the equations by Sonntag (1994). The MCH reports RH data with an average uncertainty of 4% RH (span 1% RH to 6% RH) in the middle troposphere at 4 to 8 km altitude during ascent and descent, and 5% RH (span 2% RH to 8% RH) at the tropopause and lowermost stratosphere at 10 to 12 km cruising altitude (Smit et al., 2014). The $H_2O$ VMR was finally calculated from the simultaneously measured $RH_{ice}$ and temperature data and from the pressure recordings of the aircraft avionic system.

The deployed sensor has been carefully compared to high-precision water vapour instruments in dedicated research aircraft studies (Helten et al., 1999; Neis et al., 2015a; Neis et al., 2015b) which demonstrate a remarkably good agreement between the MCH and reference instruments with $R^2 = 0.92$ and a slope of m = 1.02 from linear regression analyses. The authors report an MCH uncertainty of 5% RH which is in close agreement with the uncertainty determined from error propagation analysis (Smit et al., 2014).

Kunz et al. (2008) who performed a statistical analysis of water vapour measurements from the SPURT campaigns between 2001 and 2003 by a Lyman-α photo-fragment fluorescence hygrometer (Zöger et al., 1999; Meyer et al., 2015) and MOZAIC water vapour data from the same period determined a limit of detection (LOD) of 10 ppmv for the MOZAIC sensor. Applying the same 2-σ criterion (95% confidence level), we obtain a MCH LOD of $RH_{ice,LOD} = 10\%$ which again transfers into a minimum detectable $H_2O$ VMR of approx. 10 ppmv at typical mid-latitude upper troposphere conditions (T = 218K, p = 250 hPa); see also Neis et al. (2015a) for a detailed discussion. As is discussed by Smit et al. (2014), the uncertainty of the temperature measurement of the

MCH sensor is included in the determination of the MCH $RH_{ice}$ uncertainty so that the precision of $H_2O$ VMR data deduced from MCH $RH_{ice}$ data can be determined directly from the uncertainty of $RH_{ice}$ measurements. Overall, the 5% RH uncertainty leads to a decreasing precision of $H_2O$ VMR deeper in the stratosphere and implies a limited use of the MOZAIC $H_2O$ sensor in the stratosphere dominated by low $RH_{ice}$ and thus an increasingly large uncertainty (Kunz et al., 2008).

The Pt-100 temperature sensor of the MCH is characterised by an overall uncertainty of the ambient air temperature of $\pm$ 0.5 K, which includes the data processing (Berkes et al., 2017). The temperature range encountered during the MOZAIC observations in the Ex-UTLS ranges from 200 K to 245 K at mid-latitudes; see Fig. 9 in Berkes et al. (2017) for details.

Since the launch of MOZAIC, the programme provides also $O_3$ VMR data, in addition to $H_2O$ and $RH_{ice}$ observations. Aboard MOZAIC and now IAGOS aircraft, ozone is measured by means of a UV absorption instrument which is characterised by an instrument noise of $\pm2$ ppbv and an integration time of 4 s (Nédélec et al., 2015). We used the collocated measurement of $O_3$ and $H_2O$ / $RH_{ice}$ for the characterisation of ice-supersaturated air masses with respect to a potential stratospheric influence.

**2.3 RH data processing**

The processing of the MCH data had been subject to a calibration error from year 2000 on. This error in the data analysis caused a bias of data towards higher $RH_{ice}$ values and shifted the peak value of the $RH_{ice}$ probability distribution function (PDF) for in-cloud observations to approx. 130% $RH_{ice}$ which is far above the physically expected value of 100% $RH_{ice}$. Earlier MCH data for the period 1995 to 1999, however, are not affected. The publications by Lamquin et al. (2012) ( Fig. 5 of that publication) and Penner et al. (2018) (Fig. 6 of that publication) illustrate the shift of the erroneous MOZAIC data towards higher $RH_{ice}$ values very clearly.

The calibration error was corrected in a recent reanalysis and the PDFs of $RH_{ice}$ are now consistent for the full MOZAIC period and physically reasonable with the PDF showing a second maximum at 100% $RH_{ice,}$ as expected for in-cloud segments (Smit et al., 2014).

Besides the calibration error, another limitation of the MOZAIC RH data set stemming from MCH sensor drifts, required correction. In its standard operation mode, MCH sensors were replaced every 3 to 6 months. During their deployment periods, the sensors showed occasionally drifts of the sensor output signal caused by a shift of the sensor offset voltage, which results in erroneous $RH_{ice}$ values. To overcome this measurement artefact, the so-called in-flight calibration method (IFC) was developed by Smit et al. (2008), which references the offset voltage of the sensor to signals from flight segments in dry stratospheric air masses where the expected $RH_{ice}$ signal is below the MCH LOD and thus the true MCH signal is considered zero RH.

The method is illustrated in Figure 2: The MCH sensors leave the calibration facility with a baseline for dry conditions (green curve); the theoretical signal expected from the stratospheric $H_2O$ background of 5 ppmv is then added and this new baseline (blue curve) is the reference line for the offset determination. In the operational mode of the IFC method, the lower bound values of the MCH signal during an operational period of typically 15 consecutive flights are determined as the observations below the 1 Percentile value (P01) of the data collected

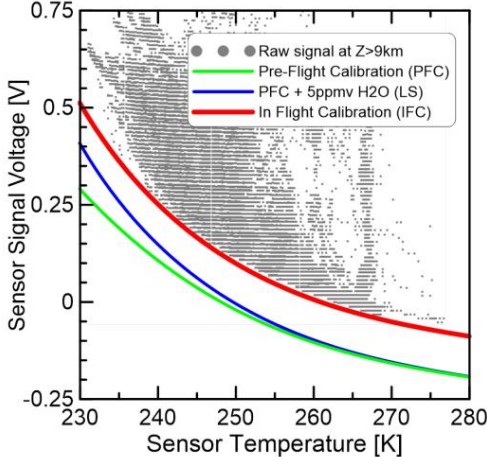

**Figure 2.** Raw signal of the MOZAIC humidity sensor aboard one MOZAIC aircraft as a function of the sensor temperature inside the aeronautic housing obtained at cruise altitude (z = 9 - 12 km). Green line: zero signal from pre-flight calibration (PFC); blue line: superposition of zero signal from PFC and contribution by 5 ppmv water vapour; red line: zero signal from In-Flight Calibration (IFC).

during the respective flight sequence. In case of a sensor offset drift during MCH operation, the lower envelope from the P01 values is similar to the baseline for dry conditions at calibration plus the 5 ppmv stratospheric $H_2O$ background value, but shifted by a voltage offset. The difference between the lower envelope and the baseline from calibration determines the sensor offset voltage which is then subtracted from the raw signal. Details of the method are described by Smit et al. (2008).

## 2.4 RH data validation

The IFC method was applied to the full reanalysis data set from 1995 to 2010. Figure 3a illustrates the effect of the IFC method for the averaged $RH_{ice}$ PDF for the entire MOZAIC data set, irrespective of the geographical regions where the data were collected. The presented average PDF and variability is calculated from annual PDFs.

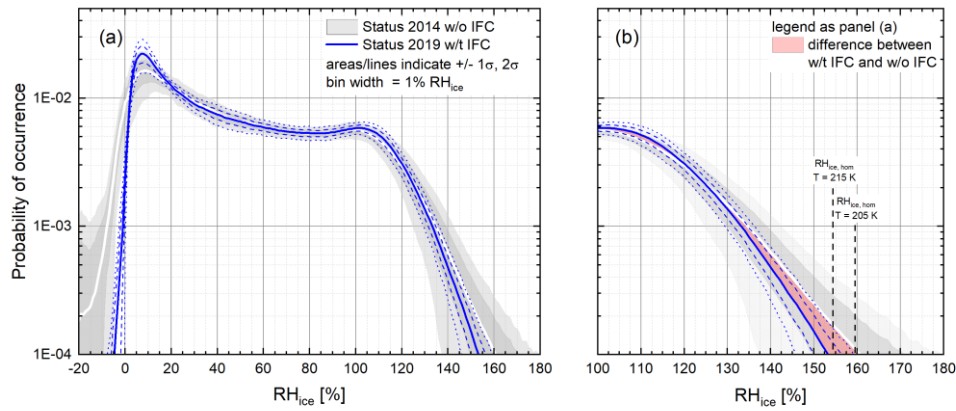

**Figure 3.** Averaged probability density functions of $RH_{ice}$ for the entire MOZAIC period from 1995 to 2010 (a), and the zoom into the region of ice-supersaturation (b); data stem from the reanalysis (Smit et al., 2014) without (w/o: white line, grey areas), and with (w/t: blue lines) the in-flight calibration method applied to the data; the red-shaded area indicates the difference between IFC applied (w/t IFC) and not applied (w/o IFC), and vertical

lines indicate the threshold $RH_{ice}$ values for homogeneous nucleation of ice at T = 205 K and T = 215 K (Koop et al., 2000).

Solid lines refer to the MOZAIC average PDF without the IFC method (white) and with the IFC method applied (blue). Grey areas (without IFC) and dashed and dotted blue lines (with IFC applied) represent the $\pm 1\sigma$ and $\pm 2\sigma$

ranges. Figure 3b shows a zoom into the PDF for the range with $RH_{ice}$ > 100%. In addition to Panel (a), the red area marks the difference between the averaged PDFs without and with IFC applied.

The overall features of the $RH_{ice}$ PDF with an overall maximum value at dry stratospheric air mass values with $RH_{ice}$ being close to the LOD of approx. 10%, and a second local maximum at $RH_{ice} \approx 100\%$ for observations inside cirrus clouds remain unaffected, whereas the deviation between the average PDFs becomes relevant for

$RH_{ice}$ values above 130%. Here, the IFC leads to an average reduction of < 5% $RH_{ice}$ for an occurrence probability of $10^{-3}$ and approx. 7.5% $RH_{ice}$ for an occurrence probability of $10^{-4}$. More relevant, the $2\sigma$ - variability of the observed ice-supersaturations at $10^{-4}$ occurrence probability reduces from max. 180% $RH_{ice}$ (without IFC applied) to 155% $RH_{ice}$ (with IFC applied). The latter value with the IFC applied fits into the range of the homogeneous freezing thresholds at typical extratropical tropopause conditions of $RH_{ice,hom}$ = 158.25% at

205 K to $RH_{ice,hom}$ = 154.15% at 215 K (Koop et al., 2000), as sampled by MOZAIC. Respective values without the IFC applied, however, exceed the homogeneous nucleation threshold significantly. Unphysical negative values of $RH_{ice}$ connected to observations below the LOD of 10% $RH_{ice}$ vanish within the range of uncertainty when applying the IFC method.

Figure 4 illustrates the distribution of $RH_{ice}$ observations from the entire MOZAIC data set shown in Figure 1 as

a function of ambient temperature, colour-coded by the probability of occurrence, i.e. the fraction of data points for a specific combination of temperature and $RH_{ice}$ with the respect to the entire ensemble. About 98% of $RH_{ice}$ observations remain inside the physical boundaries set by the water saturation line and the line for homogeneous ice nucleation. The remaining 2% are considered outliers associated to aircraft manoeuvres. Overall, ice-supersaturated air masses are characterised by $H_2O$ VWR $\geq$ 25 ppmv (1-percentile value), which is in good

agreement with observations of $H_2O$ VWR > 15 ppmv (Krämer et al., 2020) and 20 ppmv (Diao et al., 2014), both reported from research aircraft observations at mid-latitudes for T > 200 K. See also Section 3.3 for more details.

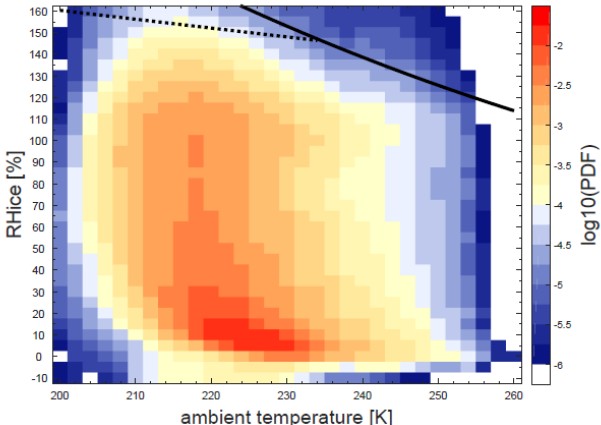

**Figure 4.** Distribution of $RH_{ice}$ for the entire MOZAIC period from 1995 to 2010 with IFC applied as a function of ambient temperature with the colour indicating the probability of occurrence; the lines represent water

saturation (solid line; Sonntag, 1994) and the threshold $RH_{ice}$ for homogeneous ice nucleation (dotted line; Koop et al., 2000; Kärcher and Lohmann, 2002).

Besides the validation of MOZAIC $RH_{ice}$ distributions with respect to the homogeneous nucleation thresholds from (Koop et al., 2000), the data were compared to the distribution of $RH_{ice}$ from observations on board of research aircraft by high-precision water vapour instruments such as Lyman-α photo-fragment fluorescence hygrometers (Zöger et al., 1999; Sitnikov et al., 2007), tunable diode laser absorption spectrometers (May and Webster, 1993; Krämer et al., 2009; Buchholz et al., 2013), and frost point hygrometers; see Meyer et al. (2015)
for details. In total, 250 research flights from 32 field campaigns conducted between 1999 and 2017 globally were analysed. To ensure comparability to the MOZAIC data set, the temperature range was restricted to 205 K to 235 K which corresponds to the MOZAIC observation range, with the upper temperature limit set by the homogeneous freezing threshold.

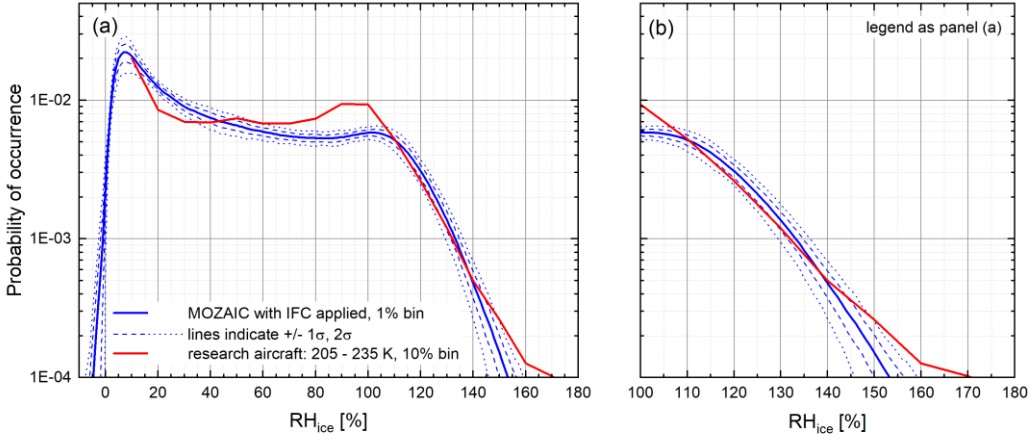

**Figure 5.** Averaged probability density functions of $RH_{ice}$ for the entire MOZAIC period from 1995 to 2010; with the in-flight calibration method applied (blue lines) and respective $RH_{ice}$ PDF from 250 research aircraft flights collected in the Juelich In-situ Airborne Database (Krämer et al., 2016).

The result of this comparison is shown in Figure 5. The MOZAIC $RH_{ice}$ PDF is plotted similar to Figure 3,
whereas the $RH_{ice}$ PDF from the research aircraft campaigns is shown as red line, calculated for $RH_{ice}$ bin widths of 10%. Both probability distribution functions show excellent agreement within the uncertainty ranges, particularly for the regime of ice-supersaturation (panel b). The differences for $RH_{ice}$ near 100% are caused by the preferred sampling of ice clouds during the field campaigns (higher probability of ice clouds at $RH_{ice} \approx$ 100%) and by frequent sampling of contrails at subsaturated conditions ($RH_{ice} < 100\%$).
$RH_{ice}$ observations from the CARIBIC passenger aircraft exhibit similar features as the observations shown here from MOZAIC and from research aircraft, with maximum probability of occurrence at $RH_{ice} = 100\%$ and maximum $RH_{ice}$ values of approx. 150% (Dyroff et al., 2014). In that respect, all observation platforms provide consistent information on the distribution of ice-supersaturation in the extratropical tropopause.

With the IFC method applied to the full MOZAIC $RH_{ice}$ data, this data set is successfully validated against $RH_{ice}$
observations by high-precision instruments and against physically justified bounding values. In summary, this data set is now considered of highest possible quality achievable by the type of sensor applied and for the type of routine observations performed.

## 3 Results

### 3.1 Annual cycles of water vapour and $RH_{ice}$ distributions at the tropopause

The annual cycles of the vertical distributions of water vapour volume mixing ratio ($H_2O$ VMR) and $RH_{ice}$ were analysed for the three target regions Eastern North America (ENA), North Atlantic (NAtl) and Europe (EU), based on 15-year averages of monthly mean profiles relative to the thermal tropopause. For all investigated regions, the annual cycles of $H_2O$ VMR vertical distributions are shown in Figure 6. For the lowest layer of the lowermost stratosphere, bounded from below by the thermal tropopause layer, the patterns are similar for the

three regions, characterised by low $H_2O$ VMR values in winter and spring months and a maximum $H_2O$ VMR during summer. For all regions, the influence of upper tropospheric air masses reaches approx. 1.0 - 2.0 km above the tropopause, with strongest influence in summer.

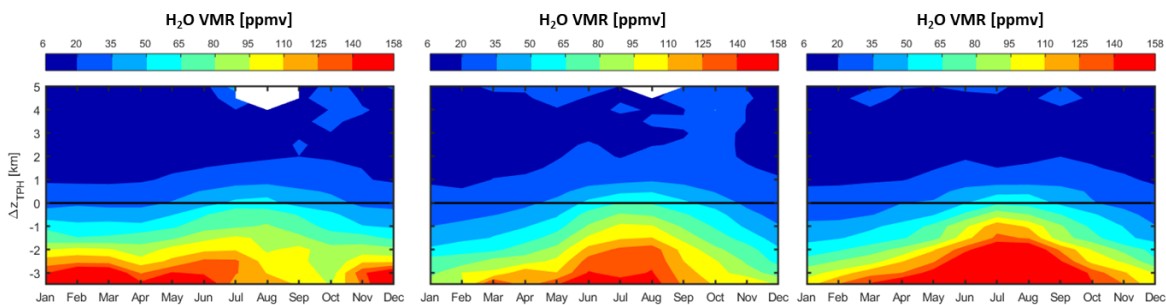

**Figure 6.** 15-year averaged annual cycles of $H_2O$ VMR vertical distributions of $H_2O$ VMR for latitudes 40 °N to

60 °N and for the regions (from left to right) Eastern North America (105 °W to 65 °W), North Atlantic (65 °W to 5 °W) and Europe (5 °W to 30 °E).

Below the tropopause layer, however, we find different behaviour for the studied regions. It appears that over the North Atlantic and over Europe which is strongly influenced by the North Atlantic synoptic weather systems due

to the prevailing westerly winds, the annual cycles of $H_2O$ VMR in the uppermost troposphere and tropopause layers are coupled, while for the Eastern North American region the upper free troposphere layers seem to exhibit higher specific humidity in winter than respective air masses over the ocean. At the tropopause level however, the differences vanish and the annual cycles converge.

A similar behaviour of the annual cycle of $H_2O$ VMR was reported by Zahn et al. (2014) from zonal-averaged

$H_2O$ VMR observations by the CARIBIC system. In contrast to MOZAIC, the CARIBIC $H_2O$ sensor provides good data also for the lower stratosphere where the MOZAIC RH sensor loses its sensitivity, but due to its limited regional coverage, the CARIBIC data set cannot provide regional-scale resolution. In that respect, these data sets complement each other with CARIBIC observations backing up the MOZAIC $H_2O$ VMR reported for the atmospheric layers just above the thermal tropopause and MOZAIC providing regional-scale resolution of

seasonal patterns which is not possible otherwise.

Potential transport pathways of water vapour into the lowermost stratosphere are not in the scope of this study, and cannot be deduced from the analysis shown in Figure 6, but are discussed in depth elsewhere; see e.g., Gettelman et al. (2011), Zahn et al. (2014) and references given therein. In summary, the seasonal variation of $H_2O$ in the first 1-2 km above the tropopause is controlled by shallow, fast, two-way cross-tropopause mixing

which is active around the year and is responsible for the extratropical tropopause mixing layer, or ExTL,

respectively (Hoor et al., 2004), localized deep convection events which occur mainly in the summer period over continents (Anderson et al., 2012; Schwartz et al., 2013), and the hemisphere-scale effect of the Asian summer monsoon (Santee et al., 2017; Rolf et al., 2018). Strong cases of the deep convection events have been reported particularly for the Central United States with unusually wet conditions in the lowermost stratosphere being associated to these events (Anderson et al., 2017). Our long-term data do not point at a significantly higher humidity over the Eastern North America region in summer compared to the North Atlantic and to Europe. However, it has to be noted that our observations are within the northern half of the continental USA and the southern half of Canada (see Figure 1 for the areal coverage of MOZAIC observations), whereas the deep convection events with strong overshooting are reported for regions further south over the Great Plains. This regional difference may explain the differing observations.

### 3.2 Annual cycles of $RH_{ice}$ and ISSR distributions at the tropopause

Our study is focusing on the vertical distribution, seasonality and regional variability of $RH_{ice}$ and ice-supersaturated regions in particular which are linked to the water vapour content of the investigated atmospheric layers. Therefore, we discussed the observed water vapour distribution patterns in the preceding section. To shift the focus on $RH_{ice}$, Figure 7 represents a similar analysis as shown in Figure 6, but for relative humidity with respect to ice. In contrast to the differing annual cycles of water vapour distributions at the tropopause as discussed above, we find similar patterns for $RH_{ice}$ over all target regions, with a tropopause layer characterised by mean $RH_{ice}$ of 60% almost independent of the season, a very humid layer just below the tropopause with mean $RH_{ice}$ reaching 80% and weak seasonality, and a stronger seasonality of $RH_{ice}$ at approx. 1 km below the tropopause and further down into the upper free troposphere with dryer air during the summer season and very humid conditions particularly during winter and spring. Similar average values of $RH_{ice}$ of 60 –70% for the uppermost troposphere without significant seasonality are reported from CARIBIC observations (Dyroff et al., 2014; Zahn et al., 2014).

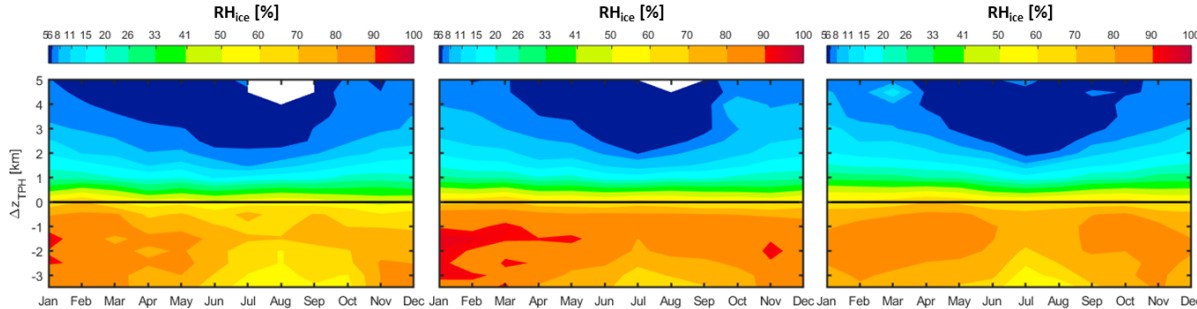

**Figure 7.** 15-year averaged annual cycles of $RH_{ice}$ for latitudes 40 °N to 60 °N and for the regions (from left to right) Eastern North America (105 °W to 65 °W), North Atlantic (65 °W to 5 °W) and Europe (5 °W to 30 °E).

Grouping the data set shown in Figure 8 into seasonal clusters of layers of 30 hPa thickness around the tropopause allows the robust statistical analysis of the vertical distributions of temperature, $H_2O$ VMR, average $RH_{ice}$ and fraction of ice-supersaturated regions. The applied concept of the vertical spacing is described in Section 2.1. The seasonal variation of the vertical distributions of the selected properties is compiled in Figure 8.

Table 1 and Table 2 present the mean fractions (Table 1) and associated standard deviations normalised to the respective mean values (Table 2) for ISSR occurrence, separated for regions and seasons, and in the last set of columns averaged over all regions. As is already indicated in Figure 7, the variation of $RH_{ice}$ with altitude and season is similar for the three target regions.

For all regions, the highest $RH_{ice}$ values and also the highest fraction of ISSR occurrence is observed for the two upper tropospheric layers closest to the tropopause layer whereas for the third layer situated deepest inside the UT, $RH_{ice}$ values and ISSR fractions are considerably lower. Only in the spring season (MAM) over the North Atlantic, the lowest third layer reaches similar values for $RH_{ice}$ values and ISSR fractions as the two layers above. Focusing on the UT layers, the relative standard deviations of the ISSR fractions are highest for the lowest layer investigated here, at least for winter and spring seasons for which the largest ISSR fractions are found.

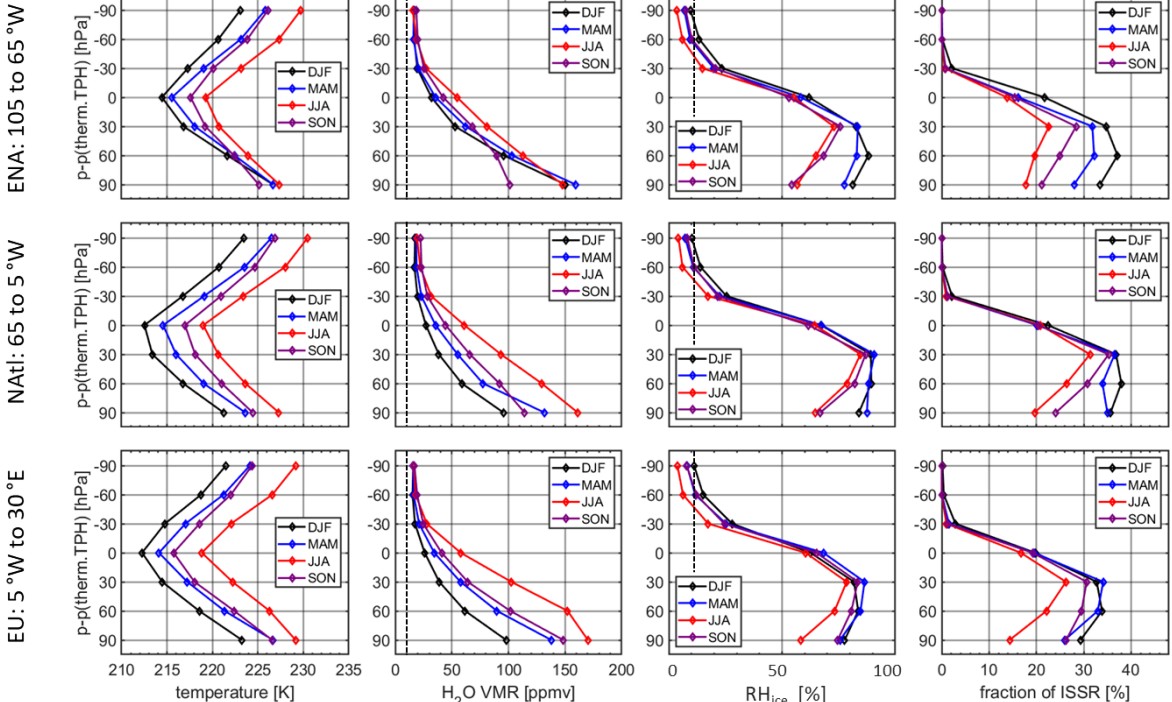

**Figure 8.** Vertical distribution of mean temperature, $H_2O$ mixing ratio, $RH_{ice}$ and fraction of ice-supersaturated regions (ISSR) for seven pressure layers around the thermal tropopause; layer thickness is 30 hPa and layers are spaced equally relative to the tropopause pressure level; dotted lines indicate the MCH 2-$\sigma$ limit of detection of $RH_{ice,LOD} = 12\%$ and the resulting minimum-detectable $H_2O$ VMR of approx.10 ppmv.

**Table 1.** ISSR frequency of occurrence: seasonal mean values are reported in %; the vertical distance to the thermal tropopause is reported as $\Delta p = p_{layer} - p_{therm.TPH}$.

| $\Delta p$ | DJF | | | MAM | | | JJA | | | SON | | | AVG(ENA, NAtl, EU) | | | |
|---|---|---|---|---|---|---|---|---|---|---|---|---|---|---|---|---|
| (hPa) | ENA | NAtl | EU | ENA | NAtl | EU | ENA | NAtl | EU | ENA | NAtl | EU | DJF | MAM | JJA | SON |
| -30 | 2.1 | 2.2 | 2.9 | 0.7 | 1.2 | 1.5 | 0.6 | 0.9 | 1.0 | 0.8 | 1.2 | 1.3 | 2.4 | 1.1 | 0.8 | 1.1 |
| 0 | 21.7 | 22.4 | 19.8 | 16.2 | 20.1 | 19.5 | 13.8 | 20.8 | 16.7 | 15.5 | 20.4 | 19.2 | 21.3 | 18.6 | 17.1 | 18.4 |
| 30 | 34.7 | 36.8 | 32.8 | 31.8 | 36.5 | 34.1 | 22.7 | 31.3 | 26.3 | 28.4 | 35.3 | 30.6 | 34.8 | 34.2 | 26.8 | 31.4 |
| 60 | 37.1 | 37.9 | 33.9 | 32.2 | 34.0 | 33.0 | 19.6 | 26.4 | 22.1 | 25.0 | 30.8 | 29.5 | 36.3 | 33.0 | 22.7 | 28.5 |

| Δp | | | | | | | | | | | | | | | |
|---|---|---|---|---|---|---|---|---|---|---|---|---|---|---|---|
| 90 | 33.5 | 35.7 | 29.3 | 28.0 | 35.0 | 26.0 | 17.7 | 19.6 | 14.3 | 21.1 | 24.1 | 26.3 | 32.8 | 29.6 | 17.2 | 23.8 |

**Table 2.** ISSR frequency of occurrence: normalised standard deviations of seasonal mean values are reported in
%; the vertical distance to the thermal tropopause is reported as $\Delta p = p_{layer} - p_{therm.TPH}$.

| Δp | DJF | | | MAM | | | JJA | | | SON | | | AVG(ENA, NAtl, EU) | | | |
|---|---|---|---|---|---|---|---|---|---|---|---|---|---|---|---|---|
| (hPa) | ENA | NAtl | EU | ENA | NAtl | EU | ENA | NAtl | EU | ENA | NAtl | EU | DJF | MAM | JJA | SON |
| -30 | 102% | 72% | 84% | 78% | 47% | 47% | 99% | 57% | 50% | 78% | 33% | 43% | 86% | 57% | 68% | 51% |
| 0 | 26% | 31% | 31% | 31% | 21% | 22% | 31% | 24% | 19% | 24% | 16% | 23% | 29% | 25% | 25% | 21% |
| 30 | 15% | 20% | 15% | 19% | 18% | 15% | 31% | 25% | 23% | 15% | 10% | 13% | 17% | 18% | 26% | 13% |
| 60 | 19% | 24% | 20% | 25% | 17% | 12% | 31% | 22% | 22% | 17% | 13% | 15% | 21% | 18% | 25% | 15% |
| 90 | 29% | 31% | 25% | 21% | 33% | 15% | 19% | 18% | 31% | 29% | 15% | 17% | 29% | 23% | 23% | 20% |

### 3.3 Physico-chemical signature of ice-supersaturated regions in the vicinity of the tropopause

As discussed in detail by Spichtinger and Leschner (2016) ice-supersaturated air masses have mostly faced decrease in temperature, or increase in water vapour mixing ratio, i.e. specific humidity, during their past
lifetime. Thus, these air parcels are generally known as both colder and of higher absolute humidity than the surrounding sub-saturated air masses (Gierens et al., 1999; Spichtinger et al., 2003b), although research aircraft observations from 87°N to 6°S, showed that 73% of the ISSRs have both lower temperature and higher $H_2O$ VMR than their horizontally adjacent sub-saturated air, while 27% of the ISSRs show higher temperature and higher $H_2O$ VMR than their surroundings (Diao et al., 2014).
. This conclusion is valid for both ISSR in the uppermost troposphere as well as for the rarer cases of ISSR above the tropopause.

In order to study the formation history of ISSR and involved processes, we analysed the occurrence frequency and physico-chemical signature of ISSR around the tropopause layer and referred our analyses to both the thermal and the dynamical tropopause. We want to recall the tropopause definitions given in Section 2.1. The
thermal tropopause according to WMO criteria (WMO, 1957) is usually seen as an effective transport barrier hampering troposphere-stratosphere exchange, whereas the dynamical tropopause is commonly used for separating tropospheric and stratospheric air masses in studies on stratosphere–troposphere transport since it represents the lower bound of the ExTL. These complementary views on the tropopause have been developed from extensive CO - $O_3$ analyses, which showed that the 2 PVU surface approximately separates the troposphere
from the stratosphere with the ExTL as a transition layer of about 2 km thickness above it and centred on the thermal tropopause (Hoor et al., 2004; Pan et al., 2010; Gettelman et al., 2011). These tracer studies in the extratropics showed that on average the dynamical tropopause is situated slightly below the thermal tropopause and the gradients of CO and $O_3$ are much sharper across the thermal tropopause compared to the dynamical tropopause (Hoor et al., 2004; Pan et al., 2010).
Similar features are observed for the gradients of temperature T, $H_2O$ VMR and $O_3$ VMR, shown in Figure 9 for the North Atlantic region. Similar to the tracer gradients, also the temperature gradient is sharper across the thermal tropopause compared to the dynamical tropopause. In addition, the results confirm the good agreement between the ERA-Interim thermal tropopause height indicated by $\Delta p_{TPH} = 0$ hPa, the lowest temperatures detected at $\Delta p_{TPH} = 0$ hPa (panel (a), blue lines), and the chemical tropopause indicated by $O_3$ VMR = 120 ppbv

at $\Delta p_{TPH} = 0$ hPa (panel (c), blue lines), and thus the consistency of the used data set. Furthermore, the analysis of the pressure difference between the thermal and dynamical tropopauses reveal an offset of approx. 25 hPa (15 - 35 hPa) which translates into an altitude difference of approx. 1 km (Neis, 2017).

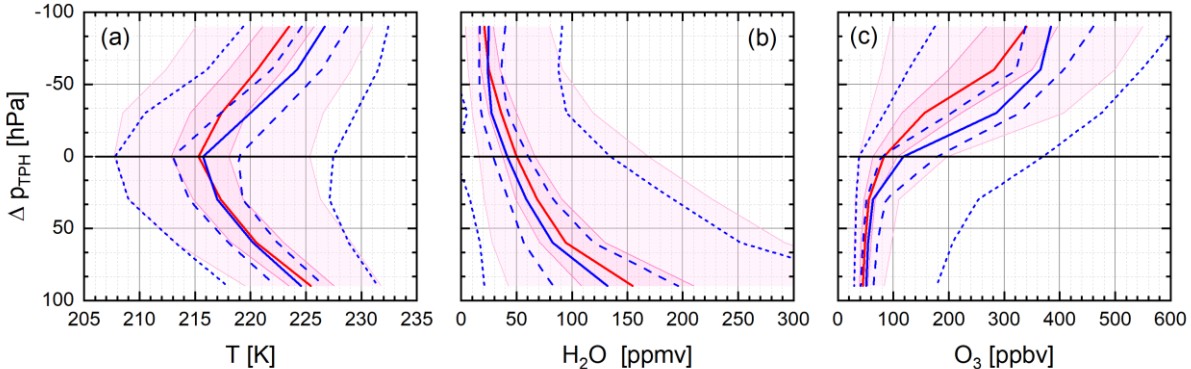

**Figure 9.** Vertical distribution of temperature T (a), $H_2O$ VMR (b), and $O_3$ VMR (c) relative to the 2 PVU
dynamical tropopause and to the thermal tropopause; vertical distributions relative to the thermal tropopause are presented as percentiles [1, 25, 50, 75, and 99] by blue lines and relative to the 2 PVU tropopause conditions by red-shaded areas.

Our analysis of ISSR occurrence in the vicinity of the exTL is confined to the North Atlantic region, for which
we have the highest data density available with respect to vertical resolution. As described generally in Section 2.1, the entire data set of individual $RH_{ice}$ observations over the North Atlantic region was divided into yearly subsets for seasons DJF, MAM, JJA, and SON. For each year, season and altitude layer relative to the thermal and dynamical tropopauses, the average frequency of occurrence of observations with $RH_{ice} > 100\%$ was determined. The probability of ISSR occurrence per altitude layer with respect to the entire period of 15 years
was then calculated from this record of seasonally averaged ISSR frequencies of occurrence. The results are compiled in Table 3 for both tropopause definitions used here. Please note that the ISSR fractions compiled for the thermal tropopause correspond to the values listed in Table 2, but without distinction for seasons.

With reference to the thermal (dynamical) tropopause, the mean ISSR occurrence probability is 31% (38%) in the upper troposphere below the tropopause layer. The observed increases of mean ISSR occurrence probabilities
towards the tropopause layer are, however, below statistical significance and the average values for the respective pressure layers differ for the two tropopause definitions. Our finding that the ISSR occurrence probability is increasing towards the tropopause agrees with results from a previous research aircraft study using the CO - $O_3$ tracer correlation approach, in which the majority (69%) of clear-sky ISSRs was found within the ExTL, while the rest was located below the transition layer (Diao et al., 2015).
Since the thermal tropopause is located at higher altitude than the dynamical tropopause, the pressure layers below the thermal tropopause include part of the ExTL which explains the lower ISSR fractions for UT1–3 below the thermal tropopause, compared to UT1-3 below the dynamical tropopause. Sorting the data according to their vertical distance to the respective tropopause results also in different data ensembles for the respective pressure layers because of the strong horizontal variability of $RH_{ice}$ along the flight trajectories. This strong
horizontal variability explains the different absolute values of ISSR occurrence with respect to the tropopause definitions. For both tropopause definitions, the standard deviation of observed ISSR fractions is largest for the lowest UT layer of the analysed atmospheric region and decreases with increasing altitude.

**Table 3.** Mean and standard deviation of seasonal fraction of ice supersaturated regions (ISSR) for the seven vertical layers distributed around the thermal and dynamical tropopause.

| Layer ID | $p - p_{TPH}$ [hPa] | ISSR fraction [%] | |
| --- | --- | --- | --- |
| | | Dynamical TP | Thermal TP |
| LMS3 | - 90 | 0.2±0.5 | 0.0±0.1 |
| LMS2 | - 60 | 0.7±1.1 | 0.1±0.3 |
| LMS1 | - 30 | 8.4±4.4 | 1.5±1.1 |
| TPL | 0 | 30.7±9.4 | 20.0±6.5 |
| UT1 | 30 | 39.9±10.0 | 33.9±9.0 |
| UT2 | 60 | 37.7±10.7 | 31.4±9.2 |
| UT3 | 90 | 35.5±14.3 | 29.1±12.1 |

When crossing the thermal tropopause, the ISSR fraction drops sharply to values of 1.5% for the lowest layer above the thermal tropopause and to statistically insignificant fractions when reaching further up into the stratosphere. In case of the dynamical tropopause, we find a significantly higher ISSR fraction of 8.4% for the lowest stratosphere layer, and again insignificant fractions further above. This strong contrast in the ISSR occurrence probability for the lowest stratosphere layers with reference to the two tropopause definitions coincides with the behaviour of other tracers in the ExTL; see Figure 9 for details.

In order to learn more about the history of ice-supersaturated air parcels we further analysed the ozone content of the ISSR compared to the sub-saturated air around, for air parcels below and above the thermal and dynamic tropopauses and combined the results with the distributions of temperature and $H_2O$ VMR. The thermodynamic and chemical properties of ISSR and the comparison between ISSR (blue lines) and ice-subsaturated air masses (red-shaded areas and red lines) are presented in Figure 10 with reference to both tropopause definitions. In general, ISSR are colder than their subsaturated counterparts. The difference is low in the UT with 1 - 2 K which compares well to the value of 2 K at 215 hPa obtained from MLS satellite measurements (Spichtinger et al., 2003b), and increases to more than 6 K difference in the stratosphere above the thermal tropopause, and approx. 4 K above the dynamical tropopause. The temperature difference of 3 - 4 K between colder tropospheric ISSR and the surrounding subsaturated air masses reported by Gierens et al. (1999) is comparable to the temperature difference in the 30 hPa thick tropopause layer we find in our analysis.

Figure 10 also indicates a similar behaviour of the vertical distribution of $H_2O$ VMR for ice-supersaturated and ice-subsaturated regions with exponentially decreasing absolute humidity up to the tropopause layer. Above both tropopause layers, $H_2O$ VMR further decreases in case of non-ISSR conditions. For ISSR conditions, however, $H_2O$ VMR remains constant with height throughout the layer just above the tropopause. Doubling of $H_2O$ VMR for tropopause ISSR conditions compared to non-ISSR conditions is close to the results reported from MLS observations (Spichtinger et al., 2003b). In contrast, Gierens et al. (1999) found an increase of only 50% for $H_2O$ VMR inside ISSR compared to non-ISSR. In turn, this value compares well with our observations in the uppermost troposphere.

The vertical distribution of the ozone VMR behaves similar to the temperature for ice-supersaturated and ice-subsaturated regions, with small differences in the ozone VMR of less than 15 ppbv in the troposphere. Already for the tropopause layer and even more pronounced for the first layer above the thermal tropopause, however, the difference increases to 60 ppmv ozone VMR and beyond.

Quantitative conclusions on air mass characteristics and history are drawn from the vertical distributions of thermodynamic and chemical properties shown in Figure 10. The underlying concept of tropospheficity (Cirisan et al., 2013) quantifies the tropospheric nature or fingerprint, respectively, of an air mass on the basis of the observed $O_3$ VMR. In the context of our study, we refer to tropospheficity for consistency with literature.

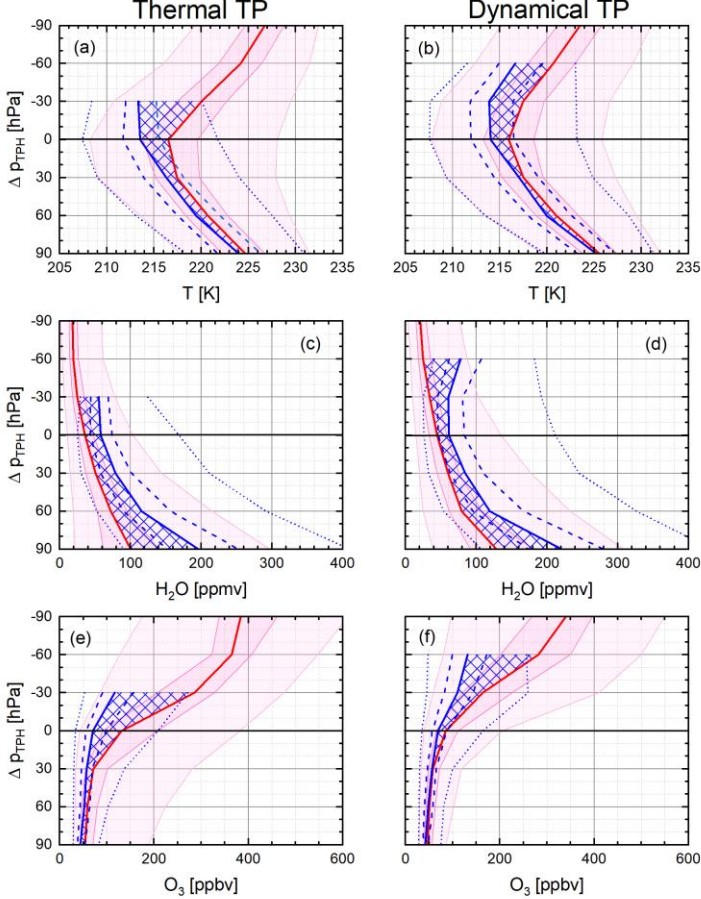

**Figure 10.** Vertical distribution of temperature, $H_2O$ VMR, and ozone VMR for ISSR relative to the thermal tropopause height (panels a, c and e) and 2 PVU dynamical tropopause height (panels b, d, and f). ISSR conditions are presented as percentiles [1, 25, 50, 75, and 99] by blue lines and non-ISSR conditions by red-shaded areas; blue cross-hatched areas highlight the deviation of median values inside ISSR from those non-ISSR conditions.

Using the $O_3$ VMR as a stratospheric air mass tracer and adapting the approach of Cirisan et al. (2013), we define the tropospheficity parameter $m$ for an ensemble of data characterised by median (med) and 99 percentile (P99) values as

$$m = \frac{[O_3]_{P99} - [O_3]_{med}}{[O_3]_{P99} - [O_3]_{tropo}}$$

and apply the median value of the lowest layer analysed here as background tropospheric value, so that $[O_3]_{tropo}$ = 42 ppbv. Petetin et al. (2018) reported a median $O_3$ VMR of 49 ppbv for the Central European mountain station Sonnblick (3106 m above sea level) in the Austrian Alps and a value 50 ppbv for the high Alpine station Jungfraujoch (3580 m above sea level), whereas Cirisan et al. (2013) use a value of 33.5 ppbv from ERA Interim air mass trajectory analyses as the tropospheric background ozone value in the upper troposphere in mid-latitudes.

Applying this definition of the tropospericity parameter $m$ to MOZAIC/IAGOS observations over Central Europe (Petetin et al., 2018) at 4000 m altitude with $[O_3]_{med}$ = 50 ppbv and $[O_3]_{P99}$ = 82 ppbv yields $m$ = 0.80, and for observations at 1500 m altitude with $[O_3]_{med}$ = 42 ppbv and $[O_3]_{P99}$ = 83 ppbv we find $m$ = 1.00.

For MOZAIC/IAGOS observations in the Ex-UTLS Cohen et al. (2018) report, e.g., for springtime lowermost stratosphere conditions values of $[O_3]_{med}$ = 400 ppbv and $[O_3]_{P95}$ = 600 ppbv, resulting in $m$ = 0.36, and for tropopause layer conditions values of $[O_3]_{med}$ = 110 ppbv, $[O_3]_{P95}$ = 200 ppbv, and $m$ = 0.57; note that P95 refers here to the 95 percentile value of the analysed data ensemble, as taken from Cohen et al. (2018). Deeper into the stratosphere beyond the reach of MOZAIC/IAGOS aircraft, the value of $m$ approaches $m$ = 0.0. Thus, similar to the tropospericity parameter defined by Cirisan et al. (2013) from trajectory analyses, a value of $m$ = 0 indicates that an air parcel contains only stratospheric air, while $m$ = 1 is fully tropospheric. Defining the tropospericity as described here, we connect the tropospericity of an air mass to the observed variability of the $O_3$ VMR.

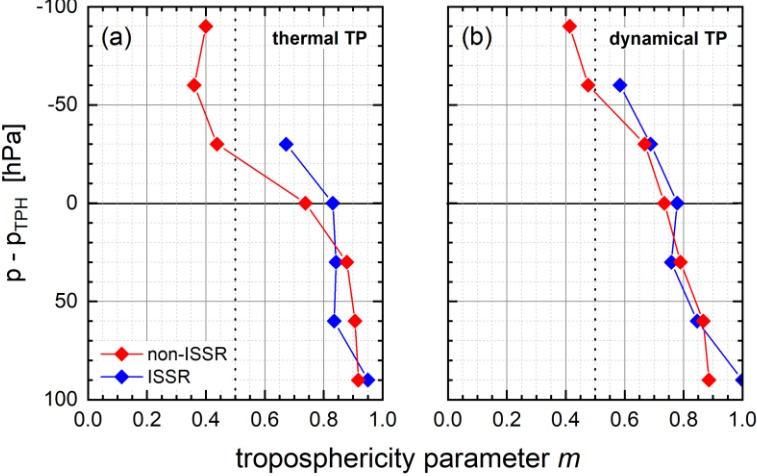

**Figure 11.** Vertical distribution of the tropospericity parameter $m$ for ISSR and non-ISSR air masses with respect to the thermal (a) and dynamical (b) tropopause.

The analysis of tropospericity of the seven investigated layers with respect to the 99 percentile and median $O_3$ VMRs is presented in Figure 11. With respect to the thermal as well as to the dynamical tropopause, the layers up to the tropopause layer are characterised by similar values of $m$ > 0.80 for ISSR and $m$ > 0.75 for non-ISSR air masses. The first layer above the thermal tropopause, however, shows a clear difference between ISSR ($m$ = 0.67) and non-ISSR ($m$ = 0.44) with respect to the thermal tropopause, but similar values of $m$ = 0.67 - 0.69 for ISSR and non-ISSR with respect to the dynamical tropopause.

Recalling the structure of the ExTL with the 2 PVU dynamical tropopause at its lower bound separating the stratosphere from the troposphere, and centred on the thermal tropopause, we find that on top of the ExTL non-ISSR air masses show a clear stratospheric signature, while ISSR air masses are still strongly influenced by

mixing and carry a significant tropospheric fingerprint compared to the non-ISSR air masses. Above the dynamical tropopause and thus inside the ExTL, the influence of mixing increases gradually for both ISSR and non-ISSR air masses and the difference in troposphericiy is much less pronounced than near the top of the ExTL.

### 3.4 ISSR fraction and cirrus cloud occurrence

Ice-supersaturation in the atmosphere is a prerequisite for the formation of cirrus clouds, and the degree of super-saturation, mostly driven by atmospheric dynamics, determines the mechanism by which ice particles form (e.g., Kärcher et al., 2014; Krämer et al., 2016; Heymsfield et al., 2017). On the other hand, $RH_{ice}$ probability distribution functions inside cirrus clouds are characterised by most probable values at or slightly above ice-saturation at $RH_{ice} = 100\%$ (Krämer et al., 2009; Diao et al., 2014; Diao et al., 2015; Petzold et al., 2017) which means that cirrus clouds exist to a considerable fraction also in ice-subsaturated air masses, depending on their state of life. Finally, ice-supersaturation can also occur in cloud-free air masses, but the fraction of ice-supersaturated air in clear sky conditions is largely unknown. However, these cloud-free ISSR are of high importance for the formation of persistent contrails and thus for the climate impact of aviation (Irvine and Shine, 2015; Kärcher, 2018).

Motivated by the high importance of ISSR for cirrus formation and existence and also for the formation and persistence of contrails, we converted the vertically resolved observations of ISSR fractions into an annual cycle of ISSR occurrence for the three target regions. The seasonal-mean occurrence probabilities were analysed for $RH_{ice}$ values of 95%, 100% and 105%, based on the sensor precision of 5% $RH_{ice}$. The resulting annual cycles for the top two UT layers, situated just below the thermal tropopause layer are shown in Figure 12. The range bound by the probabilities of occurrence for $RH_{ice} > 95\%$ and 105% defines the uncertainty of our analysis. Additionally, we analysed the interannual variability of ISSR occurrence from the standard deviation of the mean ISSR occurrence probability for $RH_{ice} > 100\%$. The respective variability range is shown as blue-shaded areas in Figure 12.

For all regions, ISSR occurrence probabilities are highest in winter/spring and lowest in summer, while the absolute values particularly in summer are considerably different. The probability for finding ice-supersaturated air masses during summer is 20% over the Eastern North America regions, but 30% over the North Atlantic, with Europe showing values in the range between.

To the present, there is only very limited in-situ information available about the occurrence probability of ice-supersaturated air masses in the upper troposphere in general and about their seasonality in particular. One source for in-situ information stems from radiosonde observations conducted by the German Weather Service over the observatory Lindenberg in Germany (Spichtinger et al., 2003a).

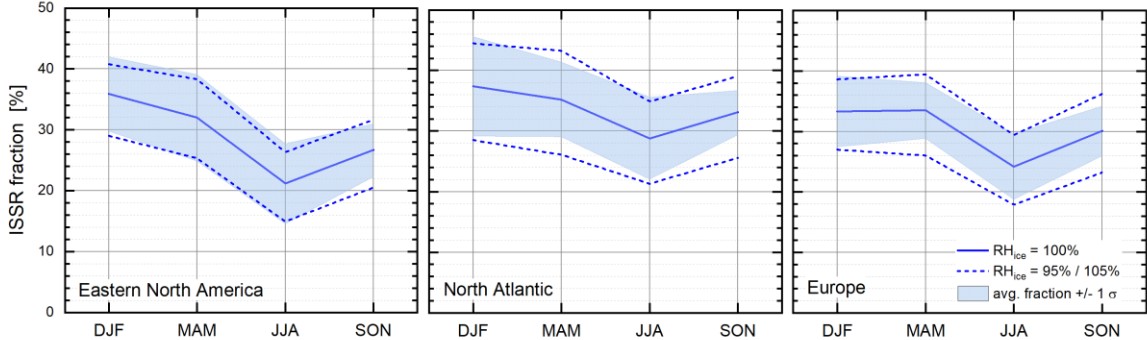

**Figure 12.** Annual cycles of ISSR occurrence shown as occurrence probability for $RH_{ice} > 100\%$, for the regions Eastern North America, North Atlantic and Europe; considered years are 1995 to 2010, with shaded areas representing probabilities for the average value (thick lines) $\pm 1\sigma$, and the short-dashed lines representing average fractions for $RH_{ice} >= 95\%$ and $105\%$, respectively; calculations were conducted for the two UT layers positioned closest to the thermal tropopause.

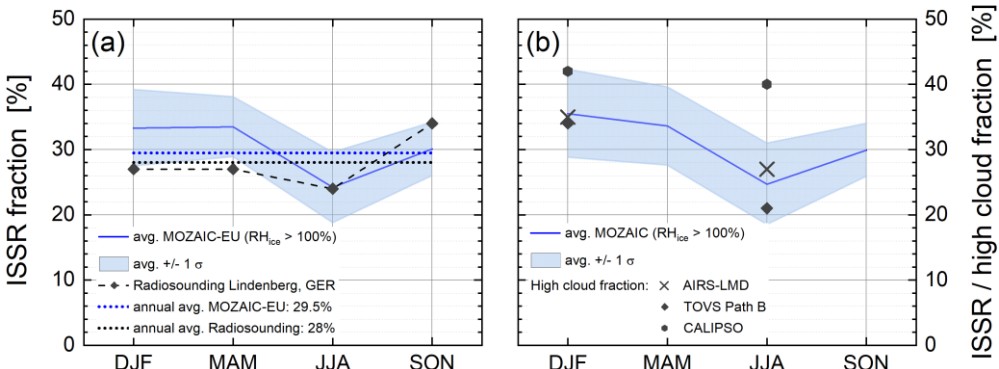

**Figure 13.** (a) Seasonal cycle of ISSR occurrence probability, i.e. $p(RH_{ice} > 100\%)$, averaged over Europe for the years 1995 to 2010 for the two UT layers positioned closest to the thermal tropopause; symbols represent the annual cycle of the Lindenberg sounding (2000 – 2001) from Spichtinger et al. (2003a); (b) Seasonal cycle of ISSR occurrence probability, as $p(RH_{ice} > 100\%)$ averaged over the Northern Mid-Latitudes from East North America to Europe for the period 1995 to 2010; symbols represent high cloud fractions from the satellite cloud climatology by Stubenrauch et al. (2010) for northern mid-latitudes and years 2003 to 2008 for AIRS-LMD, 1987 to 1995 for TOVS Path B and 2006 to 2007 for CALIPSO.

Figure 13a shows the average annual cycles of ISSR occurrence frequency from 15 years of MOZAIC observations over Europe and from 15 months of radiosonde observations over Lindenberg published by Spichtinger et al. (2003a). The 15-months cycle from the radio soundings is covered by the 15 years climatology of ISSR occurrence from MOZAIC, but contributes only a snapshot compared to the 15-years' time series. Based on the 15 months of observation, the authors report a mean frequency of occurrence of ice-supersaturation layers over Lindenberg of 28%, whereas the annual cycle of ISSR occurrence from our 15 years of MOZAIC observations over Europe yields a mean value of 29.5% with a range from 35% ($RH_{ice} > 95\%$) to 23% ($RH_{ice} >= 105\%$).

Another source of data, but for the occurrence frequency of cirrus clouds originates from long-term analyses of satellite observations (Stubenrauch et al., 2010; Stubenrauch et al., 2013). In their 6-year climatology Stubenrauch et al. (2010) report cirrus cloud coverage fractions for northern mid-latitudes of 35% in January and

27% in July from the Atmospheric Infrared Sounder analysed at the Laboratoire de Météorologie Dynamique in Paris (AIRS-LMD:2003 to 2008), and respective fractions of 34% and 21% from the TIROS-N Operational Vertical Sounder (TOVS) Path-B cloud retrieval (TOVS – Path B: 1987 to 1995), and 42% and 40% from the Cloud-Aerosol Lidar and Infrared Pathfinder Satellite Observation (CALIPSO: 2006 to 2007). The compilation of our annual cycle of ISSR occurrence and the respective observations from space-borne sensors is shown in

Figure 13b. The agreement of the observations of ISSR occurrence from the very different sources is remarkably good, with the exception of CALIPSO observations which provide higher values. According to Stubenrauch et al. (2010), the high cloud fraction of CALIPSO is about 10% larger than respective values of CALIPSO for clouds excluding subvisible cirrus. Therefore, the difference between high cloud fractions from CALIPSO and from the other instruments shown in Figure 13 can be attributed to instrument sensitivities.

The good agreement between MOZAIC in-situ observations of ISSR occurrence with the high-cloud fraction from satellite instruments encourages further detailed studies on this matter. First analyses of simultaneous observations of $RH_{ice}$ and $N_{ice}$ which are now possible within the ongoing IAGOS programme already indicate a strong correlation of high $RH_{ice}$ values with its occurrence inside cirrus clouds (Petzold et al., 2017).

**3.5 Trend analysis**

Finally, we analysed the 15-years records of the validated MOZAIC $RH_{ice}$ observations and the resulting fraction of ISSR observations for the three regions Eastern North America, North Atlantic and Europe for potential trends. The bases of our analyses were the seasonally averaged observations in the uppermost tropospheric layer (UT) with respect to the thermal tropopause, and the respective average seasonal cycles depicted in Figure 12. The resulting time series are shown in Figure 14. The seasonality of ISSR occurrence is clearly visible for each

region, but with considerable interannual variability. Similar to Figure 12, the shaded regions represent the average fractions for $95\% < RH_{ice} < 105\%$, respectively, and indicate thus the uncertainty resulting from the instrument precision of $RH_{ice} = 5\%$. For none of the regions, we find significant trends in ISSR occurrence. Therefore, the distribution of $RH_{ice}$ in the uppermost troposphere close to the tropopause layer and the resulting occurrence of ice-supersaturation seem to be stable over the investigated time period from 1995 to 2010.

In order to get a clearer understanding of the reasons for the interannual variability, we further analysed the de-seasonalised time series of the ISSR fractions by calculating the difference between each seasonal value of the ISSR fraction and the 15-years seasonal average (see Figure 12).

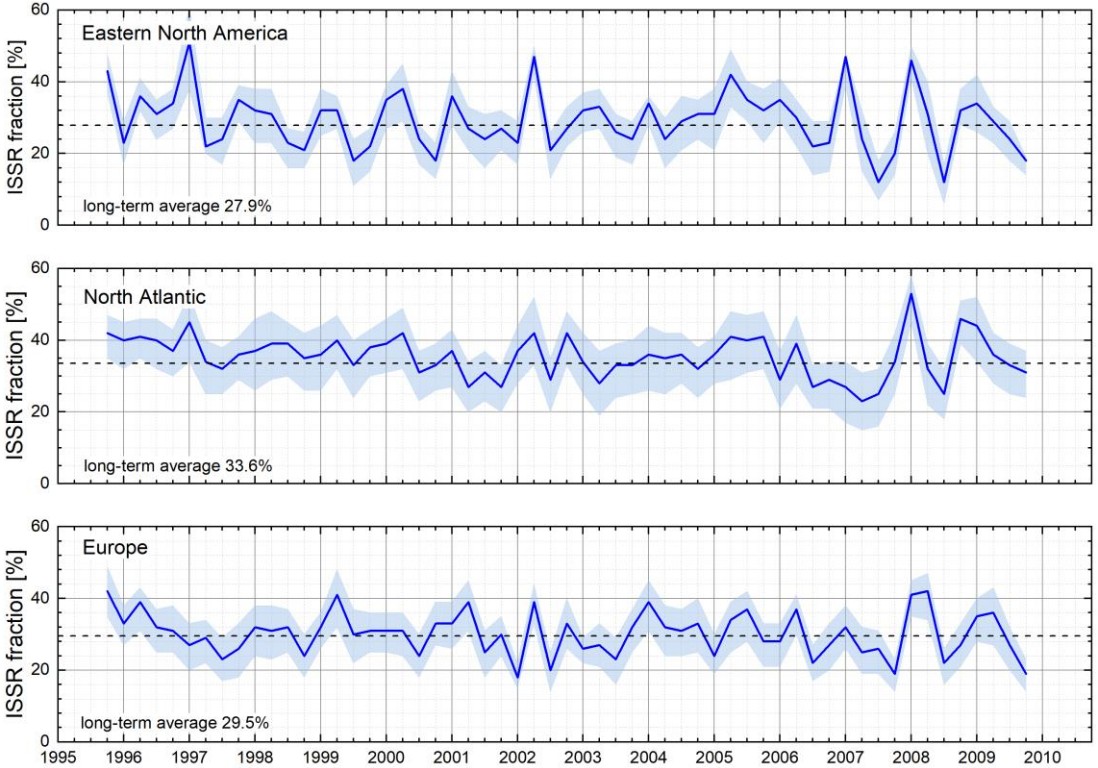

**Figure 14.** Time series of ISSR fraction (probability of occurrence) for latitudes 40°N to 60°N and for the regions (from top to bottom) Eastern North America (105°W to 65°W), North Atlantic (65°W to 5°W) and Europe (5°W to 30°E) for the top UT layer, situated just below the tropopause layer; solid lines represent probabilities for the average value for $RH_{ice} = 100\%$ and the shaded areas represent average fractions for $95\% < RH_{ice} < 105\%$, respectively, long-term average values for $RH_{ice} = 100\%$ are added in the panels.

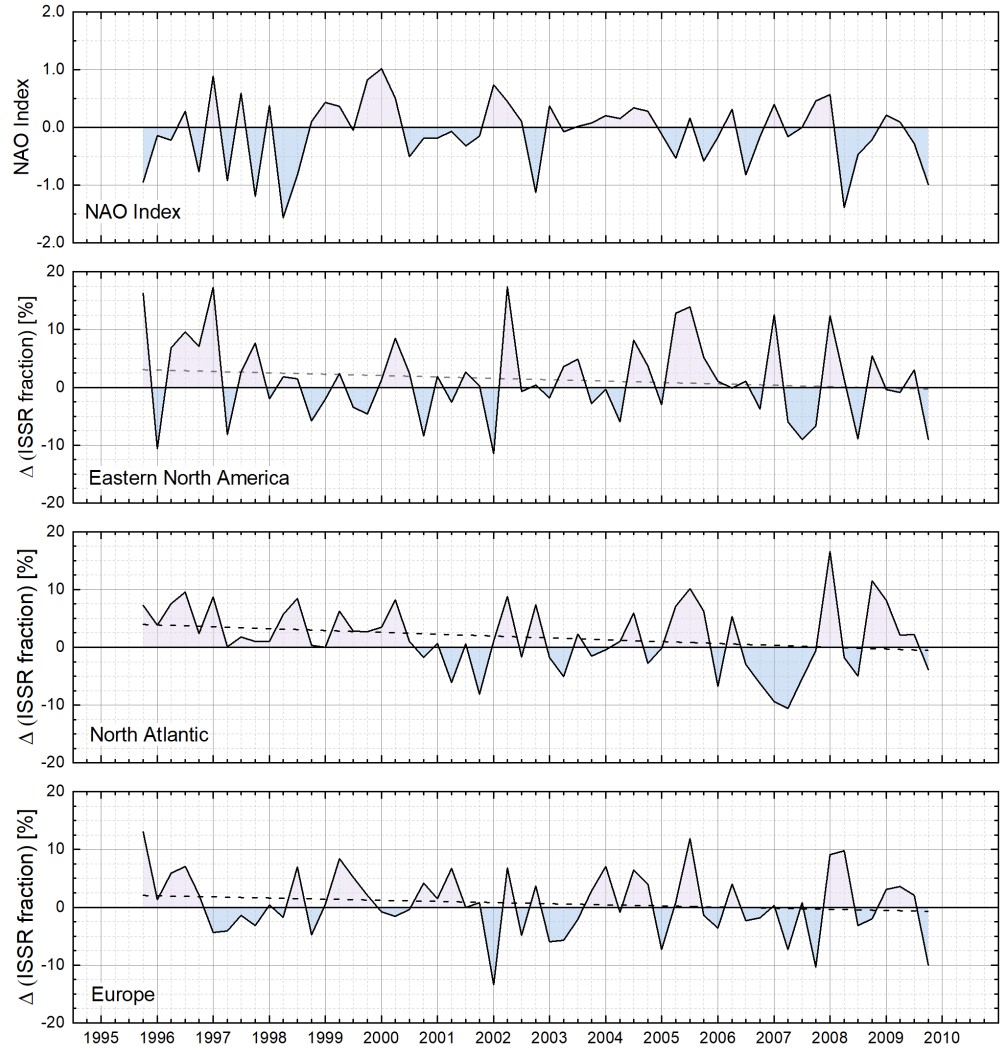

**Figure 15.** De-seasonalised time series of ISSR fraction (probability of occurrence) for latitudes 40°N to 60°N and for the regions (from top to bottom) Eastern North America (105°W to 65°W), North Atlantic (65°W to 5°W) and Europe (5°W to 30°E).

The de-seasonalised time-series thus show positive and negative deviations from the long-term seasonal average values. The resulting time series are presented in Figure 15. As for the time series of ISSR occurrence, we performed a trend analysis and added the obtained trend lines to Figure 15. Respective decadal slopes are -1.95% ± 1.77% for Eastern North America, -3.21% ± 1.78% for the North Atlantic, and -2.39% ± 2.29% for Europe and indicated uncertainties of the determined slopes refer to one standard deviation. Thus none of the slopes differs significantly from zero, and similar to the time series of ISSR occurrence, we do not observe significant trends for the seasonal deviation of ISSR occurrence from the long-term average for the three target regions.

One potential weather phenomenon driving the deviation of seasonal ISSR occurrence from the long-term average in the investigated region is the North Atlantic Oscillation (NAO). The NAO index describes the deviation of the pressure difference between the Iceland low and the Azores high pressure systems from the long-term average value. As an example, a positive value of the NAO index indicates that Δp (Iceland L to Azores H)

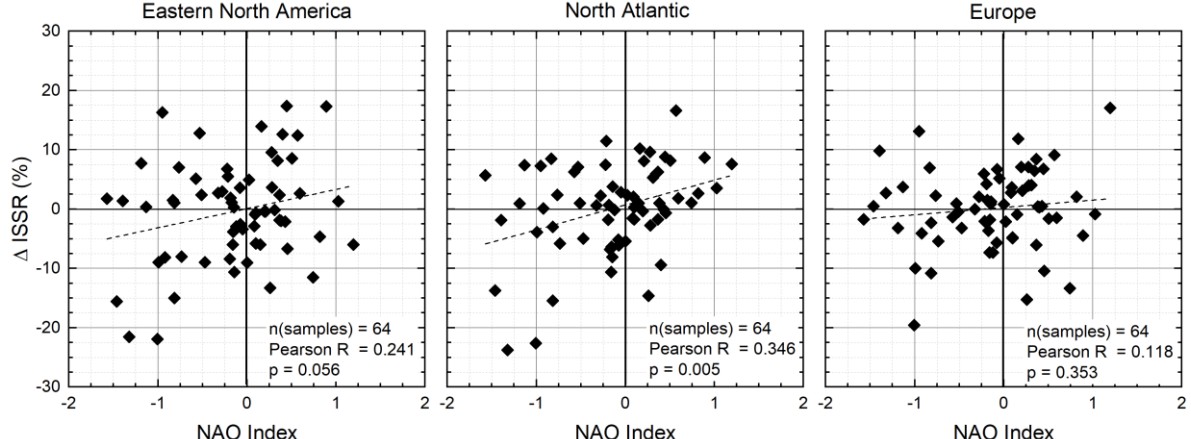

**Figure 16.** Correlation analysis with respect to the correlation of signs between NAO index and deviation of ISSR occurrence from the long-term average (Δ ISSR) for the target regions; numbers indicate he results from the correlation analysis with respect to number of samples n, Pearson R and significance level p..

is larger than on average. This larger pressure difference causes stronger westerly winds and thereby more active storm tracks over the North Atlantic. Under such conditions we would expect a higher probability of ice-supersaturation in the uppermost troposphere due to more frequent warm conveyor belts that can induce the formation of ISSRs in the upper troposphere (Spichtinger et al., 2005). Such a positive correlation between NAO

and cirrus cloud cover is reported from an analysis of cirrus cloud cover data from the International Satellite Cloud Climatology Project and relative humidity data from ECMWF/ERA40 by Eleftheratos et al. (2007).

To investigate this potential link, we added the seasonally averaged NAO index to Figure 15 (top panel). Since there is no immediate evidence given for a link between the NAO index and the deviation of ISSR occurrence from the long-term average (Δ ISSR), we further searched for a potential link of signs in the sense that positive

and negative NAO index values are associated with positive and negative deviations of ISSR occurrence from the long-term average, respectively. The results of this cross-correlation analysis are presented in Figure 16.

For the regions Eastern North America and Europe the correlation between NAO index and Δ ISSR is not statistically significant. For the North Atlantic however, the results of the cross-correlation analysis indicate statistical significance at a level of 99%.The obtained correlation of signs is in line with the observation that the

730 occurrence of ice-supersaturation is well correlated with the storm track activity (Spichtinger et al., 2003b; Gettelman et al., 2006; Lamquin et al., 2012).

## 4. Summary and Conclusions

The European Research Infrastructure IAGOS (from 2011) and its predecessor programme MOZAC (1994 -

735 2010) perform global-scale routine in-situ observations of relative humidity with respect to ice ($RH_{ice}$) by using instrumented passenger aircraft. The validated $RH_{ice}$ data set from the MOZAIC period between 1995 and 2010 was analysed for latitudes 40 °N to 60 °N and for the regions Eastern North America (105 °W to 65 °W), North Atlantic (65 °W to 5 °W) and Europe (5 °W to 30 °E) to study the occurrence of ice-supersaturated regions (ISSR) in the uppermost troposphere and tropopause layers. Determined seasonal cycles agree very well with

740 observations of ISSR occurrence from radio soundings (Spichtinger et al., 2003a) and from satellite observations (Spichtinger et al., 2003b; Lamquin et al., 2012).

The high vertical resolution of the MOZAIC $RH_{ice}$ data set with 30 hPa layer thickness allows the determination of the vertical position of the ice-supersaturated air masses with respect to the thermal tropopause. It occurs that the fraction of ice-supersaturated regions is largest for the atmospheric layers of 60 hPa thickness, directly below the thermal tropopause.

Comparing the ISSR fraction from MOZAIC in-situ observations with the high-cloud fraction from satellite instruments (Stubenrauch et al., 2010) yields remarkably close agreement between the two different observations and supports the interpretation that cirrus clouds exist to a considerable fraction also in ice-subsaturated air masses, depending on their state of life. . This interpretation is also supported by first exemplary analyses of simultaneous observations of $RH_{ice}$ and ice crystal number density $N_{ice}$ from the ongoing IAGOS programme (Petzold et al., 2017). In addition, the close agreement between satellite-based observations of ice cloud occurrence and the MOZAIC/IAGOS in-situ observations of ice-supersaturation demonstrate the unique contribution, MOZAIC and today IAGOS long-term observations can make to this scientific area, in particular with the detailed seasonality of ISSR occurrence over different regions. Future work will combine $RH_{ice}$ and $N_{ice}$ observations which are now available from IAGOS and link them to AIRS time series.

The finding that ice-supersaturated air is generally colder and associated with higher absolute humidity and – in case of observation inside or above the tropopause layer - carries less ozone than the surrounding air masses is in close agreement with reported results for temperature and absolute humidity. However, we were also able to use ozone as a tracer for stratospheric air and calculate the troposphericity of ice-supersaturated and subsaturated air masses. The analysis yields a significant impact of tropospheric air even on ISSR observed above the thermal tropopause. The thermodynamic features together with the increased troposphericity indicate vertical mixing in the vicinity of the tropopause layer as one important formation process of ice-supersaturation. Future work in this direction will be conducted, once the full IAGOS data set on $RH_{ice}$, ozone and ice clouds is validated and available.

Over the investigated period of 15 years, no significant trends are observed, neither for the occurrence of ISSR nor for the deviation of seasonal ISSR occurrence probabilities from the long-term average. This statement is valid for all three investigated regions. Yet, we identify a significant correlation of signs between the NAO index and the deviation of seasonal ISSR occurrence probabilities from the long-term average for the North Atlantic, whereas no such correlation was found for Eastern North America and Europe. The resulting interpretation is that a positive NAO index correlates with increased occurrence of ISSR (positive deviation from the long-term average). This interpretation is in agreement with the understanding that a positive NAO index leads to an increased storm track activity which then may induce more frequent formation of ISSRs in the upper troposphere.

Finally, in a concomitant study by Reutter et al. (2020) MOZAIC $RH_{ice}$ observations have been compared to ECMWF ERA-Interim data and significant deviations are reported for ice-supersaturated conditions, both in number and strength of supersaturation. Accurately representing the magnitude of ISSR as well as its coexistence with ice crystals are crucial for quantifying radiative forcing, since mistakenly representing ISSR as ice crystals can lead to an average decrease of 2.7 W m$^{-2}$ in surface radiation (Tan et al., 2016). The high quality and very good resolution of MOZAIC and later IAGOS $RH_{ice}$ observations will certainly help to further improve the representation of ice-supersaturation in ERA 5 as well as in numerical weather and climate forecasting models.

## Author contributions

AP designed the study and prepared the manuscript, with contributions from all co-authors; PN, SR, MR, and HGJS performed the quality control and analysis of MOZAIC/IAGOS water vapour data; FB provided the thermal tropopause pressure levels and performed the quality control and analysis of temperature data; MK and NS contributed the analysis of the research aircraft data. PNed performed the quality control and analysis of ozone data; AW and PS contributed to the interpretation of the study results.

## Competing interests

The authors declare that they have no conflict of interest.

## Data Availability

The IAGOS data are available through the IAGOS data portal at https://doi.org/10.25326/20. The IAGOS time series data set used for this analysis is referenced at https://doi.org/10.25326/06.

We used the following data versions for our analyses:

Version 1.0 of IAGOS air_temp and air_stag_temp data, based on the method described in Helten et al. (1998).

Version 3.0 of IAGOS RHL, RHI and H2O_gas data, based on the calibration techniques and data inversion algorithms published in Helten et al. (1998). In addition, version 3.0 has implemented the in-flight calibration technique adapted from Smit et al. (2008), which adjusts for an offset drift of the MCH sensor during a flight period.

## Acknowledgements

Parts of this study were funded by the German Ministry for Education and Research (BMBF) under Grant No. 01LK1301A as part of the joint research programme IAGOS Germany. MOZAIC/IAGOS data are created with support from the European Commission, national agencies in Germany (BMBF), France (MESR), and the UK (NERC), and the IAGOS member institutions (http://www.iagos.org/partners). The participating airlines (Deutsche Lufthansa, Air France, Austrian, China Airlines, Iberia, Cathay Pacific, Air Namibia, Sabena) supported IAGOS by carrying the measurement equipment free of charge since 1994. The data are available at https://doi.org/10.25326/20 thanks to additional support from AERIS. MK thanks JGU Mainz for support as a GFK fellow. Finally, the authors gratefully acknowledge the highly valuable comments from three anonymous reviewers and M. Diao which helped improving the manuscript.

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
