# Peer review of "Ice-supersaturated air masses in the northern mid-latitudes from regular in-situ observations by passenger aircraft: vertical distribution, seasonality and tropospheric fingerprint"

_Atmospheric Chemistry and Physics, 2019_

## Referee Comment (RC1) · Anonymous Referee #2 · 9 Oct 2019

**1 General comments**

This paper presents an analysis of data of relative humidity for the period 1995 to 2010, obtained via instrumented passenger aircraft in the framework of IAGOS and MOZAIC over the northern midlatitudes (40-60°N) in 3 longitude ranges: Northeast America, North Atlantic, and Europe. The huge amount of data makes it possible to cover several vertical altitude ranges of 30 hPa thickness with sufficient data density to allow robust statistics. The altitude bands are defined with respect to the thermal

and the dynamical tropopause, respectively, and the "troposphericity" (i.e. the fraction of tropospheric origin in an airparcel) is determined using simultaneous data of ozone VMR. The focus of the study is ice supersaturation.

The data show, that ISSRs (ice supersaturated regions) occur most often directly below the thermal tropopause, rarely directly above it, and almost never further up in the stratosphere. There is a distinct seasonal cycle in all 3 considered regions, but no significant trend over the 15 years of the study. The north-atlantic oscillation seems to have an influence on the occurrence of ISSR over the North Atlantic and Europe, but not over North America, which is physically plausible. ISSRs are colder and moister than their subsaturated surroundings (in agreement with earlier results), and they are poorer in ozone and have accordingly a larger troposphericity than the subsaturated environments, which is plausible as well considering the fact that most ice supersaturation is formed by uplifting of airmasses. The data show also that ice supersaturation is very closely related to cloudiness, that is, most ice supersaturation is found within clouds.

Thus, this paper provides a number of new and interesting results. It is well written for the most part. There are only a few points where I think the presentation can be made clearer and where perhaps the discussion can consider one or two more points. The paper should surely be published after the issues below are addressed.

**2 Major issues**

1) The paragraph lines 388 to 395 should be reworked; it is unclear what you did. For instance, what is an "average occurrence probability"? Do you mean the average frequency of occurrence or something else? What is the pdf of ISSR occurrence? Is this simply the probability of ISSR occurrence? I also do not understand what the distinction between seasons has to do with statistical quantities like median and percentiles
and how these two non-related things are linked here in one sentence. And finally, what is the statistical entity?

2) The comparison between statistics relative to the thermal and the dynamic tropopauses is not easy to understand, perhaps because it is unclear what exactly has been done. The first issue that must be clarified is whether the tropopause pressure and the pressure of the 2 PVU surface are available for each single measurement or are there only average values available, which would be bad for the analysis. What is the average  $\Delta p$  with respect to (wrt) the thermal tropopause of the 2 PVU surface? It seems that 2 PVU occurs quite often or in the majority of cases in the UT1 layer. This should be stated. However, it does not seem as if the mean profiles wrt to the dynamical TP are just shifted versions of the profiles wrt the thermal TP definition. Is this a consequence of averaging or why is this so? Next, Table 3 lists under Thermal TP numbers which I expected to be annual mean values of numbers in Table 1 under AVG, but a guick calculation shows something different. Is this because of different weights for the seasons or what is the reason? (For instance take the  $20.0 \pm 6.5$  in column 3 of Table 3. Should this not be the mean of 21.3, 18.6, 17.1 and 18.4 in the right hand box AVG in table 1?). And finally, Fig. 10 shows different behaviour in the left and right panels. Although you give a good physical explanation, I am not fully convinced. In the thermal TP coordinates there is a strong difference between ISSR and non-ISSR already at 30 hPa above the TP, but in the dynamical TP version there is only a much smaller difference at 60 hPa above the 2 PVU level. Is it possible that, on average, the 2 PVU surface is more than 60 hPa below the thermal TP? Eventually we should expect to see qualitatively the same profiles, irrespective of the actual choice of a vertical coordinate, isn't it?
**3 Minor issues**

1) Occasionally the term UTH is used. This should be avoided. UTH is a radiance based measure of a kind of mean relative humidity in a thick layer in the upper troposphere; it is a non-local measure. In contrast, IAGOS and MOZAIC yield local measures of relative humidity, and even after averaging over certain layers they should not be called UTH to avoid confusion. Better call it "the relative humidity field of the UT" or similar, but avoid UTH.

2) The last sentence of the introduction should be changed. The middle atmosphere is hardly relevant for IAGOS.

3) Figure caption 1: I do not understand what you mean with the pdf of data points. Do you mean simply the number of measurements or the fraction of measurements in a certain grid box?

4) Line 251: "Figure 4 illustrates ... of RH...": is this with or without IFC applied?

5) Figure 4: Please describe how these data are normalised. Is the sum over the whole figure 1?

6) I am a bit puzzled by the kind of averages applied. In line 176 it says " data are consolidated to 3-months season files", but in line 292 we have monthly mean profiles. Furthermore, is the distance of the current pressure level of a single 4-sec data point to the tropopause pressure recorded for every data point, or is the tropopause pressure averaged over a month and this average taken as reference (which would be a bad strategy to my view)?

7) Figures 6 and 7: why do you use geometrical height instead of the  $\Delta p$  for these figures?

8) Line 373/4: The statement may be wrong or perhaps right for the wrong reason. If the mean value of a positive quantity gets small, the variability usually gets smaller

**ACPD**
as well. Thus I suggest you to consider instead of  $\sigma$  the normalised  $\sigma$ :  $\sigma/\mu$  (i.e. std. deviation divided by mean value).

9) Comparison with RS Lindenberg (Figure 12a): has the same pressure band be selected for the RS data as for the MOZAIC/IAGOS data or are these the plain overall figures from the old publication?

10) It is not clear why CALIPSO can have higher cloud frequency than ISSR frequency. The argument that CALIPSO sees subvisible cirrus (SVC) explains only that it sees more than other satellite instruments do, unless SVC can survive in subsaturated air for a quite long time, where it is unclear to me what quite long actually means. I think that the reason for this result is rather in the difference of local vs. non-local measurements, just as the cloud fraction in a single level is always smaller than the cloud coverage over several levels.

11) Final paragraph of 3.5: Misuse of "cross-correlation". A cross-correlation is simply a correlation between two different quantities (as opposed to auto-correlation). Furthermore I suggest to replace "probability" in this paragraph with "fraction" in order to avoid wrong connotations. I am also a bit unhappy with "correlation" since I do not see that the two time-series have been correlated (in this case indeed cross-correlated) which would easily be possible. In this case there are also standard techniques to evaluate the statistical significance of the result (i.e. whether the correlation coefficient is significantly different from zero). The sentence "we consider ... statstically significant" should be deleted. This is not a question of "consideration" but of calculation. However, the physical explanation for your result is plausible. In the same sense, the statement in line 645 "significant correlations..." should be reformulated, for instance "physically plausible influence of the NAO on ISSR occurrence is detected in the time series...).

12) The data show that "by far the largest part of ISSR occurs inside cirrus clouds". You should ask yourself: Why? Doesn't this imply that most ISSR reach the humidity
threshold for heterogeneous or even homogeneous freezing shortly after the airmass began to be supersaturated? Are there further implications? Since the NH has more heterogeneous IN than the SH, do you expect that on the SH a smaller fraction of ISSR is inside clouds?

**4 Language, typos, etc.**

Line 71: remove comma after supersaturation.

Lines 227/8: Details of ... in detail .. Please reformulate.

Line 275: Please replace "validation" with "comparison". And then "The MOZAIC ... IS plotted..."

Lines 327 and 329: change to "north" and "south" (i.e. use lower case).

Line 340: I suggest to write "Similar AVERAGE values of ..."

Line 353: delete "set".

Line 491: add comma after dynamics.

Line 579: thus HAS positive ...

Figure 15: in my printout there is no grey shading.

Line 596: warMer and moister (or do you indeed mean more, i.e. a larger quantity of, moist air?)

Line 613: MOZAIC (with I).

---

## Referee Comment (RC2) · Anonymous Referee #3 · 24 Oct 2019

This manuscript describes in situ measurements of relative humidity (RH) in the upper troposphere and lower stratosphere (UTLS) from commercial aircraft and presents a detailed statistical examination of ice-supersaturated air masses (RHice >100%). The analysis is confined to a region of high measurement density between latitudes  $40^{\circ}N-60^{\circ}N$  and longitudes  $105^{\circ}W$  and  $30^{\circ}E$ , for the years 1995-2010. Several conclusions are drawn regarding the probabilities of encountering regions of ice supersaturation (ISSR) in three different longitude regimes, based on distance from the tropopause and season. There is also a minor attempt to attribute interannual variations in these

probabilities to the North Atlantic Oscillation (NAO).

Major Comments ========

Uncertainties are calculated and presented (typically 1 standard deviation) for most mean values derived in this paper. However, the uncertainties are often ignored when interpreting the mean values and making quantitative statements about them. One example, in Lines 396-397: "... the average ISSR occurrence probability is 29% in the troposphere and increases to 34% when approaching the tropopause layer." Given that the standard deviations of these mean values are each at least  $\pm$ 9%, the two averages are not statistically different, and the claimed "increase" is not significantly different from zero. A second example is Figure 11, where a horizontal line (indicating no seasonality) can easily be drawn within the uncertainties in each panel that show "annual cycles". Therefore, the statement (Line 506), "For all regions, ISSR occurrence probabilities are highest in the winter/spring and lowest in summer ..." is not supported by these seasonal averages with their statistical uncertainties.

In view of this, why are most uncertainties in this paper calculated and presented as 1 standard deviation when the vast majority of scientific uncertainties are reported as 95% confidence intervals (i.e., approximately 2 standard deviations for large sample sizes)?

The "occurrence probability" statistics are simple to understand, based on the numbers of RH measurements reflecting subsaturation, saturation, or supersaturation during a flight segment, an entire flight, or a number of flights. But it is not clear how "occurrence probability standard deviation" statistics were calculated. Are these based on calculating an average of the occurrence probabilities for a number of flights, reflecting the variability of the occurrence probabilities for individual flights around the average? This should be briefly explained, early in the paper, so the reader can immediately grasp the concept of the "occurrence probability standard deviation".

Why is the requirement for supersaturation RHice >100% when the measurement uncertainties are approximately 5% RH in the middle and upper troposphere? If some part of these uncertainties is a systematic error (a high bias of 3%, for example), wouldn't this lead to artificially high occurrence probabilities if measurements of a 98% RH air mass are 101% RH? How much do the occurrence probabilities decrease if you instead require RHice >103%, or even RHice >105% for supersaturation?

I'm not convinced that the comparison of supersaturation occurrence probabilities for atmospheric layers relative to the lapse rate ("thermal") tropopause vs the 2 PVU ("dy-namical") tropopause shows much of a difference. If 95% confidence intervals of the mean values in Table 3 are considered, none of the "thermal" and "dynamical" averages for any atmospheric layer are statistically different. A lot of text, Figures and Tables are devoted to this comparison, and what does it show? Very little, in my opinion. Instead (or in addition), I'd prefer to see some assessment of the accuracy of the ERA-Interim tropopause heights that are absolutely critical to this paper. Since ozone mixing ratios were also measured as part of MOZAIC, and ozone can be used to define a "chemical" tropopause, can you compare ozone-defined tropopauses to the ERA-Interim tropopauses to evaluate at least the consistency of the latter? For example, if ERA-Interim puts the tropopause 1 km above the aircraft and the ozone mixing ratio is 1 ppm that indicates a large (>1 km) error in the tropopause height. I'm not suggesting a full-scale comparison, but rather some comparisons that illustrate the possible errors in tropopause heights.

Water vapor mixing ratios are discussed in some sections of the paper and are shown in some Figures, but nowhere in the paper is there a description of how these were determined. Were they measured directly with different instruments (as implied in Line 17 of the abstract) or were they calculated from the RH measurements, requiring concomitant measurements of pressure and temperature with their associated uncertainties?

There are some awkward and confusing sentences in the paper that could benefit from re-writing. I will point out a few of these below, but I suggest the paper be proofread by a native English speaker to clean up and clarify some sentences.

**Minor Comments =========**

L64: Why would an "increase in pressure" change the RH of an air mass? RH is the partial pressure of water vapor divided by the saturation pressure over ice at a given temperature. Neither of these is affected in any way by an "increase in pressure". Please either explain this statement more clearly or remove it.

L109-111: Why are radiosonde network measurements of RH "considered insufficient for detecting trends and variability in UTLS water vapor"? I believe the GRUAN radiosonde data product for RH will be sufficient in this regard, and that GRUAN represents another existing global-scale network of in situ observations of atmospheric composition in the Ex-UTLS. A good reference for GRUAN is:

Bodeker, G.E., Bojinski, S., Cimini, D., Dirksen, R., Haeffelin, M., Hannigan, J., Hurst, D., Leblanc, T., Madonna, F., Maturilli, M., Mikalsen, A., Philipona, R., Reale, T., Seidel, D., Tan, D., Thorne, P., Vömel, H., and Wang, J.: Reference Upper Air Observations for Climate: From Concept to Reality, B. Am. Meteorol. Soc., 97, 123–135, https://doi.org/10.1175/bams-d-14-00072.1, 2016.

L180: Please insert "attached" between "inlet" and "to"

Lines 23-25: This statement implies there is an increasing trend in summertime water vapor mixing ratios in the lowermost stratosphere, but no similar trend in RHice. I don't think this is what you mean to say, rather that mixing ratios in this region are highest during summer months without corresponding maxima in RHice. If this is the case, doesn't it imply that temperatures in this region are also highest during summer months?

L51: The term "tropopause layer" is used throughout this paper, but where is it geophysically defined? On page 5 you limit the TPL to "tropopause pressure  $\pm$  15 hPa", but that's a definition that is neither common or geophysically-based. It would enlighten the reader to know why you chose these limits for the TPL.

L184-189: How about the uncertainty of RH measurements in the lower stratosphere? LOD is one measure, but since you determine supersaturation occurrence probabilities for several layers above the tropopause this must be somewhat known.

L211: Please change "sequences" to "segments". "Sequences" is also awkward in L105.

Figure 3: I think you intend "w/t" to mean "without" in both panels. Please change to "w/o".

L251-253: Presumably Figure 4 shows the RHice with the IFC applied, so please make this clear.

L286-287: "highest possible quality achievable by this kind of routine observations" sounds great, but what does it actually mean? This sounds like an advertisement instead of a scientific claim and I suggest toning it down or removing it.

L290-297, Figure 6: Up to this point, everything has focused on RH measurements and the tropopause-relative pressure bins you have defined. Here, the discussion suddenly turns to water vapor mixing ratios and tropopause-relative altitude bins. As above, where do the VMR data come from? And why does Figure 6 use altitude instead of pressure (or log pressure) as the vertical coordinate?

L327: "are bounded to the Great Lakes area and further North". Given the size of Figure 1, it is difficult (without magnification) to find the Great Lakes. A better description would be "are within the northern half of the continental USA and southern half of Canada".

Figure 8, Tables 1 and 2: It is not clear what the tropopause-relative pressure boundaries are for the different layers. Are the average values plotted (Figure) and presented (Tables) at deltaP = -30 hPa for the layer bounded by TPpress-15hPa and TPpress-45hPa? This should be clearly stated.

Given the standard deviations (Table 2), are the average values for different seasons

**C5**

or longitudinal regions (Table 1) statistically different at the 95% level of confidence?

L373: Why is the annual cycle of UTH increasingly damped as you get closer to the tropopause?

L377: As noted above, please explain how an increase in pressure can cause supersaturation.

L429-433: This long sentence is confusing and requires re-wording for clarification.

Figure 9: What does the black horizontal line represent at +70 hPa in the H2O panels?

L454-455: The 33.5 ppb O3 value from ERA-Interim is representative of what altitude and region?

L460: Why the sudden switch from P99 to P95, without explanation?

Figure 10: Error bars for each marker would clearly show if the troposphericity values are statistically different (or not).

L500: Here in the text you claim that Figure 11 shows results for the "top UT layer", but the caption for Figure 11 says "calculations were conducted for the two UT layers positions closest to the tropopause."

L506-507: This statement is not supported by the average values when their uncertainties are considered.

L513: I assume the Lindenberg radiosonde RH data has been corrected using the GRUAN-recommended corrections? It is important to say this because the reader may assume that uncorrected RH data from radiosondes are good enough (they are not!). You might also reference the paper describing corrections to the Vaisala RS92 data:

Dirksen, R. J., Sommer, M., Immler, F. J., Hurst, D. F., Kivi, R., and Vömel, H.: Reference quality upper-air measurements: GRUAN data processing for the Vaisala RS92 radiosonde, Atmos. Meas. Tech., 7, 4463–4490, https://doi.org/10.5194/amt-7-4463-

2014, 2014.

Figure 12: I don't see any information about the layer or layers for which the results are shown. Please state this in the caption.

L549: typo "15y ears"

L530: Why only 15 months of Lindenberg radiosonde data? There are more than 9 years of GRUAN-corrected Vaisala RS92 RH data from Lindenberg.

L551: "fits well" is an overstatement since the DJF and MAM averages for Lindenberg lie outside the MOZAIC mean  $\pm$  1 standard deviation envelope.

L549: I don't know what a "first exemplary analysis" is. Please explain.

L553: Trend analyses are performed on supersaturation occurrence probabilities based on which tropopause definition?

L579: "thus" must be a typo because it makes no sense in this sentence. Also, please change "long-term average values" to "long-term seasonal average values."

L581-582: Three significant figures for trends and their uncertainties is not justified when the uncertainty values are nearly as large as the trends themselves. Why present the 1 standard deviation uncertainties when, presumably based on 2 standard deviation uncertainties, you claim in the next sentence that none of the trends are significant?

L597: If the westerlies bring "warmer and more moist air to Europe", why would you expect a higher probability of supersaturation in the UT? More moisture increases the RH, but warmer air lowers the RH.

L608: "we consider the correlation of signs statistically significant". This is a very qualitative conclusion that needs support from a quantitative explanation.

L651: "which then generates more frequently ISSR" is awkward phrasing. Please rewrite.

---

## Referee Comment (RC3) · Anonymous Referee #1 · 25 Oct 2019

Review of "Ice-supersaturated air masses in the northern mid-latitudes from regular in-situ observations by passenger aircraft: vertical distribution, seasonality and tropospheric fingerprint" by Andreas Petzold et al.

This manuscript uses a humidity data set from commercial aircraft to analyze humidity and ice supersaturated regions in the upper troposphere. The manuscript is quite good, with careful and comprehensive analysis of the uncertainties and corrections in the data set. However, some of the analysis doesn't quite make sense, especially the discussion of thin cloud occurrence and correspondence of sign with NAO at the

end of the manuscript. That analysis needs significant modification as I outline in the specific comments below. Except for this and minor comments, this manuscript should be publishable in ACP with appropriate revisions.

Page 2, Line 67: embedded

Page 3, L78: close to the tropopause. Which tropopause? Thermal is specified in the next sentence.

Page 3, L82: again, which tropopause definition?

Page 4, L127: what is the horizontal resolution?

Page 4, L141: what does 30 - 65 hours mean? In each grid box? How many flights per grid box per season?

Page 5, L166: what is the vertical and horizontal resolution of ERAI in the UTLS?

Page 5, L178: wouldn't it be better to have PDFs in each season since it seems there are a lot of data points. More statistics than the mean it seems are available.

Page 8, L259: fig 4. The RHice line looks solid to me as well.

Page 8, L270: was Research flight data selected for the same geographic regions as MOZAIC data shown in figure 5?

Page 10, L308: might be better to state that specific humidity is lower in summer over E. N. America in the UT.

Page 11, L332: the vote part of our study focuses on the vertical....

Page 11, L353: Table 1 just restates the right column from Figure 8 correct? maybe it is not necessary? Can you put the standard deviations from Figure 2 on the plot in figure 8?

Page 13, L396: not exactly clear to me how this is different than the relevant panel in figure 8. Just adding the dynamic tropopause?

Page 14, L413: this bothers me a bit. The thermal tropopause is a robust barrier in the tropics, but here the average RH is 80% or less at the tropopause, so it does not need to be a robust barrier. Also motion is not purely vertical here., but more horizontal and isentropic. Please explain.

Page 14, L435: where does the ozone come from? Also MOZAIC I assume? Please specify. What is the minimum detectable concentration? And can you provide a validation reference and maybe a sentence or two.

Page 15, L449: please define 'their' with a reference. Assume it is the same as previous paragraph, but please be specific.

Page 16, L480: why is the thermal tropopause a transport barrier and the dynamical tropopause in extratropics not a barrier? I'm not sure I understand your logic here,

Page 18, L531: extra space in years

Page 18, L545: I don't think the casual analysis of the frequency of cirrus is helpful. Cirrus frequency is a function of instrument as you note. And cirrus layers need not be supersaturated, so there need be no link here.

Page 18, L548: I do not think you can argue that just because cirrus and ISSRs have about the same frequency (But ISSR is Lower from the best and most sensitive sensor), that most cirrus occur in ISSRs. You need to show physical and temporal coincidence.

Page 21, L607: I'm not sure I follow this here. If there is a non zero correlation then there is a correspondence of signs is there not? Not familiar with the method.

Page 21, L613: why does the same sign in a bit over half the cases mean statistical significance? Isn't 50% totally random?

Page 21, L615: How does a correspondence of anomalies correlate with the storm track?

2019.

---

## Author Comment (AC1) · 1 Mar 2020

**Response to Reviewers**

**GENERAL REMARKS**

We thank all three reviewers for their insightful reviews and helpful and constructive comments. Responding to their comments helped improve the manuscript significantly.

All reviewers raised the point that the discussion of the different tropopauses (thermal, dynamical) lacks clarity and is confusing, and also partially misleading. As a general response to all reviewers, we put now the entire discussion of the observations and their interpretation into the framework of the extratropical transition layer (ExTL) and the vertical tracer profiles observed in that region. As discussed e.g. by Hoor et al. (2004), Pan et al. (2010), and Gettelman et al. (2011), it is found that on average the dynamical tropopause is situated slightly below the thermal tropopause and trace gas gradients are more sharp above the thermal tropopause compared to the dynamical tropopause. This is exactly the same behaviour we observe here and the interpretation of our results is now linked to that known feature of tracer characteristics in the ExTL. So, we modified the following sections considerably:

**1. Introduction:**

We added the following paragraphs to the introduction:

[revised manuscript text omitted]

> LMS : $p < p_{therm.TPH}$ -15hPa; which is limited by the maximum cruise altitude with $p \approx 190$ hPa;
> TPL : $p = p_{therm.TPH}$ ± 15hPa;
> UT : $p > p_{therm.TPH}$ + 15hPa; limited to lower altitudes by $p < 350$hPa."

**3.3 Physico-chemical signature of ice-supersaturated regions in the vicinity of the tropopause**
This section is significantly modified. In general terms, the introducing paragraph to this section reads now as follows:

"In order to study the formation history of ISSR and involved processes, we analysed the occurrence frequency and physico-chemical signature of ISSR around the tropopause layer and referred our analyses to both the thermal and the dynamical tropopause. We want to recall the tropopause definitions given in Section 2.1. The thermal tropopause according to WMO criteria (WMO, 1957) is usually seen as an effective transport barrier hampering troposphere-stratosphere exchange, whereas the dynamical tropopause is commonly used for separating tropospheric and stratospheric air masses in studies on stratosphere–troposphere transport since it represents the lower bound of the ExTL. These complementary views on the tropopause have been developed from extensive CO - $O_3$ analyses, which showed that the 2 PVU surface approximately separates the troposphere from the stratosphere with the ExTL as a transition layer of about 2 km thickness above it and centred on the thermal tropopause(Hoor et al., 2004; Pan et al., 2010; Gettelman et al., 2011). These tracer studies in the extratropics showed that on average the dynamical tropopause is situated slightly

below the thermal tropopause and the gradients of CO and $O_3$ are much sharper across the thermal tropopause compared to the dynamical tropopause (Hoor et al., 2004; Pan et al., 2010). Similar features are observed for the gradients of temperature T, $H_2O$ VMR and $O_3$ VMR, shown in Figure 9 for the North Atlantic region. Similar to the tracer gradients, also the temperature gradient is sharper across the thermal tropopause compared to the dynamical tropopause. In addition, the results confirm the good agreement between the ERA-Interim thermal tropopause height indicated by $\Delta p_{TPH}$ = 0 hPa (blue lines), the lowest temperatures detected at $\Delta p_{TPH}$ = 0 hPa, and the chemical tropopause, indicated by $O_3$ VMR = 120 ppbv at $\Delta p_{TPH}$ = 0 hPa, and thus the consistency of the used data set. Furthermore, the analysis of the pressure difference between the thermal and dynamical tropopauses reveal an offset of approx. 25 hPa (15 - 35 hPa) which translates into an altitude difference of approx. 1 km (Neis, 2017).

[Figure]

**Figure 9.** Vertical distribution of temperature T (a), $H_2O$ VMR (b), and ozone VMR (c) relative to the 2 PVU dynamical tropopause and to the thermal tropopause; vertical distributions relative to the thermal tropopause are presented as percentiles [1, 25, 50, 75, and 99] by blue lines and relative to the 2 PVU tropopause conditions by red-shaded areas. "

The concluding paragraph reads now as follows:

"Recalling the structure of the ExTL with the 2 PVU dynamical tropopause at its lower bound separating the stratosphere from the troposphere, and centred on the thermal tropopause, we find that on the top of the ExTL non-ISSR air masses show a clear stratospheric signature, while ISSR air masses are still strongly influenced by mixing and carry a significant tropospheric fingerprint compared to the non-ISSR air masses. Above the dynamical tropopause and thus inside the ExTL, the influence of mixing increases gradually for both ISSR and non-ISSR air masses and the difference in tropospehricity is much less pronounced than near the top of the ExTL."

Detailed responses to reviewers concerns are given in the specific responses.

**Response to Reviewer #1**

**Reviewer:** *This manuscript uses a humidity data set from commercial aircraft to analyze humidity and ice supersaturated regions in the upper troposphere. The manuscript is quite good, with careful and comprehensive analysis of the uncertainties and corrections in the data set. However, some of the analysis doesn't quite make sense, especially the discussion of thin cloud occurrence and correspondence of sign with NAO at the end of the manuscript. That analysis needs significant modification as I outline in the specific comments below. Except for this and minor comments, this manuscript should be publishable in ACP with appropriate revisions.*

**Authors:** We appreciate the valuable review and respond in detail to the comments in the following.

**MAJOR COMMENTS**

**Tropopause definitions and roles:**

**Reviewer:** *Page 14, L413: this bothers me a bit. The thermal tropopause is a robust barrier in the tropics, but here the average RH is 80% or less at the tropopause, so it does not need to be a robust barrier. Also motion is not purely vertical here, but more horizontal and isentropic. Please explain.*

**Authors:** In contrast to larger-scale horizontal and isentropic transport processes which dominate troposphere-stratosphere exchange, we focus here on local vertical transport pathways, since these processes may influence the formation of ISSR within and above the tropopause layer. In addition we put now the entire discussion of the observations and their interpretation into the framework of the extratropical transition layer and the vertical tracer profiles observed in that region. As discussed e.g. by Hoor et al. (2004), Pan et al. (2010), and Gettelman et al. (2011), it is found that on average the dynamical tropopause is situated slightly below the thermal tropopause and trace gas gradients are more sharp above the thermal tropopause compared to the dynamical tropopause. This is exactly the same behaviour we observe here. To support this finding we added the vertically resolved data for $O_3$ VMR and $H_2O$ VMR with respect to the thermal and dynamical tropopauses as new Figure 9.

In summary, we modified the Introduction and Section 3.3 considerably; see the general remarks at the top of this document.

**Reviewer:** *Page 16, L480: why is the thermal tropopause a transport barrier and the dynamical tropopause in extratropics not a barrier? I'm not sure I understand your logic here.*

**Authors:** As explained above, we put now the entire discussion of the observations and their interpretation into the framework of the extratropical transition layer. The last paragraph of Section 3.3 summarising the results, was removed.

**Cirrus clouds and ice-supersaturation:**

**Reviewer:** *Page 18, L545: I don't think the casual analysis of the frequency of cirrus is helpful. Cirrus frequency is a function of instrument as you note. And cirrus layers need not be supersaturated, so there need be no link here.*

**Reviewer:** *Page 18, L548: I do not think you can argue that just because cirrus and ISSRs have about the same frequency (But ISSR is lower from the best and most sensitive sensor), that most cirrus occur in ISSRs. You need to show physical and temporal coincidence.*

**Authors:** We agree that the results of our comparison of ISSR occurrence from MOZAIC and of cirrus cloud occurrence from satellites might appear over-interpreted. Nevertheless, we believe that this comparison is useful for several reasons. First, there is a physical link between the formation of cirrus clouds and ice-supersaturation, since ice crystal formation only takes place at high ice-supersaturation. In addition, ice-supersaturation may occur inside cirrus also during their lifecycle, e.g., when the cloud is further lifted up while sedimentation leads to the reduction of available ice crystals for the further deposition of water vapour; see e.g. Krämer et al. (2016). Second, the lateral resolution of observations is significantly different between satellites and MOZAIC in-situ data, so that local-scale in-cloud fluctuations in $RH_{ice}$ might be seen by MOZAIC but not by the satellite

instrument. And finally, our motivation for this pilot comparison exercise was to investigate whether the observation probabilities of ISSR in MOZAIC and cirrus cloud occurrence in satellite data is of the same order of magnitude. For sure, our pilot exercise cannot replace a detailed study that picks up your valuable arguments, but will initiate this work. For these reasons we decided to keep this section in the manuscript. However, we softened our conclusions considerably, reflecting your concerns.

In response to your arguments, we added the following paragraph to the introduction to Section 3.4:

"Furthermore, the analysis of a large set of combined observation of $RH_{ice}$ and ice crystal number concentration $N_{ice}$ during a series of research flights (approx. 68000 observations of ice-supersaturation; Krämer et al., 2016) demonstrated, that approx. 80 % of the observed ice-supersaturation events are associated with in-cloud conditions. On the other hand, $RH_{ice}$ probability distribution functions inside cirrus clouds are characterised by most probable values at or slightly above ice-saturation at $RH_{ice} = 100\%$ (Krämer et al., 2009; Diao et al., 2014; Diao et al., 2015; Petzold et al., 2017) which means that cirrus clouds exist to a considerable fraction also in ice-subsaturated air masses, depending on their state of life."

The concluding paragraph of Section 3.4 was significantly softened and reads now:
"The good agreement between MOZAIC in-situ observations of ISSR occurrence with the high-cloud fraction from satellite instruments encourages further detailed studies on this matter. First exemplary analyses of simultaneous observations of $RH_{ice}$ and $N_{ice}$ which are now possible within the ongoing IAGOS programme already indicate a strong correlation of high $RH_{ice}$ values with its occurrence inside cirrus clouds (Petzold et al., 2017). "

**ISSR and NAO index:**
**Reviewer:** *Page 21, L607: I'm not sure I follow this here. If there is a non-zero correlation then there is a correspondence of signs is there not? Not familiar with the method.*
**Reviewer:** *Page 21, L613: why does the same sign in a bit over half the cases mean statistical significance? Isn't 50% totally random?*
**Authors:** The intention of this analysis was to investigate a potential link between the deviation of ISSR occurrence from the long-term average on one hand and the NAO index as an indicator of storm track activity on the other hand. As is obvious from the distribution of the data pairs in Figure 15, the correlation is weak in all cases. Equal occurrence of cases with positive correlation of signs, i.e., positive NAO index associated with positive deviation of ISSR occurrence from the long-term average and vice versa, and the occurrence of cases with negative correlation of signs, i.e., positive NAO index associated with negative deviation of ISSR occurrence from the long-term average and vice versa, indicates random occurrence. This is certainly the case for the region of Eastern North America (Fig. 15, left panel). Particularly for the region North Atlantic, the positively correlated cases occur at 61% of the observations and the negatively correlated cases at 39%. Over the Europe, this difference is again weaker.

To make our analysis of this potential link more robust, we skipped now the distinction between positively and negatively correlated signs and performed a cross-correlation analysis. The results of this analysis are now shown in the revised Figure 15 (now 16):

[Figure]

**Figure 16.** Correlation analysis with respect to the correlation of signs between NAO index and deviation of ISSR occurrence from the long-term average (Δ ISSR) for the target regions; numbers indicate he results from the correlation analysis with respect to number of samples n, Pearson R and significance level p.

The values of the correlation coefficient R and the significance level p indicate that for the North Atlantic this correlation is significant at a level of 99%. For the two other regions the correlation is not significant.

The description of the figure and the conclusions drawn read now:

"For the regions Eastern North America and Europe the correlation between NAO index and Δ ISSR fraction is not statistically significant. For the North Atlantic however, the results of the cross-correlation analysis indicate statistical significance at a level of 99%.The obtained correlation of signs is in line with the observation that the occurrence of ice-supersaturation is well correlated with the storm track activity (Spichtinger et al., 2003b; Gettelman et al., 2006; Lamquin et al., 2012)."

We also rephrased the respective statement in the Conclusions section, which reads now:
"Yet, we identify a significant correlation of signs between the NAO index and the deviation of seasonal ISSR occurrence probabilities from the long-term average for the North Atlantic, whereas no such correlation was found for Eastern North America and Europe."

*Reviewer: Page 21, L615: How does a correspondence of anomalies correlate with the storm track?*
**Authors:** To motivate the expected link between storm track activity and higher ISSR occurrence we include the following sentence in the introduction to this part of our analysis:
"As an example, a positive value of the NAO index indicates that Δp (Iceland L to Azores H) is larger than on average. This larger pressure difference causes stronger westerly winds and thereby more active storm tracks over the North Atlantic. Under such conditions we would expect a higher probability of ice-supersaturation in the uppermost troposphere due to more frequent warm conveyor belts that can induce the formation of ISSRs in the upper troposphere (Spichtinger et al., 2005). "

**SPECIFIC COMMENTS ABOUT THE DATA SET AND THE DATA ANALYSIS PROCEDURE**

**Reviewer:** *Page 4, L127: what is the horizontal resolution?*
**Authors:** We added the requested information and rephrased the sentence as follows:
The horizontal resolution of our data set is 1 km, set by the instrument time resolution of 4 s and the cruising speed of approx. 250 m s$^{-1}$. The vertical resolution is set to 30 hPa, which corresponds to a vertical distance of approx. 750 m at cruise altitude (Thouret et al., 2006) and assures sufficient statistical robustness of the conducted analyses. This vertical resolution is of similar order as the typical resolution of UTLS data with a vertical grid spacing of about 50 hPa in the vicinity of the tropopause (Reichler et al., 2003).

**Reviewer:** *Page 4, L141: what does 30 - 65 hours mean? In each grid box? How many flights per grid box per season?*
**Authors:** The descriptions means that per regional box between 30 hours and 65 hours of flight per season (3 months) were collected, which corresponds to 27,000 to 60,000 data points of 4 s duration each, per season per year. We rephrased the description as follows:
"The annual data coverage for each analysed regional box varies between 30 and 65 flight hours of MOZAIC aircraft per season (3 months) which corresponds to 27,000 to 60,000 data points of 4 s duration each, per season per year.

**Reviewer:** *Page 5, L166: what is the vertical and horizontal resolution of ERA-I in the UTLS?*
**Authors:** We rephrased the sentence for clarification as follows:
"The pressure levels of the thermal tropopause ($p_{TPHWMO}$) and the dynamical 2 PVU tropopause ($p_{TPHDYN}$) are derived from ERA-Interim (Dee et al., 2011) which uses 60 model layers with the top of the atmosphere located at 0.1 hPa."

**Reviewer:** *Page 5, L178: wouldn't it be better to have PDFs in each season since it seems there are a lot of data points. More statistics than the mean it seems are available.*
**Authors:** This is exactly what we did. In order to clarify our procedure, we rephrased the description as follows:
"Since each data set from one single flight provides only a one-dimensional snapshot of the state of the atmosphere along the flight track, and each aircraft cruises at a slightly different pressure level, the entire MOZAIC data are consolidated to season files of 3-months season files duration, allowing the analysis of vertical distributions of atmospheric state parameters on a robust statistical basis. For each season file, the statistical distribution (average and standard deviation, percentiles) of investigated properties (temperature, $O_3$ VMR, $H_2O$ VMR, $RH_{ice}$, ISSR fraction) is calculated with respect to the above defined UT, TP and LMS vertical layers. From these seasonal averages or percentiles, respective 15-year values including their variability are determined."

**Reviewer:** *Page 8, L270: was Research flight data selected for the same geographic regions as MOZAIC data shown in Figure 5?*
**Authors:** The MOZAIC data shown in Figs. 3 -5 refer to the complete data set. i.e., data include also measurements outside the geographic regions our analysis is focusing on. In consequence, the research aircraft data were also not restricted to regions but only to temperature and pressure ranges.
To clarify this, we rephrased the first paragraph of Section 2.4:
"The IFC method was applied to the full reanalysis data set from 1995 to 2010. Figure 3a illustrates the effect of the IFC method for the averaged RHice PDF for the entire MOZAIC data set, irrespective of the geographical regions the data were collected. The presented average PDF and variability is calculated from annual PDFs."

**Reviewer:** *Page 11, L353: Table 1 just restates the right column from Figure 8 correct? Maybe it is not necessary? Can you put the standard deviations from Figure 2 on the plot in Figure 8?*

**Authors:** Indeed, Table 1 restates the right column from Figure 8. However, we also want to give average values and standard deviations in a numerical format for comparison with our studies. We also added standard deviation from Table 2 to Figure 8, but then Figure 8 becomes highly unclear and hard to understand. Finally, our decision was to present only average values in Figure 8 and add numerical values in Tables 1 and 2.

**Reviewer:** *Page 13, L396: not exactly clear to me how this is different than the relevant panel in Figure 8. Just adding the dynamic tropopause?*
**Authors:** We added respective values for the dynamical tropopause, but skipped the separation into seasons. The latter step was mainly made because we wanted to analyse physico-chemical signatures of sub- and supersaturated air masses by analysing additionally the $O_3$ VMR for these air masses. Since MOZAIC $H_2O$ and $O_3$ data sets are not completely overlapping due to instrument performances, we decided to skip the distinction into seasons to generate a larger data ensemble for the targeted statistical analyses.
For clarity, we added the sentence "Please note that the ISSR fraction values compiled for the thermal tropopause correspond to the values listed in Table 2, but without separation into seasons."

**Reviewer:** *Page 14, L435: where does the ozone come from? Also MOZAIC I assume? Please specify. What is the minimum detectable concentration? And can you provide a validation reference and maybe a sentence or two.*
**Authors:** Indeed, the origin of the $O_3$ data needs to be introduced. We added the following sentence to Section 2.2 which is also renamed to RH and $O_3$ instrumentation:
"Since the launch of MOZAIC, the programme provides also $O_3$ VMR data in addition to $H_2O$ and $RH_{ice}$ observations. Aboard MOZAIC and now IAGOS aircraft, ozone is measured by means of a UV absorption instrument which is characterised by an instrument noise of ±2 ppbv and an integration time of 4 s (Nédélec et al., 2015). We used the collocated measurement of $O_3$ and $H_2O$ / $RH_{ice}$ for the characterisation of ice-supersaturated air masses with respect to a potential stratospheric influence."

**MINOR CORRECTIONS**
**Reviewer:** *Page 2, Line 67: embedded*
**Authors:** replaced as suggested

**Reviewer:** *Page 3, L78: close to the tropopause. Which tropopause? Thermal is specified in the next sentence.*
**Authors:** Sentence rephrased: "In contrast to the strong negative gradient in $H_2O$ VMR at altitudes below but close to the thermal tropopause, ISSR occur frequently in the humid and cold upper tropospheric air masses."

**Reviewer:** *Page 3, L82: again, which tropopause definition?*
**Authors:** "Thermal" added.

**Reviewer:** *Page 8, L259: Figure 4. The RHice line looks solid to me as well.*
**Authors:** Figure 4 was adjusted accordingly.

**Reviewer:** *Page 10, L308: might be better to state that specific humidity is lower in summer over E. N. America in the UT.*
**Authors:** We agree and rephrased the sentence as:
"…while for the Eastern North American region the upper free troposphere layers seem to exhibit higher specific humidity be more humid in winter than respective air masses over the ocean."

**Reviewer:** *Page 11, L332: the vote part of our study focuses on the vertical….*
**Authors:** For the sake of clarity we rephrased the sentence as: "Our study is focusing on …"

**Reviewer:** *Page 15, L449: please define 'their' with a reference. Assume it is the same as previous paragraph, but please be specific.*
**Authors:** We rephrased the sentence to be more specific:
"Using the $O_3$ VMR as a stratospheric air mass tracer and adapting the approach of Cirisan et al. (2013), we define the troposphericity parameter *m* for an ensemble of data characterised by median (med) and 99 percentile (P99) values as …"

**Reviewer:** *Page 18, L531: extra space in years*
**Authors:** Typo was corrected.

**Added References**

[revised manuscript text omitted]

---

## Author Comment (AC2) · 1 Mar 2020

**Response to Reviewers**

**GENERAL REMARKS**

We thank all three reviewers for their insightful reviews and helpful and constructive comments. Responding to their comments helped improve the manuscript significantly.

All reviewers raised the point that the discussion of the different tropopauses (thermal, dynamical) lacks clarity and is confusing, and also partially misleading. As a general response to all reviewers, we put now the entire discussion of the observations and their interpretation into the framework of the extratropical transition layer (ExTL) and the vertical tracer profiles observed in that region. As discussed e.g. by Hoor et al. (2004), Pan et al. (2010), and Gettelman et al. (2011), it is found that on average the dynamical tropopause is situated slightly below the thermal tropopause and trace gas gradients are more sharp above the thermal tropopause compared to the dynamical tropopause. This is exactly the same behaviour we observe here and the interpretation of our results is now linked to that known feature of tracer characteristics in the ExTL. So, we modified the following sections considerably:

**1. Introduction:**

We added the following paragraphs to the introduction:

[revised manuscript text omitted]

> LMS : $p < p_{therm.TPH}$ -15hPa; which is limited by the maximum cruise altitude with $p \approx 190$ hPa;
> TPL : $p = p_{therm.TPH}$ ± 15hPa;
> UT : $p > p_{therm.TPH}$ + 15hPa; limited to lower altitudes by $p < 350$hPa."

**3.3 Physico-chemical signature of ice-supersaturated regions in the vicinity of the tropopause**
This section is significantly modified. In general terms, the introducing paragraph to this section reads now as follows:

"In order to study the formation history of ISSR and involved processes, we analysed the occurrence frequency and physico-chemical signature of ISSR around the tropopause layer and referred our analyses to both the thermal and the dynamical tropopause. We want to recall the tropopause definitions given in Section 2.1. The thermal tropopause according to WMO criteria (WMO, 1957) is usually seen as an effective transport barrier hampering troposphere-stratosphere exchange, whereas the dynamical tropopause is commonly used for separating tropospheric and stratospheric air masses in studies on stratosphere–troposphere transport since it represents the lower bound of the ExTL. These complementary views on the tropopause have been developed from extensive CO - $O_3$ analyses, which showed that the 2 PVU surface approximately separates the troposphere from the stratosphere with the ExTL as a transition layer of about 2 km thickness above it and centred on the thermal tropopause(Hoor et al., 2004; Pan et al., 2010; Gettelman et al., 2011). These tracer studies in the extratropics showed that on average the dynamical tropopause is situated slightly

below the thermal tropopause and the gradients of CO and $O_3$ are much sharper across the thermal tropopause compared to the dynamical tropopause (Hoor et al., 2004; Pan et al., 2010). Similar features are observed for the gradients of temperature T, $H_2O$ VMR and $O_3$ VMR, shown in Figure 9 for the North Atlantic region. Similar to the tracer gradients, also the temperature gradient is sharper across the thermal tropopause compared to the dynamical tropopause. In addition, the results confirm the good agreement between the ERA-Interim thermal tropopause height indicated by $\Delta p_{TPH}$ = 0 hPa (blue lines), the lowest temperatures detected at $\Delta p_{TPH}$ = 0 hPa, and the chemical tropopause, indicated by $O_3$ VMR = 120 ppbv at $\Delta p_{TPH}$ = 0 hPa, and thus the consistency of the used data set. Furthermore, the analysis of the pressure difference between the thermal and dynamical tropopauses reveal an offset of approx. 25 hPa (15 - 35 hPa) which translates into an altitude difference of approx. 1 km (Neis, 2017).

[Figure]

**Figure 9.** Vertical distribution of temperature T (a), $H_2O$ VMR (b), and ozone VMR (c) relative to the 2 PVU dynamical tropopause and to the thermal tropopause; vertical distributions relative to the thermal tropopause are presented as percentiles [1, 25, 50, 75, and 99] by blue lines and relative to the 2 PVU tropopause conditions by red-shaded areas. "

The concluding paragraph reads now as follows:

"Recalling the structure of the ExTL with the 2 PVU dynamical tropopause at its lower bound separating the stratosphere from the troposphere, and centred on the thermal tropopause, we find that on the top of the ExTL non-ISSR air masses show a clear stratospheric signature, while ISSR air masses are still strongly influenced by mixing and carry a significant tropospheric fingerprint compared to the non-ISSR air masses. Above the dynamical tropopause and thus inside the ExTL, the influence of mixing increases gradually for both ISSR and non-ISSR air masses and the difference in troposphericity is much less pronounced than near the top of the ExTL."

Detailed responses to reviewers concerns are given in the specific responses.

**Response to Reviewer #2**

**Reviewer:** *This paper presents an analysis of data of relative humidity for the period 1995 to 2010, obtained via instrumented passenger aircraft in the framework of IAGOS and MOZAIC over the northern mid-latitudes (40-60N) in 3 longitude ranges: Northeast America, North Atlantic, and Europe. The huge amount of data makes it possible to cover several vertical altitude ranges of 30 hPa thickness with sufficient data density to allow robust statistics. The altitude bands are defined with respect to the thermal and the dynamical tropopause, respectively, and the "tropospheⅼricity" (i.e. the fraction of tropospheric origin in an air parcel) is determined using simultaneous data of ozone VMR. The focus of the study is ice supersaturation.*

*The data show, that ISSRs (ice supersaturated regions) occur most often directly below the thermal tropopause, rarely directly above it, and almost never further up in the stratosphere. There is a distinct seasonal cycle in all 3 considered regions, but no significant trend over the 15 years of the study. The North-Atlantic Oscillation seems to have an influence on the occurrence of ISSR over the North Atlantic and Europe, but not over North America, which is physically plausible. ISSRs are colder and moister than their subsaturated surroundings (in agreement with earlier results), and they are poorer in ozone and have accordingly a larger tropospheⅼricity than the subsaturated environments, which is plausible as well considering the fact that most ice supersaturation is formed by uplifting of airmasses. The data show also that ice supersaturation is very closely related to cloudiness, that is, most ice supersaturation is found within clouds.*

*Thus, this paper provides a number of new and interesting results. It is well written for the most part. There are only a few points where I think the presentation can be made clearer and where perhaps the discussion can consider one or two more points. The paper should surely be published after the issues below are addressed.*

**Authors:** We thank the reviewer for the positive feedback and the valuable review and respond in detail to your comments in the following.

**MAJOR COMMENTS**

**Reviewer:** *The paragraph lines 388 to 395 should be reworked; it is unclear what you did. For instance, what is an "average occurrence probability"? Do you mean the average frequency of occurrence or something else? What is the pdf of ISSR occurrence? Is this simply the probability of ISSR occurrence? I also do not understand what the distinction between seasons has to do with statistical quantities like median and percentiles and how these two non-related things are linked here in one sentence. And finally, what is the statistical entity?*

**Authors:** We agree that the paragraph is confusing. In fact, "average occurrence probability" refers to the average frequency of occurrence, as supposed by the reviewer. Also, "PDF of ISSR occurrence" refers to the probability of ISSR occurrence. The entire confusing explanation why we focused in this part of our analysis on the entire data set and skipped distinctions into seasons is removed.

Summarising, we rephrased the paragraph:

"Our analysis of ISSR occurrence in the vicinity of the exTL is confined to the North Atlantic region, for which we have the highest data density available with respect to vertical resolution. As described generally in Section 2.1, the entire data set of individual $RH_{ice}$ observations over the North Atlantic region was divided into yearly subsets for seasons DJF, MAM, JJA, and SON. For each year, season and altitude layer relative to the thermal and dynamical tropopauses, the average frequency of occurrence of observations with $RH_{ice}$ > 100% was determined. The probability of ISSR occurrence per altitude layer with respect to the entire period of 15 years was then calculated from this record of seasonally averaged ISSR frequencies of occurrence. The results are compiled in Table 3 for both tropopause definitions used here. Please note that the ISSR fractions compiled for the thermal tropopause correspond to the values listed in Table 2, but without distinction for seasons."

**Reviewer:** *The comparison between statistics relative to the thermal and the dynamic tropopauses is not easy to understand, perhaps because it is unclear what exactly has been done. The first issue that must be clarified is whether the tropopause pressure and the pressure of the 2 PVU surface are available for each single measurement or are there only average values available, which would be bad for the analysis. What is the average $\Delta p$ with respect to (wrt) the thermal tropopause of the 2 PVU surface? It seems that 2 PVU occurs quite often or in the majority of cases in the UT1 layer. This should be stated. However, it does not seem as if the mean profiles wrt to the dynamical TP are just shifted versions of the profiles wrt the thermal TP definition. Is this a consequence of averaging or why is this so? Next, Table 3 lists under Thermal TP numbers which I expected to be annual mean values of numbers in Table 1 under AVG, but a quick calculation shows something different. Is this because of different weights for the seasons or what is the reason? (For instance take the 20:0 ± 6:5 in column 3 of Table 3. Should this not be the mean of 21:3, 18:6, 17:1 and 18:4 in the right hand box AVG in table 1?). And finally, Fig. 10 shows different behaviour in the left and right panels. Although you give a good physical explanation, I am not fully convinced. In the thermal TP coordinates there is a strong difference between ISSR and non-ISSR already at 30 hPa above the TP, but in the dynamical TP version there is only a much smaller difference at 60 hPa above the 2 PVU level. Is it possible that, on average, the 2 PVU surface is more than 60 hPa below the thermal TP? Eventually we should expect to see qualitatively the same profiles, irrespective of the actual choice of a vertical coordinate, isn't it?*

**Authors:** As explained in the GENERAL REMARKS in response to all reviewers, we put now the entire discussion of the observations and their interpretation into the framework of the extratropical transition layer (ExTL) and the vertical tracer profiles observed in that region. See details hereto in the response to all reviewers.

We will now focus on specific responses to the Reviewer's concerns:

*1. The comparison between statistics relative to the thermal and the dynamic tropopauses is not easy to understand, perhaps because it is unclear what exactly has been done.*
**Authors:** In response to this criticism we introduced a description of the methodology into Section 2.1; see the response to all reviewers above.

*2. It seems that 2 PVU occurs quite often or in the majority of cases in the UT1 layer. This should be stated. However, it does not seem as if the mean profiles wrt to the dynamical TP are just shifted versions of the profiles wrt the thermal TP definition. Is this a consequence of averaging or why is this so?*
**Authors:** The vertical distributions of properties with respect to the two tropopause definitions were determined independently and cannot be simply transferred into each other. The explanation is that the vertical spacing by $\Delta p$ = 30 hPa starts from two different pressure levels $p_{therm.TPH}$ and $p_{dyn.TPH}$ which are not connected by a straightforward relationship. Over the North Atlantic we found an offset of approx. 25 hPa between dynamical and thermal tropopause with variations from 15 to 35 hPa.
A sentence stating this result was included in section 3.3, reading: "Furthermore, the analysis of the pressure difference between the thermal and dynamical tropopauses reveal an offset of approx. 25 hPa (15 - 35 hPa) which translates into an altitude difference of approx. 1 km (Neis, 2017)."

*3. Table 3 lists under Thermal TP numbers which I expected to be annual mean values of numbers in Table 1 under AVG, but a quick calculation shows something different. Is this because of different weights for the seasons or what is the reason? (For instance take the 20:0 ± 6:5 in column 3 of Table 3. Should this not be the mean of 21:3, 18:6, 17:1 and 18:4 in the right hand box AVG in Table 1?).*
**Authors:** The values listed in Table 3 are calculated from the entire data set and not from seasonally averaged values to get the full variability of observations. However, differences to the mean values calculated from Table 1 seasonal means are small:

The value of 20.0 ± 6.5% from column 3 in Table 3 corresponds to the mean of 22.4, 20.1, 20.8 and 20.4 which is 20.9. It has to be noted that only values from Table 1 for the North Atlantic region should be used. The columns of Table 3 named AVG refer to the seasonal means over all three regions. This is clarified by modifying the column description in Table 3 to "AVG(ENA, NAtl, EU).

*4. Fig. 10 shows different behaviour in the left and right panels. Although you give a good physical explanation, I am not fully convinced. In the thermal TP coordinates there is a strong difference between ISSR and non-ISSR already at 30 hPa above the TP, but in the dynamical TP version there is only a much smaller difference at 60 hPa above the 2 PVU level. Is it possible that, on average, the 2 PVU surface is more than 60 hPa below the thermal TP? Eventually we should expect to see qualitatively the same profiles, irrespective of the actual choice of a vertical coordinate, isn't it?*
**Authors:** As explained before, the vertical distributions of properties with respect to the two tropopause definitions were determined independently and cannot be simply transferred into each other. In addition, we found an average pressure difference between the thermal and dynamical tropopauses of approx. 25 hPa (15 - 35 hPa).
As said above, the 2 PVU surface approximately separates the troposphere from the stratosphere with the ExTL as a transition layer of about 2 km thickness above it and centred on the thermal tropopause. Hence, the two tropopauses describe completely different positions within the ExTL so that we cannot expect similar profiles relative to the two tropopause definitions since different processes come into play.

**Minor Issues**

**Reviewer:** *Occasionally the term UTH is used. This should be avoided. UTH is a radiance based measure of a kind of mean relative humidity in a thick layer in the upper troposphere; it is a non-local measure. In contrast, IAGOS and MOZAIC yield local measures of relative humidity, and even after averaging over certain layers they should not be called UTH to avoid confusion. Better call it "the relative humidity field of the UT" or similar, but avoid UTH.*
**Authors:** As recommended, we replaced the term UTH throughout the manuscript by RH or similar.

**Reviewer:** *The last sentence of the introduction should be changed. The middle atmosphere is hardly relevant for IAGOS.*
**Authors:** This was a typo, we simply removed the reference to the middle atmosphere. The sentence reads now: "Chemistry-climate models like L90MA and L47MA use a vertical grid spacing of 15 - 25 hPa near the extratropical tropopause (Jöckel et al., 2016) which is reflected in the selected vertical resolution of MOZAIC data layers."

**Reviewer:** *Figure caption 1: I do not understand what you mean with the pdf of data points. Do you mean simply the number of measurements or the fraction of measurements in a certain grid box?*
**Authors:** We mean the fraction of measurements in a certain grid box. The modified figure caption is: "**Figure 1.** Global coverage of water vapour observations by MOZAIC for the period 1995 to 2010, shown as decadal logarithm of the probability distribution function (PDF) of the data points (fraction of measurements in a certain grid box); red boxes indicate the target areas for our analyses."

**Reviewer:** *Line 251: "Figure 4 illustrates ... of RH...": is this with or without IFC applied?*
**Authors:** We specified the data set as "Distribution of $RH_{ice}$ for the entire MOZAIC period from 1995 to 2010 with IFC applied as a function of ambient temperature with the colour indicating the probability of occurrence …"

**Reviewer:** *Figure 4: Please describe how these data are normalised. Is the sum over the whole figure 1?*

**Authors:** We specified: "Figure 4 illustrates the distribution of RH$_{ice}$ observations from the entire MOZAIC data set shown in Figure 1 as a function of ambient temperature, colour-coded by the probability of occurrence, i.e. the fraction of data points for a specific combination of temperature and RHice with the respect to the entire ensemble."

**Reviewer:** *I am a bit puzzled by the kind of averages applied. In line 176 it says " data are consolidated to 3-months season files", but in line 292 we have monthly mean profiles. Furthermore, is the distance of the current pressure level of a single 4-sec data point to the tropopause pressure recorded for every data point, or is the tropopause pressure averaged over a month and this average taken as reference (which would be a bad strategy to my view)?*
**Reviewer:** *Figures 6 and 7: why do you use geometrical height instead of the $\Delta p$ for these figures?*
**Authors:** The calculation of monthly mean profiles and the use of km as vertical spacing was done exclusively for producing Figs. 6 and 7 which can be compared directly to respective figures from Zahn et al. (2014). All statistical analyses have been performed using 3-months season files. The pressure distance of the aircraft position to the thermal and dynamical tropopause is determined for each single data point; the procedure is described in Section 2.1.

**Reviewer:** *Line 373/4: The statement may be wrong or perhaps right for the wrong reason. If the mean value of a positive quantity gets small, the variability usually gets smaller as well. Thus I suggest you to consider instead of $\sigma$ the normalised $\sigma$: $\sigma/\mu$ (i.e. std. deviation divided by mean value).*
**Authors:** Good point! We replaced the standard deviations in Table 2 by the normalised standard deviations and rephrased as follows: "Focussing on the UT layers, the relative standard deviations of the ISSR fractions are highest for the lowest layer investigated here, at least for winter and spring seasons for which the largest ISSR fractions are found."

**Reviewer:** *Comparison with RS Lindenberg (Figure 12a): has the same pressure band be selected for the RS data as for the MOZAIC/IAGOS data or are these the plain overall figures from the old publication?*
**Authors:** We did not reanalyse the Lindenberg data but refer to the original publication.

**Reviewer:** *It is not clear why CALIPSO can have higher cloud frequency than ISSR frequency. The argument that CALIPSO sees subvisible cirrus (SVC) explains only that it sees more than other satellite instruments do, unless SVC can survive in subsaturated air for a quite long time, where it is unclear to me what quite long actually means. I think that the reason for this result is rather in the difference of local vs. non-local measurements, just as the cloud fraction in a single level is always smaller than the cloud coverage over several levels.*
**Authors:** According to Stubenrauch et al. (2010), the high cloud amount (HCA) of CALIPSO is about 10% larger than HCA of CALIPSO for clouds excluding subvisible cirrus. Therefore, the difference can be attributed to instrument sensitivities. In order to explain more in-depth we rephrased the statement about CALIPSO to:
"According to Stubenrauch et al. (2010), the high cloud fraction of CALIPSO is about 10% larger than respective values of CALIPSO for clouds excluding subvisible cirrus. Therefore, the difference between high cloud fractions from CALIPSO and from the other instruments shown in Figure 13 can be attributed to instrument sensitivities."

**Reviewer:** *Final paragraph of 3.5: Misuse of "cross-correlation". A cross-correlation is simply a correlation between two different quantities (as opposed to auto-correlation). Furthermore I suggest to replace "probability" in this paragraph with "fraction" in order to avoid wrong connotations. I am also a bit unhappy with "correlation" since I do not see that the two time-series have been correlated (in this case indeed cross-correlated) which would easily be possible. In this case there are also standard techniques to evaluate the statistical significance of the result (i.e. whether the correlation*

*coefficient is significantly different from zero). The sentence "we consider ... statistically significant" should be deleted. This is not a question of "consideration" but of calculation. However, the physical explanation for your result is plausible. In the same sense, the statement in line 645 "significant correlations..." should be reformulated, for instance "physically plausible influence of the NAO on ISSR occurrence is detected in the time series...).*

**Authors:** The entire section was modified in response to a comment by Reviewer #1. We performed a cross-correlation analysis for all three regions and show the results in the modified Figure 16 (formerly 15). This cross-correlation analysis reported a statistically significant correlation between NAO index and deviation of ISSR occurrence from the long term average for the North Atlantic, but no significant correlations for the other two regions. The description of the figure and the conclusions drawn read now:

"For the regions Eastern North America and Europe the correlation between NAO index and $\Delta$ ISSR fraction is not statistically significant. For the North Atlantic however, the results of the cross-correlation analysis indicate statistical significance at a level of 99%.The obtained correlation of signs is in line with the observation that the occurrence of ice-supersaturation is well correlated with the storm track activity (Spichtinger et al., 2003b; Gettelman et al., 2006; Lamquin et al., 2012)."

[Figure]

**Figure 16.** Correlation analysis with respect to the correlation of signs between NAO index and deviation of ISSR occurrence from the long-term average ($\Delta$ ISSR) for the target regions; numbers indicate he results from the correlation analysis with respect to number of samples n, Pearson R and significance level p.

**Reviewer:** *The data show that "by far the largest part of ISSR occurs inside cirrus clouds". You should ask yourself: Why? Doesn't this imply that most ISSR reach the humidity threshold for heterogeneous or even homogeneous freezing shortly after the air mass began to be supersaturated? Are there further implications? Since the NH has more heterogeneous IN than the SH, do you expect that on the SH a smaller fraction of ISSR is inside clouds?*

**Authors:** In response to a comment by Reviewer #1 we added the following paragraph to the introduction to Section 3.4:

"Furthermore, the analysis of a large set of combined observation of $RH_{ice}$ and ice crystal number concentration $N_{ice}$ during a series of research flights (approx. 68000 observations of ice-supersaturation; Krämer et al., 2016) demonstrated, that approx. 80 % of the observed ice-supersaturation events are associated with in-cloud conditions. On the other hand, $RH_{ice}$ probability distribution functions inside cirrus clouds are characterised by most probable values at or slightly above ice-saturation at $RH_{ice}$ = 100% (Krämer et al., 2009; Diao et al., 2014; Diao et al., 2015; Petzold et al., 2017) which means that cirrus clouds exist to a considerable fraction also in ice-subsaturated air masses, depending on their state of life."

The concluding paragraph of Section 3.4 was significantly softened and reads now:
"The good agreement between MOZAIC in-situ observations of ISSR occurrence with the high-cloud fraction from satellite instruments encourages further detailed studies on this matter. First exemplary analyses of simultaneous observations of $RH_{ice}$ and $N_{ice}$ which are now possible within the ongoing IAGOS programme already indicate a strong correlation of high $RH_{ice}$ values with its occurrence inside cirrus clouds (Petzold et al., 2017). "

**Language, typos, etc.**
**Reviewer:** *Line 71: remove comma after supersaturation.*
**Authors:** Done.

**Reviewer:** *Lines 227/8: Details of ... in detail .. Please reformulate.*
**Authors:** Done, removed … in detail …

**Reviewer:** *Line 275: Please replace "validation" with "comparison". And then "The MOZAIC ... IS plotted..."*
**Authors:** Done.

**Reviewer:** *Lines 327 and 329: change to "north" and "south" (i.e. use lower case).*
**Authors:** Done.

**Reviewer:** *Line 340: I suggest to write "Similar AVERAGE values of ..."*
**Authors:** Done.

**Reviewer:** *Line 353: delete "set".*
**Authors:** We replaced the sentence by "…and in the last set of columns averaged over all regions" since we refer here not only to one column but to a block of four columns.

**Reviewer:** *Line 491: add comma after dynamics.*
**Authors:** Done.

**Reviewer:** *Line 579: thus HAS positive ...*
**Authors:** Modified to "…thus show positive …"

**Reviewer:** *Figure 15: in my printout there is no grey shading.*
**Authors:** Figure has been replaced, see annotated manuscript.

**Reviewer:** *Line 596: warMer and moister (or do you indeed mean more, i.e. a larger quantity of moist air?)*
**Authors:** The phrase "which brings warmer and more moist air to Europe" is removed.

**Reviewer:** Line 613: MOZAIC (with I).
**Authors:** Done

**Added References**

[revised manuscript text omitted]
 <s>average</s> mean ISSR occurrence probability is <s>29</s>31% (38%) in the upper troposphere <s>and increases to 34% when approaching</s>below the tropopause layer. <s>With reference to the dynamical tropopause, the overall behaviour is similar with an</s> The observed increases <s>increasing average</s> of mean ISSR occurrence probabilities <s>y</s> <s>when reaching</s>towards the tropopause layer are below statistical significance, <s>but the absolute values are larger since the analysed layers reach deeper into the upper troposphere.</s> For both tropopause definitions, the <s>variability</s> standard deviation of observed ISSR fractions is largest for the lowest UT layer of the analysed atmospheric region and decreases with increasing altitude.

520

525 **Table 3.** Mean and standard deviation of seasonal fraction of ice supersaturated regions (ISSR) for the seven vertical layers distributed around the thermal and dynamical tropopause.

| Layer ID | p − p$_{TPH}$ [hPa] | ISSR fraction [%] |
| --- | --- | --- |

|  |  | Dynamical TP | Thermal TP |
|---|---|---|---|
| LMS3 | - 90 | 0.2±0.5 | 0.0±0.1 |
| LMS2 | - 60 | 0.7±1.1 | 0.1±0.3 |
| LMS1 | - 30 | 8.4±4.4 | 1.5±1.1 |
| TPL | 0 | 30.7±9.4 | 20.0±6.5 |
| UT1 | 30 | 39.9±10.0 | 33.9±9.0 |
| UT2 | 60 | 37.7±10.7 | 31.4±9.2 |
| UT3 | 90 | 35.5±14.3 | 29.1±12.1 |

When crossing the thermal tropopause, the ISSR fraction drops sharply to values of 1.5% for the lowest layer above the thermal tropopause and to statistically insignificant fractions when reaching further up into the stratosphere. In case of the dynamical tropopause, we find a significantly higher ISSR fraction of 8.4% for the lowest stratosphere layer, and again insignificant fractions further above. This strong contrast in the ISSR occurrence probability for the lowest stratosphere layers with reference to the two tropopause definitions is caused by the different physical natures of the thermal and dynamic tropopauses. coincides with the behaviour of other tracers in the ExTL; see Figure 9 for details.

While the thermal tropopause forms a robust barrier for the vertical transport of water vapour, the dynamic tropopause serves as the lower bound for an atmospheric layer characterised by dynamically driven mixing processes . As a consequence, we expect different chemical signatures for the ISSR above the thermal and dynamical tropopauses.

In order to learn more about the history of ice-supersaturated air parcels we further analysed the ozone content of the ISSR compared to the sub-saturated air around, for air parcels below and above the thermal and dynamic tropopauses and combined the results with the distributions of temperature and $H_2O$ VMR. The thermodynamic and chemical properties of ISSR and the comparison between ISSR (blue lines) and ice-subsaturated air masses (red-shaded areas and red lines) are presented in Figure 10 with reference to both tropopause definitions. In general, ISSR are colder than their subsaturated counterparts. The difference is low in the UT with 1 - 2 K which compares well to the value of 2 K at 215 hPa obtained from MLS satellite measurements (Spichtinger et al., 2003b), and increases to more than 6 K difference in the stratosphere above the thermal tropopause, and approx. 4 K above the dynamical tropopause. The temperature difference of 3 - 4 K between colder tropospheric ISSR and the surrounding subsaturated air masses reported by Gierens et al. (1999) is comparable to the temperature difference in the 30 hPa thick tropopause layer we find in our analysis.

Figure 10 also indicates a similar behaviour of the vertical distribution of $H_2O$ VMR for ice-supersaturated and ice-subsaturated regions with exponentially decreasing absolute humidity up to the tropopause layer. Above both tropopause layers, $H_2O$ VMRit further decreases for thein case of non-ISSR conditions. For ISSR conditions, however, $H_2O$ VMR, whereas the water vapour VMR remains constant with height at the tropopause layer value of about 55 ppmv in the case of ISSR throughout the layer just above the tropopause. The increaseDoubling of $H_2O$ VMR in for the 
[revised manuscript text omitted]

---

## Author Comment (AC3) · 1 Mar 2020

**Response to Reviewers**

**GENERAL REMARKS**

We thank all three reviewers for their insightful reviews and helpful and constructive comments. Responding to their comments helped improve the manuscript significantly.

All reviewers raised the point that the discussion of the different tropopauses (thermal, dynamical) lacks clarity and is confusing, and also partially misleading. As a general response to all reviewers, we put now the entire discussion of the observations and their interpretation into the framework of the extratropical transition layer (ExTL) and the vertical tracer profiles observed in that region. As discussed e.g. by Hoor et al. (2004), Pan et al. (2010), and Gettelman et al. (2011), it is found that on average the dynamical tropopause is situated slightly below the thermal tropopause and trace gas gradients are more sharp above the thermal tropopause compared to the dynamical tropopause. This is exactly the same behaviour we observe here and the interpretation of our results is now linked to that known feature of tracer characteristics in the ExTL. So, we modified the following sections considerably:

**1. Introduction:**
We added the following paragraphs to the introduction:

[revised manuscript text omitted]

> LMS : $p < p_{therm.TPH}$ -15hPa; which is limited by the maximum cruise altitude with $p \approx 190$ hPa;
> TPL : $p = p_{therm.TPH}$ ± 15hPa;
> UT : $p > p_{therm.TPH}$ + 15hPa; limited to lower altitudes by $p < 350$hPa."

**3.3 Physico-chemical signature of ice-supersaturated regions in the vicinity of the tropopause**
This section is significantly modified. In general terms, the introducing paragraph to this section reads now as follows:

"In order to study the formation history of ISSR and involved processes, we analysed the occurrence frequency and physico-chemical signature of ISSR around the tropopause layer and referred our analyses to both the thermal and the dynamical tropopause. We want to recall the tropopause definitions given in Section 2.1. The thermal tropopause according to WMO criteria (WMO, 1957) is usually seen as an effective transport barrier hampering troposphere-stratosphere exchange, whereas the dynamical tropopause is commonly used for separating tropospheric and stratospheric air masses in studies on stratosphere–troposphere transport since it represents the lower bound of the ExTL. These complementary views on the tropopause have been developed from extensive CO - $O_3$ analyses, which showed that the 2 PVU surface approximately separates the troposphere from the stratosphere with the ExTL as a transition layer of about 2 km thickness above it and centred on the thermal tropopause(Hoor et al., 2004; Pan et al., 2010; Gettelman et al., 2011). These tracer studies in the extratropics showed that on average the dynamical tropopause is situated slightly

below the thermal tropopause and the gradients of CO and $O_3$ are much sharper across the thermal tropopause compared to the dynamical tropopause (Hoor et al., 2004; Pan et al., 2010). Similar features are observed for the gradients of temperature T, $H_2O$ VMR and $O_3$ VMR, shown in Figure 9 for the North Atlantic region. Similar to the tracer gradients, also the temperature gradient is sharper across the thermal tropopause compared to the dynamical tropopause. In addition, the results confirm the good agreement between the ERA-Interim thermal tropopause height indicated by $\Delta p_{TPH}$ = 0 hPa (blue lines), the lowest temperatures detected at $\Delta p_{TPH}$ = 0 hPa, and the chemical tropopause, indicated by $O_3$ VMR = 120 ppbv at $\Delta p_{TPH}$ = 0 hPa, and thus the consistency of the used data set. Furthermore, the analysis of the pressure difference between the thermal and dynamical tropopauses reveal an offset of approx. 25 hPa (15 - 35 hPa) which translates into an altitude difference of approx. 1 km (Neis, 2017).

[Figure]

**Figure 9.** Vertical distribution of temperature T (a), $H_2O$ VMR (b), and ozone VMR (c) relative to the 2 PVU dynamical tropopause and to the thermal tropopause; vertical distributions relative to the thermal tropopause are presented as percentiles [1, 25, 50, 75, and 99] by blue lines and relative to the 2 PVU tropopause conditions by red-shaded areas. "

The concluding paragraph reads now as follows:

"Recalling the structure of the ExTL with the 2 PVU dynamical tropopause at its lower bound separating the stratosphere from the troposphere, and centred on the thermal tropopause, we find that on the top of the ExTL non-ISSR air masses show a clear stratospheric signature, while ISSR air masses are still strongly influenced by mixing and carry a significant tropospheric fingerprint compared to the non-ISSR air masses. Above the dynamical tropopause and thus inside the ExTL, the influence of mixing increases gradually for both ISSR and non-ISSR air masses and the difference in troposphericity is much less pronounced than near the top of the ExTL."

Detailed responses to reviewers concerns are given in the specific responses.

**Response to Reviewer #3**

**Reviewer:** This manuscript describes in situ measurements of relative humidity (RH) in the upper troposphere and lower stratosphere (UTLS) from commercial aircraft and presents a detailed statistical examination of ice-supersaturated air masses (RHice >100%). The analysis is confined to a region of high measurement density between latitudes 40°N- 60°N and longitudes 105°W and 30°E, for the years 1995-2010. Several conclusions are drawn regarding the probabilities of encountering regions of ice supersaturation (ISSR) in three different longitude regimes, based on distance from the tropopause and season. There is also a minor attempt to attribute interannual variations in these probabilities to the North Atlantic Oscillation (NAO).

**MAJOR COMMENTS**

**Reviewer:** *Uncertainties are calculated and presented (typically 1 standard deviation) for most mean values derived in this paper. However, the uncertainties are often ignored when interpreting the mean values and making quantitative statements about them. One example, in Lines 396-397: "... the average ISSR occurrence probability is 29% in the troposphere and increases to 34% when approaching the tropopause layer." Given that the standard deviations of these mean values are each at least _9%, the two averages are not statistically different, and the claimed "increase" is not significantly different from zero. A second example is Figure 11, where a horizontal line (indicating no seasonality) can easily be drawn within the uncertainties in each panel that show "annual cycles". Therefore, the statement (Line 506), "For all regions, ISSR occurrence probabilities are highest in the winter/spring and lowest in summer ..." is not supported by these seasonal averages with their statistical uncertainties. In view of this, why are most uncertainties in this paper calculated and presented as 1 standard deviation when the vast majority of scientific uncertainties are reported as 95% confidence intervals (i.e., approximately 2 standard deviations for large sample sizes)?*

**Authors:** The standard deviations reported from airborne observations include not only the variability of observations caused by the noise of the instrument, but also the natural variability of the atmosphere when sampling from a fast-moving platform, and the interannual variability when averaging over a period of 15 years. Hence, we added the following statement to Section 2.1: "In our study, we use statistical analyses in the following manner: when assessing results from laboratory studies and calibration experiment based on reproducible observations, we apply the 2-σ criterion for the 95% confidence level; when interpreting results from atmospheric observations which are taken from fast-flying airborne platforms and cover 15 years of observations, including their interannual and lateral variabilities, we report the mean values and respective 1-σ standard deviations and state statistical significance or insignificance, respectively."

In the light of this introductory statement, we rephrased the interpretation of ISSR occurrence frequency as: "With reference to the thermal (dynamical) tropopause, the mean ISSR occurrence probability is 31% (38%) in the upper troposphere below the tropopause layer. The observed increases of mean ISSR occurrence probabilities towards the tropopause layer are below statistical significance. For both tropopause definitions, the standard deviation of observed ISSR fractions is largest for the lowest UT layer of the analysed atmospheric region and decreases with increasing altitude."

For the same reason, we did not modify Figures 12 and 13 (formerly 11 and 12) but report the mean values including their standard deviations.

**Reviewer:** *The "occurrence probability" statistics are simple to understand, based on the numbers of RH measurements reflecting subsaturation, saturation, or supersaturation during a flight segment, an entire flight, or a number of flights. But it is not clear how "occurrence probability standard deviation" statistics were calculated. Are these based on calculating an average of the occurrence probabilities for a number of flights, reflecting the variability of the occurrence probabilities for individual flights*

*around the average? This should be briefly explained, early in the paper, so the reader can immediately grasp the concept of the "occurrence probability standard deviation".*

**Authors:** We agree, that the procedure of calculating mean values and standard deviations from the data set is not sufficiently explained. Hence, we included a description in Section 2.1, reading: "Since each data set from one single flight provides only a one-dimensional snapshot of the state of the atmosphere along the flight track, and each aircraft cruises at a slightly different pressure level, the entire MOZAIC data are consolidated to season files of 3-months duration, allowing the analysis of vertical distributions of atmospheric state parameters on a robust statistical basis. For each season file, the statistical distribution (average and standard deviation, median and percentiles) of investigated properties (temperature, $O_3$ VMR, $H_2O$ VMR, $RH_{ice}$, ISSR fraction) is calculated with respect to the above defined UT, TP and LMS vertical layers. From these seasonal averages or percentiles, respective 15-year mean values and standard deviations are determined."

**Reviewer:** *Why is the requirement for supersaturation RHice >100% when the measurement uncertainties are approximately 5% RH in the middle and upper troposphere? If some part of these uncertainties is a systematic error (a high bias of 3%, for example), wouldn't this lead to artificially high occurrence probabilities if measurements of a 98% RH air mass are 101% RH? How much do the occurrence probabilities decrease if you instead require RHice >103%, or even RHice >105% for supersaturation?*

**Authors:** As stated in Section 2.2, the MOZAIC capacitive humidity sensor MCH was extensively tested against research-grade instruments on the same platform, avoiding biases from sampling different air masses(Neis et al., 2015a; Neis et al., 2015b). The intercomparison study proved a zero offset with respect to $RH_{ice}$ data with a reported offset of -0.15 ± 1.1.29 % $RH_L$ (Neis et al. 2015a). From these results we conclude that we don't see a systematic bias in the $RH_{ice}$ data reported from the MCH instrument. Hence, we assume a random scatter of the $RH_{ice}$ data around the zero value of $RH_{ice}$ = 100% which justifies the assumption of $RH_{ice}$ = 100% as threshold for ice-supersaturation.

**Reviewer:** *I'm not convinced that the comparison of supersaturation occurrence probabilities for atmospheric layers relative to the lapse rate ("thermal") tropopause vs the 2 PVU ("dynamical") tropopause shows much of a difference. If 95% confidence intervals of the mean values in Table 3 are considered, none of the "thermal" and "dynamical" averages for any atmospheric layer are statistically different. A lot of text, Figures and Tables are devoted to this comparison, and what does it show? Very little, in my opinion. Instead (or in addition), I'd prefer to see some assessment of the accuracy of the ERA-Interim tropopause heights that are absolutely critical to this paper. Since ozone mixing ratios were also measured as part of MOZAIC, and ozone can be used to define a "chemical" tropopause, can you compare ozone-defined tropopauses to the ERA-Interim tropopauses to evaluate at least the consistency of the latter? For example, if ERA-Interim puts the tropopause 1 km above the aircraft and the ozone mixing ratio is 1 ppm that indicates a large (>1 km) error in the tropopause height. I'm not suggesting a full-scale comparison, but rather some comparisons that illustrate the possible errors in tropopause heights.*

**Authors:** As we argue in the explanation of the values listed in Table 3, the ISSR fractions relative to the thermal and the dynamical tropopauses do not differ significantly for the analysed layers, with the exception of the layer LMS1. For the LMS 1 layer the mean ISSR fraction wrt the dynamical tropopause is 3.5 times the mean value wrt to the thermal tropopause. For the other layers, the difference is 1.5 or less. We are confident that our analysis presented in Section 3.3 contributes new insights to science.

Concerning the request for a detailed assessment of ERA-Interim tropopause heights, we agree that this is a valuable study. However, related work has been recently published by Reutter et al. (2020) in close collaboration with the MOZAIC/IAGOS team. Besides, we analysed the vertical distributions of temperature, $H_2O$ VMR and $O_3$ VMR over the North Atlantic. The new Figure 9 presents the results:

[Figure]

**Figure 9.** Vertical distribution of temperature T (a), $H_2O$ VMR (b), and $O_3$ VMR (c) relative to the 2 PVU dynamical tropopause and to the thermal tropopause; vertical distributions relative to the thermal tropopause are presented as percentiles [1, 25, 50, 75, and 99] by blue lines and relative to the 2 PVU tropopause conditions by red-shaded areas.

These results confirm the good agreement between the ERA-Interim thermal tropopause height indicated as $\Delta p_{TPH}$ = 0 hPa (blue lines), the lowest temperatures detected by MOZAIC at $\Delta p_{TPH}$ = 0 hPa, and the chemical tropopause, indicated by $O_3$ VMR = 120 ppbv at $\Delta p_{TPH}$ = 0 hPa. We consider these results as sufficient proof for the consistency of the used data set. A respective sentence was added to the manuscript: "In addition, the results confirm the good agreement between the ERA-Interim thermal tropopause height indicated by $\Delta p_{TPH}$ = 0 hPa (blue lines), the lowest temperatures detected at $\Delta p_{TPH}$ = 0 hPa, and the chemical tropopause, indicated by $O_3$ VMR = 120 ppbv at $\Delta p_{TPH}$ = 0 hPa, and thus the consistency of the used data set."

**Reviewer:** *Water vapor mixing ratios are discussed in some sections of the paper and are shown in some Figures, but nowhere in the paper is there a description of how these were determined. Were they measured directly with different instruments (as implied in Line 17 of the abstract) or were they calculated from the RH measurements, requiring concomitant measurements of pressure and temperature with their associated uncertainties?*
**Authors:** Indeed, it was not explicitly mentioned how the $H_2O$ VMR values have been obtained. We introduced on sentence in Section 2.1: "The $H_2O$ VMR was finally calculated from the simultaneously measured $RH_{ice}$ and temperature data and from the pressure recordings of the aircraft avionic system."
In order to avoid confusion, we also rephrased the abstract which reads now: "Observation data originate from regular and continuous long-term measurements on board of instrumented passenger aircraft in the framework of the European research program MOZAIC (1994 – 2010) which is continued as European research infrastructure IAGOS (from 2011). Data used in our study result from collocated observations of $O_3$, $RH_{ice}$ and temperature, and $H_2O$ VMR deduced from $RH_{ice}$ and temperature data."
The paragraph on the uncertainty of MCH water vapour measurements was also modified and reads now: "Kunz et al. (2008) who performed a statistical analysis of water vapour measurements from the SPURT campaigns between 2001 and 2003 by a Lyman-$\alpha$ photo-fragment fluorescence hygrometer (Zöger et al., 1999; Meyer et al., 2015) and MOZAIC water vapour data from the same period determined a limit of detection (LOD) of 10 ppmv for the MOZAIC sensor. Applying the same 2-$\sigma$ criterion (95% confidence level), we obtain a MCH limit of detection (LOD) of $RH_{ice,LOD}$ = 10% which again transfers into a minimum detectable $H_2O$ VMR of approx. 10 ppmv at typical mid-latitude upper troposphere conditions (T = 218K, p = 250 hPa); see also Neis et al. (2015a) for a detailed discussion. As is discussed by Smit et al. (2014), the uncertainty of the temperature measurement of the MCH sensor is included in the determination of the MCH $RH_{ice}$ uncertainty so

that the precision of $H_2O$ VMR data deduced from MCH $RH_{ice}$ data can be determined directly from the uncertainty of $RH_{ice}$ measurements. Overall, the 5% RH uncertainty leads to a decreasing precision of $H_2O$ VMR deeper in the stratosphere and implies a limited use of the MOZAIC $H_2O$ sensor in the stratosphere dominated by low $RH_{ice}$ and thus an increasing large uncertainty (Kunz et al., 2008)."

**Reviewer:** *There are some awkward and confusing sentences in the paper that could benefit from re-writing. I will point out a few of these below, but I suggest the paper be proofread by a native English speaker to clean up and clarify some sentences.*
**Authors:** A language check was performed and indicated sentences have been clarified.

**MINOR COMMENTS**
**Reviewer:** *Lines 23-25: This statement implies there is an increasing trend in summertime water vapor mixing ratios in the lowermost stratosphere, but no similar trend in RHice. I don't think this is what you mean to say, rather that mixing ratios in this region are highest during summer months without corresponding maxima in RHice. If this is the case, doesn't it imply that temperatures in this region are also highest during summer months?*
**Authors:** Indeed we observe highest temperatures in this region for the summer months as well, see Figure 8 of the discussion paper. Adopting your suggestion, we rewrote the sentence:
"Annual cycles of the investigated properties document highest $H_2O$ VMR and temperatures above the thermal tropopause in the summer months, whereas $RH_{ice}$ above the thermal tropopause remains almost constant in the course of the year

**Reviewer:** *L51: The term "tropopause layer" is used throughout this paper, but where is it geophysically defined? On page 5 you limit the TPL to "tropopause pressure _ 15 hPa", but that's a definition that is neither common or geophysically-based. It would enlighten the reader to know why you chose these limits for the TPL.*
**Authors:** As explained in the general response to all reviewers we rewrote the introduction and refer to the extratropical transition layer, see line 55ff. Here, the references to the literature about the tropopause region and its structure are given. Concerning the pressure width of 30 hPa for the atmospheric layers around the tropopause, we refer to the commonly used approach when analysing MOZAIC data (Thouret et al., 2006).

**Reviewer:** *L64: Why would an "increase in pressure" change the RH of an air mass? RH is the partial pressure of water vapor divided by the saturation pressure over ice at a given temperature. Neither of these is affected in any way by an "increase in pressure". Please either explain this statement more clearly or remove it.*
**Authors:** As explained in details in Spichtinger and Leschner (2016), relative humidity can be expressed as $RH_{ice} = p \, q / (\varepsilon \, p_{ice}(T))$, using ideal gas assumptions; thus, ambient air pressure p and relative humidity $RH_{ice}$ are linked by a linear relationship. However, the formation of ice-supersaturated regions is mostly driven by vertical uplift (i.e. decrease in temperature AND pressure), thus this possible effect is not probable and negligible.

We have rephrased the sentence as follows:
"Air masses supersaturated with respect to ice ($RH_{ice} > 100\%$), so called ice-supersaturated regions (ISSR), have mostly faced a decrease in temperature or increase in water vapour mixing ratio, i.e. specific humidity during their past lifetime (Spichtinger and Leschner, 2016). As a result, these air parcels are both colder and of higher relative humidity than the embedded sub-saturated atmosphere (Gierens et al., 1999; Spichtinger et al., 2003b) which did not experience similar changes in their atmospheric state parameters."

**Reviewer:** *L109-111: Why are radiosonde network measurements of RH "considered insufficient for detecting trends and variability in UTLS water vapor"? I believe the GRUAN radiosonde data product for RH will be sufficient in this regard, and that GRUAN represents another existing global-scale network of in situ observations of atmospheric composition in the Ex-UTLS. A good reference for GRUAN is:*
*Bodeker, G.E., Bojinski, S., Cimini, D., Dirksen, R., Haeffelin, M., Hannigan, J., Hurst, D., Leblanc, T., Madonna, F., Maturilli, M., Mikalsen, A., Philipona, R., Reale, T., Seidel, D., Tan, D., Thorne, P., Vömel, H., and Wang, J.: Reference Upper Air Observations for Climate: From Concept to Reality, B. Am. Meteorol. Soc., 97, 123–135, https://doi.org/10.1175/bams-d-14-00072.1, 2016.*

**Authors:** Thank you for the recommendation. We have rephrased the paragraph to clarify the argument and added a specific sentence on GRUAN. The paragraph reads now:

"Concerning in-situ observations of water vapour, the international radiosonde network of weather balloons is in operation for many decades but the observations are considered insufficient for detecting trends and variability in UTLS water vapour; see Müller et al. (2016) and references therein. The GCOS Reference Upper-Air Network (GRUAN) targets the provision of climate-quality measurements of tropospheric and lower stratospheric variables (Seidel et al., 2009). GRUAN has established rigorous data quality assessment measures to provide reference-quality in situ and ground-based remote sensing observations of upper-air essential climate variables and serves as another source of high-quality water vapour data, however, for a limited number of certified surface stations yet (Bodeker et al., 2016)."

**Reviewer***: L180: Please insert "attached" between "inlet" and "to"*
**Authors:** Done.

**Reviewer***: L184-189: How about the uncertainty of RH measurements in the lower stratosphere? LOD is one measure, but since you determine supersaturation occurrence probabilities for several layers above the tropopause this must be somewhat known.*
**Authors:** We have expanded the discussion and added uncertainty ranges for different altitudes. The paragraph reads now:

"The MCH reports RH data with an average uncertainty of 4% RH (span 1% RH to 6% RH) in the middle troposphere at 4 to 8 km altitude during ascent and descent, and 5% RH (span 2% RH to 8% RH) at the tropopause and lowermost stratosphere at 10 to 12 km cruising altitude (Smit et al., 2014)."

**Reviewer:** *L211: Please change "sequences" to "segments". "Sequences" is also awkward in L105.*
**Authors:** Done.

**Reviewer:** *Figure 3: I think you intend "w/t" to mean "without" in both panels. Please change to "w/o".*
**Authors:** Done

**Reviewer:** *L251-253: Presumably Figure 4 shows the RHice with the IFC applied, so please make this clear.*
**Authors:** The figure caption reads now: "Distribution of $RH_{ice}$ for the entire MOZAIC period from 1995 to 2010 with IFC applied as a function of ambient temperature with the colour indicating the probability of occurrence; the lines represent water saturation (solid line; Sonntag, 1994) and the threshold $RH_{ice}$ for homogeneous ice nucleation (dotted line; Koop et al., 2000; Kärcher and Lohmann, 2002)."

**Reviewer***: L286-287: "highest possible quality achievable by this kind of routine observations" sounds great, but what does it actually mean? This sounds like an advertisement instead of a scientific claim and I suggest toning it down or removing it.*

**Authors:** We want to express that all possible corrections and consideration of potential sources of uncertainties have been made. We rephrased it to: "In summary, this data set is now considered of highest possible quality achievable by the type of sensor applied and for the type of routine observations performed."

**Reviewer:** *L290-297, Figure 6: Up to this point, everything has focused on RH measurements and the tropopause-relative pressure bins you have defined. Here, the discussion suddenly turns to water vapor mixing ratios and tropopause-relative altitude bins. As above, where do the VMR data come from? And why does Figure 6 use altitude instead of pressure (or log pressure) as the vertical coordinate?*
**Authors:** The calculation of monthly mean profiles and the use of km as vertical spacing was done exclusively for producing Figs. 6 and 7 which can be compared directly to respective figures from Zahn et al. (2014). The main purpose is to illustrate the distinctly different vertical distributions of these two properties to the reader. The detailed information on the vertical distribution of related properties T, $H_2O$ VMW and $RH_{ice}$ is shown in Fig. 8.
The source of $H_2O$ VMR measurements is introduced in: "The $H_2O$ VMR was finally calculated from the simultaneously measured $RH_{ice}$ and temperature data and from the pressure recordings of the aircraft avionic system."

**Reviewer:** *L327: "are bounded to the Great Lakes area and further North". Given the size of Figure 1, it is difficult (without magnification) to find the Great Lakes. A better description would be "are within the northern half of the continental USA and southern half of Canada".*
**Authors:** Good suggestion, done.

**Reviewer:** *Figure 8, Tables 1 and 2: It is not clear what the tropopause-relative pressure boundaries are for the different layers. Are the average values plotted (Figure) and presented (Tables) at deltaP = -30 hPa for the layer bounded by TPpress-15hPa and TPpress- 45hPa? This should be clearly stated.*
**Authors:** We clarified the vertical spacing of the layers in section 2.1:
"In order to reach both a sufficiently large data set for robust statistical analyses and good vertical resolution, the Ex-UTLS is subdivided into seven layers of 30 hPa thickness each, with three layers located below the thermal tropopause height and three layers above. Thouret et al. (2006) used a similar definition, but referenced to the dynamical tropopause at 2 PVU, i.e. they defined the tropopause as a mixing zone 30 hPa thick across the 2 PVU potential vorticity surface.
The seven layers of 30 hPa thickness each are centred at $p_{therm.TPH}$ = 0 hPa for the tropopause layer (TPL) itself and then at $p_{therm.TPH} \pm 30$ hPa, $p_{therm.TPH} \pm 60$ hPa, and finally at $p_{therm.TPH} \pm 90$ hPa. From this vertical spacing, the separation of air masses is achieved by applying the following criteria (formulated for the thermal tropopause only):
LMS : $p < p_{therm.TPH}$ -15hPa; which is limited by the maximum cruise altitude with $p \approx 190$ hPa;
TPL : $p = p_{therm.TPH} \pm 15$hPa;
UT : $p > p_{therm.TPH} + 15$hPa; limited to lower altitudes by $p < 350$hPa.

**Reviewer:** *Given the standard deviations (Table 2), are the average values for different seasons or longitudinal regions (Table 1) statistically different at the 95% level of confidence?*
**Authors:** Following an argument of another reviewer, we replaced the standard deviations in Table 2 by the normalized standard deviations to make the values better comparable. Nevertheless, from a rigorous statistical point of view, the mean values for the different seasons are not significant at the 95% level of confidence. However, they reflect the interannual variability of the atmosphere since the seasonal means are calculated from 15 annual values. This interannual variability should not be neglected here.

**Reviewer:** *L373: Why is the annual cycle of UTH increasingly damped as you get closer to the tropopause?*

**Authors:** The sentence was removed.

**Reviewer:** *L377: As noted above, please explain how an increase in pressure can cause supersaturation.*
**Authors:** This statement has been removed.

**Reviewer:** *L429-433: This long sentence is confusing and requires re-wording for clarification.*
**Authors:** The section was divided into several sentences. It reads now:
"Above both tropopause layers, $H_2O$ VMR further decreases in case of non-ISSR conditions. For ISSR conditions, however, $H_2O$ VMR remains constant with height throughout the layer just above the tropopause. Doubling of $H_2O$ VMR for tropopause ISSR conditions compared to non-ISSR conditions is close to the results reported from MLS observations (Spichtinger et al., 2003b). In contrast, Gierens et al. (1999) found an increase of only 50% for $H_2O$ VMR inside ISSR compared to non-ISSR. In turn, this value compares well with our observations in the uppermost troposphere."

**Reviewer:** *Figure 9: What does the black horizontal line represent at +70 hPa in the H2O panels?*
**Authors:** This was an error in the plotting routine. In all panels the line $\Delta p = 0$ is highlighted as guideline to the eye for the tropopause layer.

**Reviewer:** *L454-455: The 33.5 ppb O3 value from ERA-Interim is representative of what altitude and region?*
**Authors:** Cirisan et al. (2013) use a value of 33.5 ppbv from ERA Interim air mass trajectory analyses as the tropospheric background ozone value in the upper troposphere in midlatitudes. The respective explanation was added.

**Reviewer:** *L460: Why the sudden switch from P99 to P95, without explanation?*
**Authors:** The values are taken from literature. We explain now as follows (line 570): "Note that P95 refers here to the 95 percentile value of the analysed data ensemble, as taken from Cohen et al. (2018)."

**Reviewer:** *Figure 10: Error bars for each marker would clearly show if the troposphericity values are statistically different (or not).*
**Authors:** The troposphericity values are determined from median and 99 percentile values of the entire ensemble. Hence, we cannot associate an uncertainty to the troposphericity parameters. Instead they describe a property of the entire ensemble.

**Reviewer:** *L500: Here in the text you claim that Figure 11 shows results for the "top UT layer", but the caption for Figure 11 says "calculations were conducted for the two UT layers positions closest to the tropopause."*
**Authors:** This was an error in the manuscript and has been corrected accordingly in the text while the figure caption is correct.
**Reviewer:** *L506-507: This statement is not supported by the average values when their uncertainties are considered.*
**Authors:** The statement was removed.

**Reviewer:** *L513: I assume the Lindenberg radiosonde RH data has been corrected using the GRUAN-recommended corrections? It is important to say this because the reader may assume that uncorrected RH data from radiosondes are good enough (they are not!). You might also reference the paper describing corrections to the Vaisala RS92 data:*
*Dirksen, R. J., Sommer, M., Immler, F. J., Hurst, D. F., Kivi, R., and Vömel, H.: Reference quality upper-air measurements: GRUAN data processing for the Vaisala RS92 radiosonde, Atmos. Meas. Tech., 7, 4463–4490, https://doi.org/10.5194/amt-7-4463- 2014, 2014.*

**Authors:** The Lindenberg data and the respective analysis were taken from the publication by Spichtinger et al. (2003). Data originate from corrected RS80A routine radiosonde observations which were calibrated against collocated RS90 ascents. Details of the applied data correction procedure are described in the publication.

**Reviewer:** *Figure 12: I don't see any information about the layer or layers for which the results are shown. Please state this in the caption.*
**Authors:** As in Figure 11, the analysis refers to the two UT layers positioned closest to the tropopause. A respective statement is added to the figure caption.

**Reviewer:** *L549: typo "15y ears"*
**Authors:** Done.

**Reviewer:** *L530: Why only 15 months of Lindenberg radiosonde data? There are more than 9 years of GRUAN-corrected Vaisala RS92 RH data from Lindenberg.*
**Authors:** As explained above, we used the data from literature, as published by Spichtinger et al. (2003). A detailed intercomparison between IAGOS and GRUAN $RH_{ice}$ observations is for sure highly valuable but beyond the scope of this manuscript.

**Reviewer:** *L551: "fits well" is an overstatement since the DJF and MAM averages for Lindenberg lie outside the MOZAIC mean _ 1 standard deviation envelope.*
**Authors:** The statement was revised to: "The 15-months cycle from the radio soundings is covered by the 15 years climatology of ISSR occurrence from MOZAIC, but contributes only a snapshot compared to the 15-years' time series."

**Reviewer:** *L549: I don't know what a "first exemplary analysis" is. Please explain.*
**Authors:** The term "exemplary" means that we conducted a first analysis from a shorter data set to demonstrate the possibilities of the combined $N_{ice} - RH_{ice}$ data set. We have now removed the term "exemplary" from the manuscript for clarity.

**Reviewer:** *L553: Trend analyses are performed on supersaturation occurrence probabilities based on which tropopause definition?*
**Authors:** We stated earlier in the manuscript that all analysis except the results presented in section 3.3 were conducted with respect to the thermal tropopause. For clarity we added a respective sentence to the manuscript: "The bases of our analyses were the seasonally averaged observations in the uppermost tropospheric layer (UT) with respect to the thermal tropopause, and the respective average seasonal cycles depicted in Figure 12."

**Reviewer:** *L579: "thus" must be a typo because it makes no sense in this sentence. Also, please change "long-term average values" to "long-term seasonal average values."*
**Authors:** Indeed there is a word missing. It reads now: "The de-seasonalised time-series thus show positive …". The term "seasonal" was inserted as suggested.

**Reviewer:** *L581-582: Three significant figures for trends and their uncertainties is not justified when the uncertainty values are nearly as large as the trends themselves. Why present the 1 standard deviation uncertainties when, presumably based on 2 standard deviation uncertainties, you claim in the next sentence that none of the trends are significant?*
**Authors:** To our understanding, one of the key messages of the manuscript is that we do NOT find trends in ISSR occurrence. In that respect we believe that it is justified showing the time series of ISSR fractions in Fig. 14 (formerly 13) and the de-seasonalised time series of ISSR fraction in Fig. 15 (formerly 14).

**Reviewer:** *L597: If the westerlies bring "warmer and more moist air to Europe", why would you expect a higher probability of supersaturation in the UT? More moisture increases the RH, but warmer air lowers the RH.*
**Authors:** The original wording was misleading. WE have rephrased the statement as: "This larger pressure difference causes stronger westerly winds and thereby more active storm tracks over the North Atlantic. Under such conditions we would expect a higher probability of ice-supersaturation in the uppermost troposphere due to more frequent warm conveyor belts that can induce the formation of ISSRs in the upper troposphere (Spichtinger et al., 2005)."

**Reviewer:** *L608: "we consider the correlation of signs statistically significant". This is a very qualitative conclusion that needs support from a quantitative explanation.*
**Authors:** The entire section was modified in response to a comment by Reviewer #1. We performed a cross-correlation analysis for all three regions and show the results in the modified Figure 16 (formerly 15). This cross-correlation analysis reported a statistically significant correlation between NAO index and deviation of ISSR occurrence from the long term average for the North Atlantic, but no significant correlations for the other two regions. The description of the figure and the conclusions drawn read now:

"For the regions Eastern North America and Europe the correlation between NAO index and ⧄ ISSR fraction is not statistically significant. For the North Atlantic however, the results of the cross-correlation analysis indicate statistical significance at a level of 99%.The obtained correlation of signs is in line with the observation that the occurrence of ice-supersaturation is well correlated with the storm track activity (Spichtinger et al., 2003b; Gettelman et al., 2006; Lamquin et al., 2012)."

[Figure]

**Figure 16.** Correlation analysis with respect to the correlation of signs between NAO index and deviation of ISSR occurrence from the long-term average (⧄ ISSR) for the target regions; numbers indicate he results from the correlation analysis with respect to number of samples n, Pearson R and significance level p.

**Reviewer***: L651: "which then generates more frequently ISSR" is awkward phrasing. Please rewrite.*

[revised manuscript text omitted]

---

## Author Comment (AC4) · 1 Mar 2020

The comment was uploaded in the form of a supplement:
https://www.atmos-chem-phys-discuss.net/acp-2019-735/acp-2019-735-AC4-supplement.pdf

---

## Author Comment (AC5) · 1 Mar 2020

The comment was uploaded in the form of a supplement:
https://www.atmos-chem-phys-discuss.net/acp-2019-735/acp-2019-735-AC5-supplement.pdf

---

## Author Response (AR2)

**Response to Anonymous Referee #2**

We gratefully acknowledge the efforts of Anonymous Referee #2 for reviewing the revised manuscript. Regrettably we missed correcting a number of issues that were already remarked for the first version of the paper. They should now be in good order.

**Point-by point reply**
1) Line 41: what do you mean with "continentally shaped"?
*Reply: "continentally shaped" was removed.*

2) L 63: As it is a threshold, you should write O3 VMR = 120 ppbv.
*Reply: corrected as suggested.*

3) L 88: colder and of higher ABSOLUTE humidity.
*Reply: corrected as suggested.*

4) L 144: Please avoid UTH (cf. 1st review!).
*Reply, now L150: UTH was replaced by „water vapour and relative humidity".*

5) L 145: MOZAIC (typo)
*Reply, now: L151: corrected as suggested.*

6) L 160: I think you mean the EMAC model here with two different vertical resolutions. So please mention EMAC if it is EMAC.
*Reply, now L166: The sentence reads now "Chemistry-climate models like EMAC with vertical resolutions L90MA and L47MA use a vertical grid spacing of 15 - 25 hPa near the extratropical tropopause (Jöckel et al., 2016) which is reflected in the selected vertical resolution of MOZAIC data layers."*

7) Figure 1: x-label should be log10(fraction of data points) and probably the Colour bar should end at "0" (fraction is 1, or 100%; higher fractions are impossible). In the caption remove "probability density function". What we see here isn't a probability density function.
*Reply: corrected as suggested.*

8) L 213: "consolidated" sounds inappropriate to my ears, but I may be wrong.
*Reply, now L221: The term "consolidated" is replaced by the term "merged".*

9) L 251: increasingLY large....
*Reply, now L259: corrected as suggested.*

10) LL 273 and 288: I suggest to write "method" instead of "methodology", because you use it. A methodology cannot be used; it is a discussion or similar about methods.
*Reply, now 286 and 301: corrected as suggested.*

11) Figure 3: Panel a: Can you please add a sentence to explain the negative values of RHice? Panel b) replace "w/t" with "w/o" in the legend.

*Reply: The legend of the figure and the figure caption were updated as requested and the terms "with" and "without" were replaced by "w/t" and "w/o" for consistency.*
*To explain the negative values of $RH_{ice}$ the following sentence was added (line 328): "Unphysical negative values of RHice connected to observations below the LOD of 10% $RH_{ice}$ vanish within the range of uncertainty when applying the IFC method."*

12) L 318: let -> set.
*Reply, now L334: corrected as suggested.*

13) LL 403 and 405: Avoid UTH!
*Reply: The term "UTH" was removed (line 424) or replaced by $RH_{ice}$ (line 426).*

14) L 480: Check punctuation marks.
*Reply, now L507: corrected.*

15) Table 3: It still puzzles me that the ISSR fractions are over all layers higher in the PV coordinates than in the thermal TP co-ordinates. I understand this for the higher layers (TPL and STL), but downwards into the UT I cannot understand why the PC-based ISSR fraction is still much higher than the thermal TP-based fraction even in UT3. If you would take an integral, say from the 350 hPa limit or from your upper temperature limit upwards through the atmosphere, the result should be independent of coordinates. Here it seems that the integral would be much larger in PV coordinates than in the thermal TP-coordinates. Please check your software.
*Reply: We agree that some discussion is needed here and thank the reviewer for this hint. We rephrased the paragraph starting at line 510 as follows:*
*"With reference to the thermal (dynamical) tropopause, the mean ISSR occurrence probability is 31% (38%) in the upper troposphere below the tropopause layer. The observed increases of mean ISSR occurrence probabilities towards the tropopause layer are, however, below statistical significance, and the average values for the respective pressure layers differ for the two tropopause definitions. Our finding that the ISSR occurrence probability is increasing towards the tropopause agrees with results from a previous research aircraft study using the CO - O3 tracer correlation approach, in which the majority (69%) of clear-sky ISSRs was found within the ExTL, while the rest was located below the transition layer (Diao et al., 2015).*
*Since the thermal tropopause is located at higher altitude than the dynamical tropopause, the pressure layers below the thermal tropopause include part of the ExTL which explains the lower ISSR fractions for UT1–3 below the thermal tropopause, compared to UT1-3 below the dynamical tropopause. Sorting the data according to their vertical distance to the respective tropopause results also in different data ensembles for the respective pressure layers because of the strong horizontal variability of RHice along the flight trajectories. This strong horizontal variability explains the different absolute values of ISSR occurrence with respect to the tropopause definitions. For both tropopause definitions, the standard deviation of observed ISSR fractions is largest for the lowest UT layer of the analysed atmospheric region and decreases with increasing altitude."*

16) L 559: "almost similar" sounds funny.
*Reply, now line 597: "almost" was removed.*

17) around line 575: How certain is it that there is no selection bias in the statement that 80% of ISSRs are within cirrus clouds, when you take data mainly from ci-campaigns?
*Reply, starting at line 611: Since we cannot rule out a sampling bias, we removed the statement and will repeat this study once the large IAGOS $RH_{ice} - N_{ice}$ data set is available.*

18) Page 19, 20, 21, several locations: I suggest that you write RHice > 90% and RHice > 105%, similar to RHice>100%, when the uncertainty of the measurement is taken into account here.
*Reply: corrected as suggested on pages 20 - 22*

19) L 653: I suggest to write "... and the 15-years seasonal average".
*Reply, now line 693: corrected as suggested.*

20) L 722: associated with higher absolute humidity.
*Reply, now line 762: corrected as suggested.*

21) L 724: temperature and ABSOLUTE humidity.
*Reply, now line 764: corrected as suggested.*

**Response to Review by Minghui Diao**

This manuscript uses a combined data set from MOZAIC and IAGOS programs to analyze a series of atmospheric conditions in the upper troposphere and lower stratosphere (UT/LS). These conditions include temperature, relative humidity with respect to ice (RHice), water vapor, and ozone. In addition, the relationship of RHi, water vapor volume mixing ratio (H2O VMR) and the occurrence of ice supersaturated regions (ISSR) are analyzed in relation to several tropopause height, i.e., thermal tropopause, dynamic tropopause and chemical tropopause. Long-term time series are used to analyze the trend of ISSR occurrence frequencies. The analysis is focused on three regions – Eastern North America, North Atlantic, and Europe. Overall, the results shown in this study are consistent with previous MOZAIC data, satellite observations of ice cloud occurrence frequency, and research aircraft observations from other field campaigns. The long-term measurements from MOZAIC and IAGOS provide a valuable data set that can be used to compare with reanalysis data, such as ERA-interim.

Overall, the manuscript is well written, especially after addressing the comments from the other three reviewers. The reviewer has some comments below. The manuscript can be considered for publication in ACP after addressing these comments.

Line numbers refer to the clean manuscript without tracked changes, called "acp-2019-735-manuscript-version3".

*Reply: The authors gratefully acknowledge the supportive evaluation by the reviewer and respond to the comments point-by-point.*

**Point-by point reply**

1. For the analysis of RHice versus temperature in Figure 4, the figure cropped off some of the higher RHi. Please include the entire RH data set unless there is a reason against that. In addition, there are some measurements of RHice above liquid saturation line. Can the authors comment on the percentage of the RHice data that is over the liquid saturation line?

*Reply: In the data analysis, the entire data set has been included. The reason for limiting the plot to $RH_{ice} \leq 160\%$ can be taken from Fig. 3. The probability for observations of $RH_{ice} > 160\%$ is below $10^{-4}$ which we consider negligible. In total less than 2% of all data fall in the range of $RH_{ice} > 160\%$ which can be regarded as outliers, mostly related to manoeuvres of the aircraft like change of altitude or turns (see next reply).*

In Section 2.2 RH and O3 instrumentation, the RH measurements have been discussed for their uncertainties, which are 4% and 5% for the middle troposphere and tropopause, respectively. However, Figure 4 shows much higher RHice data than liquid saturation line, such as 40%. Can the authors comment on whether these very high RHi data above liquid saturation are artifacts that are not real? And if so, what is the implication to the interpretation of the RHi accuracy? For example, does this suggest that even with the rigorous method applied to process RH data by comparing with the limit of detection plus 5 ppmv (as stated in Section 2.3), there still are some RHi data that have large biases? Any suggestion on the atmospheric conditions that these large biases usually occur?

*Reply: Good point. Since we asked ourselves during the data analysis about the reason for these values, we traced extraordinary high $RH_{ice}$ for some of the flights where they occurred and could identify them as outliers which are associated to aircraft manoeuvres. So, these data are not considered "real" with respect to the atmospheric conditions the aircraft is sampling in, but they may be "real" for the sensor itself. In case of turns or altitude changes of the aircraft the humidity and temperature sensors inside the Rosemount housing may see conditions different from the conditions assumed for the data inversion as described by Smit et al. (2014), so that the data inversion scheme provides erroneous values, but only for very rare cases. From our understanding, they don't have an impact on the accuracy of the sensor. According to the good practice in data stewardship that we use for the IAGOS data, we did not exclude these values from the data base. This is now also mentioned in the paper as follows (see next reply).*

In line 342, there is a statement of "Obviously, RHice observations remain inside the physical boundaries led by the water saturation line and the line for homogeneous ice nucleation." This is incorrect. Suggest revising to: [X% and Y% of] RHice observations remain inside the physical boundaries led by the water saturation line and the line for homogeneous ice nucleation[, respectively.]

*Reply: We rephrased the sentence starting line 333 as:*
*"About 98% of RHice observations remain inside the physical boundaries set by the water saturation line and the line for homogeneous ice nucleation. The remaining 2% are considered outliers associated to aircraft manoeuvres."*

2. A related suggestion about Figure 4 is to add another sub-panel to Figure 4, which plots RHice versus temperature in scatter plot, and color coded by H2O VMR (ppmv). Please refer to Diao et al. (2014, ACP) figure 5. The observations of RHice in Diao et al. (2014) showed that the H2O VMR of most ISS observations (99 %) are above 20 ppmv. This new figure will help to illustrate the magnitude of H2O VMR for the high RHice seen in the current Figure 4.

*Reply: The information requested here is given in Figure 10 where we separate ISSR from non-ISSR conditions. We see the same behaviour as reported by Diao et al. (2014) with a 1-Percentile value of 25 ppmv for $H_2O$ VMR at the tropopause and above; see Figure 10, panels (c) and (d). Since the paper has already a large number of figures, we abstain from adding another figure.*

*We like to add an information to the point from our side: in the new paper of Krämer et al. (2020), ACPD (Krämer, M., et al. (2020): A Microphysics Guide to Cirrus – Part II: Climatologies of Clouds and Humidity from Observations, Atmos. Chem. Phys. Discuss. 2020, 1-63, DOI: 10.5194/acp-2020-40.), a large data set of clear sky RHice measured from research aircraft is presented, covering the temperature range 182 – 245 (their Figures 6 and 9). In this new data set, ISSRs up to the homogeneous freezing threshold are seen across the entire temperature range. The corresponding $H_2O$ VMRs are about > 2 ppmv in the tropics and > 15 ppmv at mid-latitudes.*

*We added the following sentence related to Fig. 4, starting line 336: "Overall, ice-supersaturated air masses are characterised by $H_2O$ VWR $\geq$ 25 ppmv (1-percentile value), which is in good agreement with observations of $H_2O$ VWR > 15 ppmv (Krämer et al., 2020) and 20 ppmv (Diao et al., 2014), respectively, both reported from research aircraft observations at mid-latitudes for T > 200 K. See also Section 3.3 for more details."*

3. It is interesting that Figure 13 shows similar frequency of ISSR and satellite-based ice cloud fraction. Some of these satellite retrievals are at much coarser horizontal scales than 1 km. Can the author comment on why the different spatial scales do not seem to affect the similarity between the aircraft and satellite data?

*Reply: Please note that the annual cycles shown in Fig. 13 originate from a 15-year average over a large horizontal area. This averaging procedure with respect to time and space will smooth the spatial scales.*

4. It would be helpful to add a sub-panel figure to Figure 1 for number of samples of various temperatures (such as binned by 1 K) for the MOZAIC and IAGOS data used in this study. This new figure will complement the current Figure 1, which only showed the horizontal distribution of samples. Can the authors add precision of accuracy of temperature measurements into Section 2.2? Is vertical velocity measurement available from MOZAIC and IAGOS programs, such as from the aircraft avionic system? They would be helpful for analysis of updrafts and their impacts on the ISSR occurrences.

*Reply: The MOZAIC temperature measurements were analysed in detail by Berkes et al. (2017) (Berkes, F., et al. (2017): In situ temperature measurements in the upper troposphere and lowermost stratosphere from 2 decades of IAGOS long-term routine observation, Atmospheric Chemistry and Physics 17(20), 12495-12508, DOI: 10.5194/acp-17-12495-2017.). For details of the temperature measurement and the T – rage covered by MOZAIC and IAGOS observations, we refer to this publication. Vertical velocity is unfortunately not provided by the avionic system of passenger aircraft.*

*In short, we added the following sentence to the manuscript, starting line 260: "The Pt-100 temperature sensor of the MCH is characterised by an overall uncertainty of the ambient air temperature of ± 0.5 K, which includes the data processing (Berkes et al., 2017). The temperature range encountered during the MOZAIC observations in the Ex-UTLS ranges from 200 K to 245 K at mid-latitudes; see Fig. 9 in Berkes et al. (2017) for details."*

Several places in the introduction and result sections could benefit from more comparisons with previous studies. The reviewer suggests adding a few references that are highly relevant to analysis of ice supersaturated regions. Brackets show where new text is added.

Line 94, "As a result, these air parcels are both colder and of higher relative humidity than the embedded sub-saturated atmosphere." This is not always the case that ISSRs are colder than their horizontally adjacent sub-saturated air. Suggest revising this sentence to:

"As a result, these air parcels are [generally] both colder and of higher relative humidity than the embedded sub-saturated atmosphere. [Using research aircraft observations from 87N to 67S, Diao et al. (2014) (their Figure 7) showed that 73% of the ISSRs have both lower temperature and higher H2O VMR than their horizontally adjacent sub-saturated air, while 27% of the ISSRs show higher temperature and higher H2O VMR than their surroundings.]"
Diao, M., M.A. Zondlo, A.J. Heymsfield, L.M. Avallone, M.E. Paige, S.P. Beaton, T. Campos and D.C. Rogers. "Cloud-scale ice supersaturated regions spatially correlate with high water vapor heterogeneities", Atmospheric Chemistry and Physics, 14, 2639-2656, 2014.

*Reply: Good suggestion! However, we prefer shifting this valuable statement from the introduction section to Section 3.3 (from line 467) where the results are discussed; see our reply to the comment referring to Line 471.*

Line 97, "In the northern mid-latitudes, ISSR occurrence coincides strongly with the storm tracks over the North Atlantic (…), [on the anticyclonic side of the polar jet stream (Diao et al., 2015), and inside the anvil cirrus clouds (D'Alessandro et al. 2017).]"
Diao, M., J.B. Jensen, L.L. Pan, C.R. Homeyer, S. Honomichl, J.F. Bresch and A. Bansemer. "Distributions of ice supersaturation and ice crystals from airborne observations in relation to upper tropospheric dynamical boundaries", Journal of Geophysical Research: Atmosphere, 120, 5101–5121. doi: 10.1002/2015JD023139, 2015.
D'Alessandro, J. J., M. Diao, C. Wu, X. Liu, M. Chen, H. Morrison, T. Eidhammer, J.B. Jensen, A. Bansemer, M.A. Zondlo, J.P. DiGangi. Dynamical conditions of ice supersaturation and ice nucleation in convective systems: a comparative analysis between in-situ aircraft observations and WRF simulations, Journal of Geophysical Research: Atmosphere, 122, doi:10.1002/2016JD025994, 2017.
*Reply: Inserted as suggested, from line 92.*

Line 110, "occurs in most cases below the thermal tropopause … [In addition, research aircraft observations over North America shows that most of the clear-sky ISSRs are located within +/- 500 m from the thermal tropopause (Diao et al., 2015 their figure 10).] This is the same reference as above (doi: 10.1002/2015JD023139).
*Reply: Inserted as suggested, from line 99.*

Line 135, "However, the vertical resolution provided by space-borne instruments in the Ex-UTLS is very limited… [In addition, satellite observations such as NASA AIRS data contain biases in temperature and water vapor retrievals compared with aircraft observations by $1 - 2$ Kelvin and 30%-40% of $H_2O$ VRM, respectively (Diao et al., 2013)]." Adding this sentence helps to show that it is not only the resolution problem but also the accuracy of satellite data that can limit the usage of retrievals of RHice.
Diao, M., L. Jumbam, J. Sheffield, E. Wood and M.A. Zondlo. "Validation of AIRS/AMSU-A water vapor and temperature data with in situ aircraft observations from surface to UT/LS at 87°N–67°S", Journal of Geophysical Research: Atmospheres, 118 (12), 6816–6836, 2013.
*Reply: Good suggestion, thank you; added as proposed from line 134.*

Line 471, "Thus, these air parcels are known as both colder and more humid than the embedding sub-saturated air masses." As mentioned above, this is not always the case. Suggest revising to: Thus, these air parcels are [generally] both colder and more humid than the [surrounding] sub-saturated air masses (Gierens et al., 1999; Spichtinger et al., 2003b)[, although 27% of them were found to be warmer than the surroundings (Diao et al., 2014). Same reference as above for Diao et al. (2014): doi.org/10.5194/acp-14-2639-2014.
*Reply: Inserted as suggested and combined with the comment above. The revised paragraph reads now, starting at line 467:*
*"Thus, these air parcels are generally known as both colder and of higher absolute humidity than the surrounding sub-saturated air masses (Gierens et al., 1999; Spichtinger et al., 2003b), although research aircraft observations from 87°N to 6°S, showed that 73% of the ISSRs have both lower*

*temperature and higher H2O VMR than their horizontally adjacent sub-saturated air, while 27% of the ISSRs show higher temperature and higher H2O VMR than their surroundings (Diao et al., 2014)."*

Line 486, at the end of this paragraph, add "Previously, using the CO-O3 tracer correlation, the majority (69%) of clear-sky ISSRs were found within the transition layer of the extratropical tropopause, while the rest was located below the transition layer (Diao et al., 2015 their figure 14)." This helps to corporates the finding in this paper that most ISSRs were found to locate at the chemical tropopause defined by O3 = 120 ppbv.
*Reply: Good suggestion, thank you; added as proposed, starting at line 513.*

Line 626, "… approx. 80% of the observed ice-supersaturation events are associated with in-cloud conditions. [An analysis of 11 hours of ISSR observations from flight campaigns that did not specifically target cirrus clouds showed that 25% ISSRs were associated with in-cloud conditions (Diao et al., 2015).]" There may be more cirrus clouds being targeted in the campaigns used in Kramer et al. (2016), and therefore citing a different study that did not target on cirrus clouds can be helpful.
*Reply: Since we cannot rule out a sampling bias, we removed the statement and will repeat this study once the large IAGOS $RH_{ice}$ – $N_{ice}$ data set is available.*

Line 813, "… both in number and strength of supersaturation. [Accurately representing the magnitude of ISSR as well as its coexistence with ice crystals are crucial for quantifying radiative forcing, since mistakenly representing ISSR as ice crystals can lead to an average decrease of 2.7 W/m2 in surface radiation (Tan et al., 2016)." Adding this sentence helps to reinforce the importance of representing ISSR accurately in reanalysis or model data.

Tan, X., Y. Huang, M. Diao, A. Bansemer, M. A. Zondlo, J. P. DiGangi, R. Volkamer, and Y. Hu. An assessment of the radiative effects of ice supersaturation based on in situ observations, Geophysical Research Letter, 43, 11,039–11,047, doi:10.1002/2016GL071144, 2016.
*Reply: Very good suggestion, inserted as proposed, from line 782.*

Other minor comments:

Line 151, a few places still use the terminology of "UTH" instead of RHice, such as line 428 and 430. Please change all UTH to RHice to be consistent.
*Reply: The term "UTH" was removed or replaced by $RH_{ice}$.*

Line 202, ERA-I (0:75 degree * 0:75 degree), do you mean 0.75 instead of 0:75?
*Reply: Corrected as suggested, now line 192.*

Line 230, "The conversion to RHice uses the equations by Sonntag (1994)." Many other studies use the equation of saturation pressure with respect to ice (es_ice) from Murphy and Koop (2005). Please provide the differences in RHice calculations between Sonntag (1994) and Murphy and Koop (2005) for the temperature range analyzed in this study.
Murphy, D. M. and Koop, T.: Review of the vapour pressures of ice and supercooled water for atmospheric applications, Q. J. Roy. Meteorol. Soc., 131(608), 1539–1565, doi:10.1256/qj.04.94, 2005.

*Reply: In the temperature range relevant for MOZAIC and IAGOS observations (T > 200 K) the difference of saturation pressures with respect to ice between Murphy and Koop (2005) and Sonntag (1994) are below 0.15%, see Table 1 below. Since the equations by Sonntag (1994) have been implemented in the MOZAIC data analysis from the beginning and the differences are small, we decided to continue with the Sonntag parameterisation for consistency reasons.*

Table 1. Saturation pressures with respect to ice

| Temp (K): | Murphy & Koop (2005) (hPa) | Sonntag (1994) (hPa) | Marti & Mauersberger (1996) (hPa): | Delta Sonntag - MK (%): | Delta MM – MK (%): |
|---|---|---|---|---|---|
| 180 | 5.40E-05 | 5.39E-05 | 5.49E-05 | -0.22696 | 1.76183 |
| 185 | 0.00014 | 0.000135136 | 0.000137944 | -0.190473 | 1.88332 |
| 190 | 0.00032 | 0.000323244 | 0.000330049 | -0.164029 | 1.93764 |
| 195 | 0.00074 | 0.000739718 | 0.000755136 | -0.144531 | 1.93667 |
| 200 | 0.00163 | 0.00162481 | 0.00165768 | -0.12953 | 1.89078 |
| 205 | 0.00344 | 0.00343575 | 0.00350201 | -0.11691 | 1.80926 |
| 210 | 0.00702 | 0.00701275 | 0.00713948 | -0.106561 | 1.69853 |
| 215 | 0.01386 | 0.0138502 | 0.0140808 | -0.0962255 | 1.56708 |
| 220 | 0.02655 | 0.0265267 | 0.0269266 | -0.0858727 | 1.42036 |
| 225 | 0.0494 | 0.0493677 | 0.050029 | -0.0751018 | 1.26345 |
| 230 | 0.0895 | 0.0894395 | 0.0904826 | -0.0638776 | 1.10159 |
| 235 | 0.15809 | 0.158006 | 0.159572 | -0.0523792 | 0.93816 |

Line 311, "The difference between… determines the sensor offset voltage…" Can the authors elaborate on whether a constant voltage offset is applied to these 15 consecutive flights, or the voltage offset is a function sensor temperature? Figure 2 seems to show that the voltage offset is a function of temperature.

*Reply: The sensor offset is determined from the difference between the lower envelope of the observation values from 15 flights and the calibration curve as a function of temperature. So we do neither apply a constant sensor offset voltage, nor an offset voltage as function of temperature, but we determine the correction from a fitting procedure. Normally, the offset voltage does not show any temperature dependence, but for some sensors, we also see an impact of the temperature. However, the method is designed to handle both cases.*

Line 358, MOAZIC should be MOZAIC.
*Reply: Corrected as suggested, line 354.*

Line 427, Focussing should be Focusing.
*Reply: Corrected as suggested, line 448.*

Line 491. "…, indicated by O3 VMR = 120 ppbv at delta_p_TPH = 0 hPa [for ISSRs]". Add "for ISSRs" because the non-ISSRs show O3 VMR = 80 ppbv at thermal tropopause.
*Reply: This is a misunderstanding because in Figure 9 the different values refer to the dynamical and thermal tropopauses. The sentence starting line 389 was rephrased for clarification:*

*"In addition, the results confirm the good agreement between the ERA-Interim thermal tropopause height indicated by $\Delta p_{TPH}$ = 0 hPa, the lowest temperatures detected at $\Delta p_{TPH}$ = 0 hPa (panel (a), blue lines), and the chemical tropopause indicated by $O_3$ VMR = 120 ppbv at $\Delta p_{TPH}$ = 0 hPa (panel (c), blue lines), and thus the consistency of the used data set."*

Line 680-681, The terminologies of TOVS and CALIPSO have not been defined.
*Reply: The sentence starting at line 665 was rephrased as:*
*"Another source of data, but for the occurrence frequency of cirrus clouds originates from long-term analyses of satellite observations (Stubenrauch et al., 2010; Stubenrauch et al., 2013). In their 6-year climatology Stubenrauch et al. (2010) report cirrus cloud coverage fractions for northern mid-latitudes of 35% in January and 27% in July from the Atmospheric Infrared Sounder analysed at the Laboratoire de Météorologie Dynamique in Paris (AIRS-LMD: (2003 to 2008), and respective fractions of 34% and 21% from the TIROS-N Operational Vertical Sounder (TOVS) Path-B cloud retrieval (TOVS – Path B: (1987 to 1995), and 42% and 40% from the Cloud-Aerosol Lidar and Infrared Pathfinder Satellite Observation (CALIPSO: (2006 to 2007)."*

Figure 3b legend still says w/t IFC, should be "w/o" IFC.
*Reply: The legend of the figure and the figure caption were updated as requested and the terms "with" and "without" were replaced by "w/t" and "w/o" for consistency.*

[revised manuscript text omitted]